# TOWARD PRINCIPLED FLEXIBLE SCALING FOR SELF-GATED NEURAL ACTIVATION

**Sudong Cai**[1,3]**, Shuyuan Zheng**[2]⋆**, Bingzhi Chen**[3]**, Shuai Yuan**[3]**, Chuan Xiao**[2]**,
Jianbin Qin**[4]**, Bing Wang**[1]⋆
[1]The Hong Kong Polytechnic University; [2]The University of Osaka;
[3]Beijing Institute of Technology, Zhuhai; [4]Shenzhen University
{sudong.cai,bing-w.wang}@polyu.edu.hk; zheng@ist.osaka-u.ac.jp

## ABSTRACT

Neural networks necessitate nonlinearities to achieve universal approximability. Traditional activation functions introduce nonlinearities through rigid feature rectifications. Recent self-gated variants improve traditional methods in fitting flexibility by incorporating learnable content-aware factors and non-local dependencies, enabling dynamic adjustments to activation curves via adaptive translation and scaling. While SOTA approaches achieve notable gains in conventional CNN layers, they struggle to enhance Transformer layers, where fine-grained context is inherently modeled, severely reducing the effectiveness of non-local dependencies leveraged in activation processes. We refer to this critical yet unexplored challenge as the **non-local tension** of activation. Drawing on a decision-making perspective, we systematically analyze the origins of the non-local tension problem and explore an initial solution to foster a more discriminative and generalizable neural activation methodology. This is achieved by rethinking how non-local cues are encoded and transformed into adaptive scaling coefficients, which in turn recalibrate the contributions of features to filter updates through neural activation. Grounded in these insights, we present **FleS**, a novel self-gated activation model for discriminative pattern recognition. Extensive experiments on various popular benchmarks validate our interpretable methodology for improving neural activation modeling.

## 1 INTRODUCTION

The essence of neural operations in pattern recognition lies in approximating the underlying input-output relationships, which are inherently nonlinear at the level of individual neurons. This necessitates the use of nonlinear activation functions for learning effective neural representations (Cybenko, 1989; Hornik et al., 1989; Hornik, 1991; Leshno et al., 1993). Conventional activation functions (Dugas et al., 2000; Nair & Hinton, 2010), inspired by the neuronal stimulus-response mechanism (Serre et al., 2005; Serre, 2006; Serre et al., 2007; Kouh, 2007), model neural activation in a rigid paradigm. Recent efforts (Hendrycks & Gimpel, 2016; Elfwing et al., 2018; Ma et al., 2021) have been made to enhance the fitting flexibility of activation by leveraging smooth self-gating or incorporating content-aware inductive biases. A typical self-gated activation process $\phi : \mathbb{R} \to \mathbb{R}$ can be expressed as:

$$\phi(\tilde{x}) = \rho(\tilde{x})\tilde{x}, \tag{1}$$

where each $\tilde{x} = \langle \boldsymbol{w}, \boldsymbol{x} \rangle + b \in \mathbb{R}$ represents a projected (affine/linear transformed) feature element, computed from the inner product of a filter vector $\boldsymbol{w}$ and a feature vector $\boldsymbol{x}$, together with a bias term $b$. A weighting function $\rho : \mathbb{R} \to \mathbb{R}$ then assigns a gating weight $\rho(\tilde{x})$ (which typically lies in the interval $(0, 1)$), to recalibrate the pre-activation $\tilde{x}$.

However, current inspirations (Biswas et al., 2022b;a; Misra, 2020; Ramachandran et al., 2018) for activation modeling largely stem from empirical heuristics (e.g., biological cues), while the mechanism of effective activation remains abstract and lacks robust theoretical guidance. This explanatory gap hampers activation modeling and evaluation (Cai, 2024b), significantly limiting further progress.

---

⋆Corresponding authors: Bing Wang and Shuyuan Zheng.
 Code: https://github.com/SudongCAI/FleS

For example, recent approaches have incorporated dynamic non-local cues to enhance self-gated activation and provide additional fitting flexibility (Ma et al., 2021; Chen et al., 2020). Although these methods yield gains in standard CNNs, they fall drastically short in enhancing Transformer layers, which inherently encode fine-grained non-local dependencies outside the activation module. **More intuitively, aggregating gains from non-local cues learned both within and beyond activation processes appears contradictory. We refer to this critical yet unstudied challenge as *non-local tension*.** Grounded in a decision-making perspective, our work is the first to investigate the *non-local tension* problem in self-gated activation and to propose a principled approach for its resolution.

**Intuition 1.1** (Decision-making-inspired interpretation of activation). *Our interpretation is inspired by* multi-criteria decision making*, in particular **grey relational analysis** (Deng, 1982; Liu, 2025) and related models (Rezaei, 2016; Qin et al., 2017; Xu et al., 2020; Joshi & Kumar, 2016), where the goal is to score and rank* alternative solutions *based on a given set of* criteria*, often by comparing each alternative to one or more* ideal solutions *when raw criteria values are* not directly comparable*. We view the **neural affine–activation pipeline** of a single neuron as an instance of this setting and make the following identification. Specifically: (1) each filter (weight vector) $\boldsymbol{w}$ acts as an updatable ideal alternative (solution) that aims to approximate an underlying ideal pattern $\boldsymbol{w}^{\times}$; (2) each feature vector $\boldsymbol{x}$ is treated as a realistic alternative that proposes a candidate update direction for $\boldsymbol{w}$; (3) the channels serve as decision criteria, since both $\boldsymbol{w}$ and $\boldsymbol{x}$ are represented as channel-wise vectors; and (4) the pre-activation $\tilde{x} = \|\boldsymbol{w}\| \|\boldsymbol{x}\| \cos\theta_{\boldsymbol{w},\boldsymbol{x}} + b$ is viewed as an indication signal of importance score for $\boldsymbol{x}$ with respect to $\boldsymbol{w}^{\times}$, where the feature-to-filter similarity $\cos\theta_{\boldsymbol{w},\boldsymbol{x}}$ is the primary contributor, and the filter norm $\|\boldsymbol{w}\|$, feature norm $\|\boldsymbol{x}\|$, and bias term $b$ act as rectifying components. From this perspective, the weighting function $\rho(\cdot)$ acts as a sign-aware recalibration mechanism for indication signals of importance across alternatives and criteria, analogous to determining decision weights in multi-criteria decision making. Under this view, the activation process can be interpreted as a form of directed feature selection, and we leverage this conceptual lens to explore* non-local tension.

In this view, we identify a key underlying factor—the *trivially discriminative gating weights* phenomenon—as a major cause of *non-local tension*: given two **feature (vectors)** $\boldsymbol{x}_i$ and $\boldsymbol{x}_j$ and a **filter (vector)** $\boldsymbol{w}$, all associated with the ideal patterns $\boldsymbol{w}^{\times}$, even if $\boldsymbol{x}_i$ is significantly more important than $\boldsymbol{x}_j$ w.r.t. $\boldsymbol{w}^{\times}$, the smooth weighting function $\rho$ may assign them close gating weights $\rho(\tilde{x}_i)$ and $\rho(\tilde{x}_j)$, leading to only a trivial difference in the recalibration of their importance scores $\tilde{x}_i$ and $\tilde{x}_j$. As gating weights can modulate the contributions of features to filter updates (refer to Sec. A.1 for a detailed discussion), insufficient discriminative power in the assigned weights may limit the effective use of features for filter learning. Consequently, this phenomenon leads to a situation where, when Transformer layers integrate beneficial information provided by non-local cues into a feature, the activation does not correspondingly increase the gating weight to reflect the pre-activation's enhanced importance, thereby causing the *non-local tension* problem (Intuition 3.2)

We identify the saturation behavior of $\rho$ in typical self-gated activation models as a critical cause of the *trivially discriminative gating weights* phenomenon. We refer to this underlying issue as the *convergence limitation*. Specifically, assume that $\tilde{x}$ monotonically reflects the relative importance of a feature $\boldsymbol{x}$ w.r.t. the $\boldsymbol{w}^{\times}$. When both $\tilde{x}_i$ and $\tilde{x}_j$ are relatively large positive values but $\tilde{x}_i$ is significantly larger than $\tilde{x}_j$, the gating weights $\rho(\tilde{x}_i)$ and $\rho(\tilde{x}_j)$ remain distinguishable to effectively recalibrate the pre-activations $\tilde{x}_i$ and $\tilde{x}_j$, enabling the activation mechanism to effectively emphasize or suppress the **contributions of features (to filter update)**. However, saturating gating functions $\rho$, such as the Sigmoid and ERF-based functions, tend to lose discriminability under the above condition, causing the contrast between feature contributions to vanish. Accordingly, we interpret the *non-local tension* problem as a downstream effect of this limitation within self-gated activation.

This identification motivates our novel remedy, **FleS**, which addresses *trivially discriminative gating weights* by modeling flexible scaling coefficients. Guided by decision-making principles, these coefficients adaptively control the bound and steepness of $\rho$, enabling it to attend to informative response intervals in accordance with appropriate non-local cues (see Sec. 4). Consequently, FleS sustains fine-grained recalibration for activation, preserving meaningful differences among relatively important features under *convergence limitation*, thereby mitigating the *non-local tension* challenge.

Our main contributions are threefold: (1) We present new insights that extend decision-making-in-spired activation analysis tools. We identify the convergence limitation as a key cause of *non-local tension*, and highlight flexible scaling as a critical property for overcoming this limitation, enabling more discriminative neural activation. (2) Based on (1), we address the under-explored non-local

tension problem by presenting the novel activation model FleS, which extends the methodology for interpretable neural activation modeling for pattern recognition. (3) Extensive experiments across popular vision and NLP benchmarks validate our new insights, highlighting the effectiveness, versatility, robustness, and extensibility of the FleS methodology, and demonstrating its notable advantages over SOTA activation methods, especially in neural networks with non-local token mixers.

## 2 RELATED WORK

Inspired by the primate stimulus–response mechanism (Serre et al., 2007; 2005; Kouh, 2007), activation functions such as Softplus (Dugas et al., 2000) and its hard approximation ReLU (Nair & Hinton, 2010) were proposed. ReLU, in particular, leverages a rigid $0/1$ mask to activate features, effectively mitigating gradient vanishing in range-limited nonlinearities (*e.g.*, Sigmoid and Tanh) and motivating a series of variants: LeakyReLU (Maas et al., 2013) alleviates "dead" units via a leakage factor, while PReLU (He et al., 2015) learns the negative slope adaptively. More recently, self-gated alternatives relax such rigidity: SiLU (Elfwing et al., 2018) enables soft selection via a sigmoid gate, GELU (Hendrycks & Gimpel, 2016) performs smooth feature recalibration based on Gaussian Error Function (ERF), and Mish (Misra, 2020) combines Tanh and Softplus to form a smooth weighting curve. Although these self-gated functions improve the fitting capability of conventional activation methods, their adaptability remains limited.

SOTA activation designs improve adaptivity via dynamic bounds and context-aware gating. Swish (Ramachandran et al., 2018) (a parametric SiLU) scales inputs within a sigmoid gate; ACON-C (Ma et al., 2021) further adds a learnable bound; Biswas et al. (2022a) extend GELU with ERF-based parametrizations (ErfAct, Pserf via Softplus); SMUs (Biswas et al., 2022b) use a smoothed maximum to enhance ERF-style rectification; and Meta-ACON (Ma et al., 2021) generalizes lightweight channel attention (Hu et al., 2020) for context-conditioned modulation. However, these gates/attentions inject relatively coarse non-local cues; on architectures that already model non-local context (e.g., Transformer layers), this often induces *non-local tension*, limiting gains and applicability.

More related to our work, Cai (2023) interpreted neural activation from a decision-making perspective and identified the overlooked *mismatched feature scoring* (MFS) problem. They demonstrated that standard CNNs, by addressing the MFS problem, can be strengthened to rival advanced Transformers in image recognition solely by leveraging effective activation functions without major architectural changes (Cai, 2024a). Nevertheless, prior interpretations underexplored the contradictory use of different forms of non-local cues, thus struggling to enhance Transformers due to *non-local tension*.

In this work, we introduce new insights to extend activation analysis by elucidating the *convergence limitation* in typical self-gated activation. This enables us to derive the first solution to *non-local tension* by introducing a flexible, FleS-style scaling mechanism with explainability, designed to adaptively recalibrate the bounds and steepness of activation functions to sufficiently leverage the contributions of features.

## 3 NON-LOCAL TENSION CHALLENGE

We elucidate the cause of the *non-local tension* challenge (Fig. 1) in self-gated activation, following a step-by-step analysis. Our investigation is grounded in a simple yet effective decision-making lens, which forms our methodological insights and motivates our FleS activation model as the first solution. We first introduce the preliminaries and then clarify how the *convergence limitation* induces the *trivially discriminative gating weights* phenomenon, which eventually triggers the *non-local tension* challenge. Formal **proofs** and **derivations** supporting this section are provided in Sec. A.

### 3.1 PRELIMINARY

Our analysis is based on the preliminary settings (Cai, 2023; 2024a), which involve simple settings with image inputs: (1) A network includes $T$ sequential learning layers, where each layer is indexed by $\tau = 1, 2, \ldots, T$. (2) Let $\mathbf{X}^\tau \in \mathbb{R}^{C^\tau \times H^\tau \times L^\tau}$ denote the input feature map of layer-$\tau$, where $C^\tau$ and $H^\tau \times L^\tau$ represent the number of channels and the spatial resolution, respectively. (3) The operation at layer-$\tau$ at a spatial location $(h, l) \in \Omega_{H^\tau \times L^\tau}$ is defined as $x_c^{\tau+1}(h, l) := \phi\left(\tilde{x}_c^\tau(h, l)\right)$, where $\boldsymbol{w}^\tau(c) \in \mathbb{R}^{C^\tau}$ and $\boldsymbol{x}^\tau(h, l) \in \mathbb{R}^{C^\tau}$ denote the $c$-th filter vector and the feature vector at location $(h, l)$,

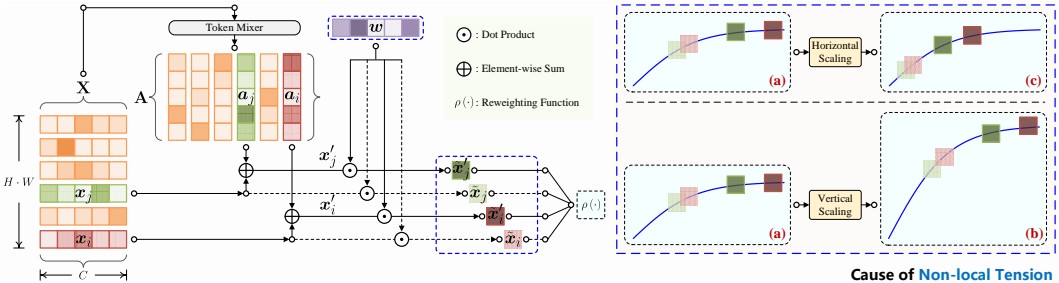

Figure 1: Illustration of the *non-local tension* challenge, and the key intuitions behind our remedy, FleS (Intuition 3.2). **(a)** In a typical self-gated activation $\phi(\tilde{x})$, a saturating, monotonically non-decreasing function $\rho(\tilde{x})$ is used to weight a pre-activation $\tilde{x}$, thus recalibrating the contribution of the corresponding input feature vector $\boldsymbol{x}$ to the filter $\boldsymbol{w}$. When the pre-activations become sufficiently large—even when they are identified as important by the non-local mechanism and pushed to even larger values—$\rho$ assigns almost indistinguishable gating weights to such $\tilde{x}$ that have notably different importance levels. This *convergence limitation* causes the additional contributions brought by non-local cues to be nearly neutralized (*trivially discriminative gating weights*), which in turn triggers *non-local tension*. **(b)** and **(c)** depict an intuitively accessible strategy underlying FleS to alleviate *non-local tension*: an adaptive dual activation scaling, where *vertical scaling* rectifies the activation range and *horizontal scaling* modulates the activation steepness.

respectively. Here, $\Omega_{H^\tau \times L^\tau}$ represents the spatial lattice of $\mathbf{X}^\tau$, and $\tilde{x}_c^\tau(h,l) = \langle \boldsymbol{w}^\tau(c), \boldsymbol{x}^\tau(h,l) \rangle + b_c$ denotes the pre-activation obtained by applying an affine transformation, parameterized by the filter $\boldsymbol{w}^\tau(c)$ and bias $b_c$, to the feature vector $\boldsymbol{x}^\tau(h,l)$. Notably, (i) layer-$\tau$ includes $C_{\tau+1}$ filters and (ii) $\phi : \mathbb{R} \to \mathbb{R}$ represents an activation function. For simplicity, we omit the layer index ($\tau$) and pixel coordinate $(h,l)$ in the following discussion. For example, $\boldsymbol{w}^\tau(c)$, $\boldsymbol{x}^\tau(h,l)$, $\tilde{x}_c^\tau(h,l)$, and $b_c$ become $\boldsymbol{w}$, $\boldsymbol{x}$, $\tilde{x}$, and $b$ respectively. Our analysis focuses on a typical self-gated activation process $\phi : \mathbb{R} \to \mathbb{R}$, which can be expressed by Eq. (1) (*i.e.*, $\phi(\tilde{x}) = \rho(\tilde{x})\tilde{x}$). From a decision-making perspective, we interpret the pre-activation $\tilde{x}$ as an importance measure associated with the input feature $\boldsymbol{x}$ relative to the ideal pattern $\boldsymbol{w}^{\times}$. Furthermore, we treat $\rho(\tilde{x})$ as a gating weight that modulates the response $\tilde{x}$ to emphasize/suppress the contribution of $\boldsymbol{x}$ to the update of $\boldsymbol{w}$. **More intuitively, the more important $\boldsymbol{x}$ is for filter update, the larger the assigned weight $\rho(\tilde{x})$ should be** (the supporting reasons are elaborated in Sec. A.1). To ensure convergence, $\rho$ is commonly assumed to satisfy (Wu, 2022): (1) $\lim_{\tilde{x} \to -\infty} \rho(\tilde{x})\tilde{x} = 0$; (2) $\lim_{\tilde{x} \to +\infty} \rho(\tilde{x}) = \mathcal{M} > 0$. Moreover, our work considers a constant-sign monotonic function $\rho$ to ensure effective self-gated activation by adopting a relevant conclusion ((Cai, 2023, Proposition 2)) and assumes $\rho$ is non-negative without loss of generality. Note that we omit normalization layers (*e.g.*, BN (Ioffe & Szegedy, 2015) and LN (Ba et al., 2016)) in the analysis for simplicity, as their inclusion does not affect the conclusions.

## 3.2 PROBLEM ANALYSIS

*Cause of trivially discriminative weights.* We identify the *trivially discriminative gating weights* phenomenon as a key underlying factor that triggers the *non-local tension*, which widely exists in the $\rho$ of popular/SOTA self-gated activation functions (*e.g.*, Sigmoid in SiLU (Elfwing et al., 2018) and ERF in GELU (Hendrycks & Gimpel, 2016)).

To appropriately modulate the contribution of a feature $\boldsymbol{x}$ to updating a filter $\boldsymbol{w}$ by recalibrating its raw response (*i.e.*, the importance score) $\tilde{x}$, we further assume the weighting function $\rho$ satisfies two basic properties. Specifically, for arbitrary $\boldsymbol{x}_i$ and $\boldsymbol{x}_j$, and a given filter $\boldsymbol{w}$ corresponding to the ideal pattern $\boldsymbol{w}^{\times}$, their responses $\tilde{x}_i$ and $\tilde{x}_j$ are expected to satisfy: (a) *Proper Importance Scoring*: $\tilde{x}_i > \tilde{x}_j$ if $\boldsymbol{x}_i$ is considered more important than $\boldsymbol{x}_j$ relative to $\boldsymbol{w}^{\times}$. (b) *Importance–Weight Alignment*: $\rho(\tilde{x}_i) > \rho(\tilde{x}_j) > 0$ if $\tilde{x}_i > \tilde{x}_j$. However, typical functions $\rho$ satisfying properties (a) and (b) alone are insufficient to guarantee effective self-gated activation due to the *convergence limitation*, which happens if $\rho$ has a fixed upper-bound (see preliminary condition (2)):

**Intuition 3.1** (Convergence limitation). *Specifically, for any $\boldsymbol{x}_i$ and $\boldsymbol{x}_j$, when their raw importance measures $\tilde{x}_i$ and $\tilde{x}_j$, are both sufficiently large, the difference in their gating weights $\rho(\tilde{x}_i)$ and $\rho(\tilde{x}_j)$ can become arbitrarily and trivially small. This indicates that even if $\boldsymbol{x}_i$ contributes significantly*

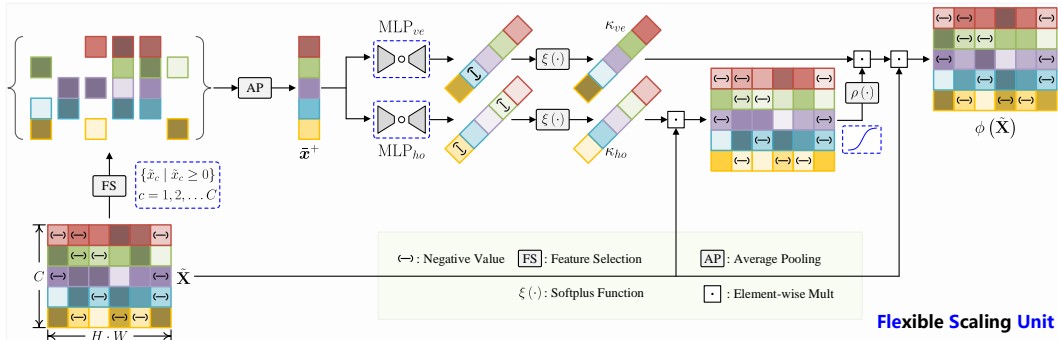

Figure 2: Operational illustration of FleS. Features from different channels are distinguished by distinct color families, where darker shades within each color family indicate higher feature responses.

*more to the update of $\boldsymbol{w}$ than $\boldsymbol{x}_j$, $\rho$ may still fail to assign sufficiently discriminative weights to them, thereby limiting the effective use of them for model learning.*

We refer to this property as *convergence limitation*, which results in *trivially discriminative gating weights* problem, which we characterized as follows:

**Theorem 3.1** (Convergence limitation). *For any $\tilde{x}_i$ and $\tilde{x}_j$ corresponding respectively to $\boldsymbol{x}_i$ and $\boldsymbol{x}_j$ w.r.t. $\boldsymbol{w}$, if $\lim_{\tilde{x} \to +\infty} \rho(\tilde{x}) = \mathcal{M} > 0$, then, for any given $\epsilon > 0$, there must exist a threshold $\mathcal{X}$ such that for all $\tilde{x}_i, \tilde{x}_j > \mathcal{X}$, we have $|\rho(\tilde{x}_i) - \rho(\tilde{x}_j)| < \epsilon$.*

*Cause of non-local tension.* We clarify the *non-local tension* challenge based on the *trivially discriminative gating weights* phenomenon, which hinders the activation module from fully leveraging the context cues already modeled outside activation:

**Intuition 3.2** (*Trivially discriminative gating weights* cause *non-local tension*). *Transformer layers utilize the attention mechanism to capture non-local cues to enhance features. More generally, consider an abstract self-gated activation process with inputs contained non-local cues, where a token mixer casts dynamic translation $\boldsymbol{a}$ to modulate $\boldsymbol{x}$, and then inputs the adjusted features into a neuron (leveraging $\boldsymbol{w}$ and $b$) to produce the finer feature $\tilde{x}'$ for activation:*

$$\tilde{x}' = \langle \boldsymbol{w}, (\boldsymbol{x} + \boldsymbol{a}) \rangle + b; \quad \phi(\tilde{x}') = \rho(\tilde{x}') \tilde{x}'. \tag{2}$$

*Then, suppose we have $\boldsymbol{x}_i, \boldsymbol{x}_j$ such that $\tilde{x}_i \colon \langle \boldsymbol{w}, \boldsymbol{x}_i \rangle + b > 0, \tilde{x}_j \colon \langle \boldsymbol{w}, \boldsymbol{x}_j \rangle + b > 0$, and the adjustments introduced by $\boldsymbol{a}_i$ and $\boldsymbol{a}_j$ effectively push the finer outputs further away from zero, such that $\tilde{x}'_i > \tilde{x}_i$ and $\tilde{x}'_j > \tilde{x}_j$, respectively. Suppose that $\tilde{x}'_i$ and $\tilde{x}'_j$ are sufficiently large such that the differences between the weights assigned by $\rho(\tilde{x}'_i)$ and $\rho(\tilde{x}'_j)$ become trivial (i.e., almost degrading to ReLU's rigid masking process), the learning contributions introduced by $\boldsymbol{a}_i$ and $\boldsymbol{a}_j$ can be significantly neutralized. This leads to a failure to fully exploit non-local cues from informative features for model learning.*

## 4 MODELING

Based on the theoretical awareness, we identify effectively resolving the *non-local tension* challenge as a key avenue to enhancing self-gated activation models in networks that leverage non-local token mixers. In particular, addressing the *non-local tension* challenge hinges on resolving the *convergence limitation*. These insights underpin our novel methodology: FleS-style adaptive scaling mechanism.

### 4.1 PROTOTYPE: FLES-PROTO

*Overview.* Drawing on the insights and conclusions in Sec. 3.2, we introduce FleS prototype:

$$\phi(\tilde{x}) = \kappa_{ve} \rho(\kappa_{ho} \tilde{x}) \tilde{x}, \tag{3}$$

where $\kappa_{ve}$ and $\kappa_{ho}$ denote the vertical and horizontal scaling coefficients, respectively.

*New insights into scaling from non-local cues.* We embody $\kappa_{ve}$ and $\kappa_{ho}$ based on a set of interdependent heuristic insights:

**Intuition 4.1.** *(1)* non-local tension *is a statistical effect: For a given filter, a subset of features that exhibit high importance, relative to the layer-wise feature space,* **may collectively trigger** **non-local** **tension***, **leading to their contributions being underutilized.** (2) Based on (1), any numerical modulation of an activation process for a feature (or its response) should consider its relative relation to a reference feature group.* **Independent modulation of individual activation input is thus** **inadequate for capturing the contextual nature of** non-local tension*. (3) A channel corresponds to the response of a specific filter. Given that different filters may vary significantly in magnitude and direction, two implications follow: (a) Even for the same feature group, response magnitudes can vary across channels* (e.g., **larger-norm filters tend to produce larger-magnitude responses**)*. With a static $\rho$, such responses may be pushed into flatter regions of the gating curve, diminishing the discriminability of activation. (b)* **The triggering interval for** non-local tension *may differ across filters: the same response magnitude may correspond to different* relative *feature contributions under different filters. For example, if two filters share the same direction but differ significantly in norm, a given feature will generally have a greater gradient influence on the smaller-norm filter. Hence, scaling strategies should differentiate between channels to account for such discrepancies. (4) Any (ideal) pattern can be represented by an order-sensitive sequence of filters. Object categories can be viewed as semantically meaningful abstractions of high-level patterns. In particular,* **we posit** **that object category information offers a meaningful basis for grouping effective responses and** **converting them into adaptive scaling coefficients.**

In particular, insights (1), (2), and (3) collectively suggest that: (i) different filters (*i.e.*, channels) may exhibit different triggering zones to *non-local tension* w.r.t. their pre-activation. (ii) *non-local tension* is primarily associated with high-importance features and is negligibly influenced by unimportant ones. Therefore, features should be utilized discriminatively according to their importance levels when extracting statistical (non-local) cues for generating appropriate scale coefficients. (iii) Furthermore, according to (i), different filters may adopt different criteria for "important features" based on their responses. We posit that a feature is considered important if it yields at least a positive response. We formalize this insight via Intuition 4.2 and Proposition 4.1, which play a key role in our methodology.

**Intuition 4.2.** *For a given filter* $\mathbf{w}$*, consider a set of features with positive responses,* $\{\boldsymbol{x} \mid \tilde{x} > 0\}$*, and a set of features with negative responses,* $\{\boldsymbol{x} \mid \tilde{x} < 0\}$*. Assume that: (1) the contributions of features are recalibrated by a sigmoid-like function* $\rho(\tilde{x}) = \beta_{ve}\text{sigmoid}(\beta_{ho}\tilde{x})$*, where* $\beta_{ve}, \beta_{ho} \in \mathbb{R}^+$ *are fixed positive values (note that commonly used* $\rho$ *can be approximated by a sigmoid-like function); and (2)* $\tilde{x} \sim \mathcal{N}(\mu, \sigma)$*, which represents the random variable as a proxy for generating filter responses. Then, (i) positive features are more likely to have higher expected contributions than negative ones after recalibration. Moreover, the more even the distribution is, the more likely positive features are to dominate the overall contribution; (ii) particularly, for an extremely even distribution, the contribution of negative features is negligible compared to that of positive features.*

Proposition 4.1 formalizes Intuition 4.2.

**Proposition 4.1** (Relative recalibration bias)**.** *For conditions assumed in Intuition 4.2, for any fixed* $\mu \in \mathbb{R}$*, the conditional expectation ratio:* $\mathcal{R}(\mu, \sigma) = \frac{\mathbb{E}(\rho(\tilde{x}) \mid \tilde{x} < 0)}{\mathbb{E}(\rho(\tilde{x}) \mid \tilde{x} > 0)}$ *satisfies* $\lim_{\sigma \to \infty} \mathcal{R}(\mu, \sigma) = 0$*.*

Further discussion and proofs of Intuition 4.2 are provided in Sec. B.1.

*Modeling coupled scaling coefficients.* We then follow the above insights to model the FleS-style coupled vertical and horizontal scaling coefficients in a simple yet effective manner (more technical choices are discussed in Sec. E.12):

$$\kappa_{ve} = \text{softplus}\left(\alpha_{ve}\mu\left(\{\bar{x}_c^+\}\right) + \gamma_{ve}\right), \quad \kappa_{ho} = \text{softplus}\left(\alpha_{ho}\mu\left(\{\bar{x}_c^+\}\right) + \gamma_{ho}\right), \quad (4)$$

respectively, where,

$$\bar{x}_c^+ = \text{mean}_{\tilde{x} \in \mathbb{X}_c}\left\{\tilde{x} \mid \tilde{x} \geqslant 0\right\}, \quad (5)$$

denotes the mean filter response of non-negative features within the largest accessible set $\mathbb{X}_c$ of channel-$c$ (*e.g.*, in ImageNet model training, $\mathbb{X}_c$ represents the set of feature vectors in channel-$c$ across the entire mini-batch). We refer to $\bar{x}_c^+$ as the **effective mean response**, which isolates the influence of negative features, preventing it from neutralizing the contribution of positive features (further elaborated in Sec. B.1). Then, $\mu\left(\{\bar{x}_c^+\}\right)$ defines the normalized effective mean response

Table 1: Evaluation of FleS-Proto on ImageNet dataset (Deng et al., 2009).

| Backbone | Activation | #Shuffle | #Params. | FLOPs | Top-1(%)↑ |
|---|---|---|---|---|---|
| Swin-Micro | GELU | — | 21.1M | 2.6G | 78.7 |
| | **FleS-Proto** | — ✓ | 21.1M | 2.6G | **85.2** **77.3** |
| Swin-Base | GELU | — | 87.7M | 15.1G | 83.5 |

\* The Swin-Micro (Liu et al., 2021) backbone is applied, where FleS activation function is compared with the GELU (Hendrycks & Gimpel, 2016) baseline.

across all the channels:

$$\mu\left(\left\{\bar{x}_c^+\right\}\right) = \frac{\bar{x}_c^+}{\frac{1}{C}\sum_{i=1}^{C}\bar{x}_i^+} . \tag{6}$$

$\alpha_{ve}$ and $\alpha_{ho}$, initialized to a small value (*e.g.*, $1 \times 10^{-3}$), are a pair of learnable parameters that scales $\mu\left(\left\{\bar{x}_c^+\right\}\right)$ to introduce adaptability. $\gamma_{ve}$ and $\gamma_{ho}$, initialized to a fixed value (*e.g.*, 0.6, ensuring that $\kappa_{ve}$ and $\kappa_{ho}$ are initially close to 1.0), are learnable parameters to stabilize training in the early stages. Notably, the Softplus functions are applied to impose a smooth positive constraint, as the importance levels measured by pre-activations are sign-sensitive.

**In particular, we identify that this interpretable yet simple design of FleS-Proto can introduce incredible enhancements to Transformer layers** (using Swin-Transformer for example (Liu et al., 2021)) on ImageNet with the standard non-shuffle evaluation setting. Specifically, in the non-shuffle setting, images are arranged in the order of their categories, so that the largest clean channel-specific statistical range corresponds to the entire mini-batch. This provides highly valid channel-wise effective responses $\{\bar{x}_c^+\}$ for calculating $\kappa_{ve}$ and $\kappa_{ho}$.

As shown in Tab. 1, using the standard 300-epoch Transformer-tailored recipe (Touvron et al., 2021; Liu et al., 2021) without auxiliary training data, a small-size Swin-Micro variant (*i.e.*, Swin-$[1, 2, 2, 2]$, consisting of 9 blocks, requires only about 50% of the computational cost of Swin-T (Liu et al., 2021)) achieves significant performance improvements only by replacing GELU with FleS-Proto for activation. It outperforms Swin-B by a remarkable margin (85.2% vs. 83.5%) while requiring only approximately $1/6$ of the computational cost (2.6G FLOPs vs. 15.1G FLOPs). However, when the channel effective mean responses are calculated on a shuffled batch for evaluation, they can no longer provide clean class-specific statistics. As a result, the Top-1 accuracy of FleS-Proto Swin-Micro drops to 77.3%, performing even worse than the vanilla Swin-Micro baseline. These two phenomena motivate our practical design of FleS for broader applicability.

## 4.2 PRACTICAL MODEL: FLES

*Practical modeling.* Experimental evidence in Tab. 1 demonstrates the critical importance of the effective channel mean response in modeling the scaling coefficients. Building upon this awareness, we design FleS, applicable to scenarios where obtaining clean channel-specific statistics is challenging. Specifically, we utilize a lightweight MLP (with a channel reduction ratio of 32 by default) as a channel attribute recorder to compute each scaling coefficient as follows:

$$\kappa_{ve} = \mathrm{MLP}_{ve}\left(\bar{\boldsymbol{x}}^+\right), \quad \kappa_{ho} = \mathrm{MLP}_{ho}\left(\bar{\boldsymbol{x}}^+\right) . \tag{7}$$

Notably, for realistic recognition tasks, we compute each effective channel mean response $\bar{x}_c^+$ over a readily accessible region in practice, $\hat{\mathbb{X}}_c$, where $\bar{\boldsymbol{x}}^+ \in \mathbb{R}^C$ represents the effective channel mean vector, and $\bar{x}_c^+$ its $c$-th element. For example, on ImageNet, we set $\bar{x}_c^+ = \mathrm{mean}_{\tilde{x}\in\boldsymbol{X}_c}\{\tilde{x} \mid \tilde{x} \geqslant 0\}$, where $\boldsymbol{X}_c$ is the $c$-th channel slice of the input feature map $\mathbf{X} \in \mathbb{R}^{H\times W\times C}$. For dense tasks (*e.g.*, object detection), $\hat{\mathbb{X}}_c$ uses a finer neighborhood (*e.g.*, a $9 \times 15$ patch on COCO (Lin et al., 2014)).

As the key to realizing adaptive scaling in realistic pattern recognition tasks, MLPs exhibit translation equivariance, allowing them to detect informative regularities in the effective channel mean vectors $\bar{\boldsymbol{x}}^+ \in \mathbb{R}^C$ across the inputs with complex class distributions (e.g., shuffled single-class images or multi-class road scene images). These regularities are then adaptively converted into scale coefficients. The operational diagram of FleS is illustrated in Fig. 2.

Table 2: Comparison of different activation functions on ImageNet (Deng et al., 2009) with **(left)** Swin-Min (Liu et al., 2021) (Swin-$[1, 1, 1, 1]$) and **(right)** PoolFormer-S12 (Yu et al., 2022) backbones.

| Backbone | Swin-Min (Liu et al., 2021) | | | PoolFormer-S12 (Yu et al., 2022) | | |
|---|---|---|---|---|---|---|
| #Epochs | 120 | | | 300 | | |
| Activation | #Params. | FLOPs | Top-1 (%)↑ | #Params. | FLOPs | Top-1 (%)↑ |
| GELU (Hendrycks et al., 2016) | 11.8M | 1.6G | 68.7 | 11.9M | 1.8G | 77.2 |
| ReLU (Nair & Hinton, 2010) | 11.8M | 1.6G | 68.1 | 11.9M | 1.8G | 76.6 |
| SiLU (Elfwing et al., 2018) | 11.8M | 1.6G | 68.9 | 11.9M | 1.8G | 77.0 |
| Mish (Misra, 2020) | 11.8M | 1.6G | 68.6 | 11.9M | 1.8G | 77.1 |
| Pserf (Biswas et al., 2022a) | 11.8M | 1.6G | 69.0 | 11.9M | 1.8G | NaN |
| SMU (Biswas et al., 2022b) | 11.8M | 1.6G | 68.9 | 11.9M | 1.8G | 77.3 |
| IIEU (Cai, 2023) | 13.4M | 1.6G | 69.5 | 14.3M | 1.8G | 78.6 |
| AdaS (Cai, 2024a) | 13.7M | 1.7G | 69.7 | 15.1M | 1.9G | 78.2 |
| StarReLU (Yu et al., 2024) | 11.8M | 1.6G | 69.1 | 11.9M | 1.8G | 76.8 |
| Meta-ACON (Ma et al., 2021) | 13.4M | 1.6G | 68.3 | 14.3M | 1.8G | 78.0 |
| **FleS (Ours)** | 13.0M | 1.6G | **71.4** | 13.8M | 1.8G | **79.4** |
| **FleS-AdaS** | 14.1M | 1.7G | **73.0** | — | — | — |

\* All competing methods are trained from scratch following the same recipe outlined in ***Implementation details***. "#Epochs" denotes the epochs of training; "NaN" denotes failed training; The baselines use GELU activation.

## 5 EXPERIMENT

We evaluate the effectiveness, versatility, and robustness of our proposed FleS. Experiments are conducted on four major vision benchmarks: ImageNet (Deng et al., 2009) and CIFAR-100 (Krizhevsky, 2009) (I) for standard image classification, ImageNet-LT (Liu et al., 2019) for classification under long-tailed distributions (F), and COCO (Lin et al., 2014) for object detection (J). To further assess its generalizability beyond vision, we validate FleS on **GLUE** (Wang et al., 2018) (C), the popular NLP benchmark. *Moreover, by adapting FleS to context-sensitive semantics in NLP, we validate its extensibility by introducing FleS-SeqGate, a stronger variant with markedly improved performance.*

We evaluate FleS against widely used and SOTA activation functions. From our decision-making lens, **activation functions can be distinguished by the properties of $\rho(\cdot)$ (and thus the overall behavior of $\phi(\cdot)$).** We summarize the main competing activation methods as follows: (i) **Monotonic activation functions** $\phi(\cdot)$ **with discontinuous** $\rho(\cdot)$: Softplus (Dugas et al., 2000); ReLU (Nair & Hinton, 2010); and StarReLU (Yu et al., 2024). (ii) **Static self-gated functions** $\phi(\cdot)$ **with a smooth, monotonic** $\rho(\cdot)$: GELU (Hendrycks & Gimpel, 2016); SiLU (Elfwing et al., 2018); and Mish (Misra, 2020). (iii) **Dynamic self-gated functions** $\phi(\cdot)$ **with a modified** $\rho(\cdot)$ **integrating adaptive components**; Pserf (Biswas et al., 2022a); SMU (Biswas et al., 2022b); Meta-ACON (Ma et al., 2021); IIEU (Cai, 2023); AdaShift (Cai, 2024a); and **FleS**. Notably, FleS can be viewed as a particular form of dynamic self-gated activation (category (iii)), where $\rho(\cdot)$ is constructed from *sign-aware, channel-wise statistics of a reference feature group*, and these statistics are then processed by small MLPs to produce a feature-importance-calibrated adaptive scaling of activations.

Furthermore, we provide methodological insights into the modeling of FleS-Proto/FleS through targeted ablation studies in Sec. 5.2, with further details available in Sec. E.

### 5.1 IMAGENET CLASSIFICATION

**Implementation details.** We evaluate FleS across two representative MetaFormer backbones to validate its effectiveness in alleviating the *non-local tension* challenge (that is, whether it effectively activates neural features that intrinsically contain non-local cues): (1) Swin-Transformer (Liu et al., 2021), the most popular vision Transformer backbone, ranging from Swin-Min (*i.e.*, the minimal Swin model, Swin-$[1, 1, 1, 1]$) to Swin-T (*i.e.*, Swin-$[2 - 2 - 6 - 2]$); and (2) PoolFormer-S12 (Yu et al., 2022), an efficient yet effective MetaFormer model for visual recognition. To further validate the generalizability of our insight and modeling, we also evaluate FleS with (3) ResNet (He et al., 2016), the most prevalent CNN backbone (using ResNet-50). Note that the baseline Swin-Transformers and PoolFormer-S12 use GELU (Hendrycks & Gimpel, 2016) activation functions, and ResNets employ ReLU (Nair & Hinton, 2010), respectively. For fair comparisons, (1) we adopt the standard

Table 3: **(Left)** Comparison of the FleS-enhanced and vanilla GELU Swin-M(icro) (*i.e.*, Swin-[1, 2, 2, 2]), Swin-T, and ViT-B/16 (Dosovitskiy et al., 2021) models on ImageNet (Deng et al., 2009). **(Right)** Comparison of different activation functions on ImageNet using ResNet-50 backbone.

| Activation | Backbone | #Params. | FLOPs | Top-1(%)↑ |
|---|---|---|---|---|
| GELU | | 21.1M | 2.6G | 78.7 |
| SiLU | | 21.1M | 2.6G | 78.6 |
| SMU | Swin-M | 21.1M | 2.6G | 78.8 |
| Mt-ACON | | 24.2M | 2.6G | 78.9 |
| **FleS** | | 23.5M | 2.6G | **80.3** |
| GELU | | 28.3M | 4.4G | 81.3 |
| SiLU | | 28.3M | 4.4G | 81.4 |
| SMU | Swin-T | 28.3M | 4.4G | 81.4 |
| Mt-ACON | | 32.7M | 4.4G | 81.5 |
| **FleS** | | 31.7M | 4.4G | **82.3** |
| GELU | ViT-B/16 | 86.6M | 16.9G | 79.7 |
| **FleS** | | 97.4M | 16.9G | **80.7** |

| Activation | Backbone | #Params. | FLOPs | Top-1(%)↑ |
|---|---|---|---|---|
| ReLU | | 25.6M | 4.1G | 77.2 |
| +SE-Net | | 28.1M | 4.1G | 77.8 |
| PReLU | | 25.6M | 4.1G | 77.1 |
| PWLU | | N/A | N/A | 77.8 |
| SMU | | 25.6M | 4.1G | 77.5 |
| SMU-1 | | 25.6M | 4.1G | 76.9 |
| FReLU | ResNet-50 | 25.7M | 4.0G | 77.6 |
| DY-ReLU | | 27.6M | N/A | 77.2 |
| ACON-C | | 25.6M | 3.9G | 76.8 |
| Mt-ACON | | 25.8M | 3.9G | 78.0 |
| IIEU | | 25.6M | 4.2G | 79.7 |
| AdaS | | 25.6M | 4.1G | 79.9 |
| **FleS** | | 28.1M | 4.1G | **80.1** |

\* Note: FleS with $\kappa_{ve}$ and $\kappa_{ho}$ omitted is equivalent to SiLU (Elfwing et al., 2018).

training-evaluation recipe (Touvron et al., 2021; Liu et al., 2021) for Vision Transformers, except for (2) Swin-Min, for which we reduce the 300-epoch training to 120 epochs (due to time and resource constraints); (3) For ResNets, we adopt the standard CNN training-evaluation recipe (Zhou et al., 2021; Ma et al., 2021) (details are included in Sec. G). Experiments are conducted using four A6000 GPUs.

**Experimental results.** The comparative results of our activation function, FleS, and the SOTA/popular competing methods across various types and sizes of MetaFormer and ResNet backbones are reported in Tabs. 2 and 3 and Tab. 3, respectively, leading to four key observations: (1) On Swin-Transformer and PoolFormer-S12 backbones, FleS demonstrates significant improvements over all existing popular and SOTA activation functions. In particular, **the accuracy gains introduced by FleS over SOTA activation methods are even more pronounced than the improvements of those methods over the GELU baseline. Notably, prior to the introduction of FleS, SOTA methods like Meta-ACON and SMU also exploited non-local information to re-scale the bounds of the activation functions but fell short in enhancing the static self-gated baseline, GELU, for Transformer layers. This validates the critical importance of our interpretable methodological insights for addressing the *non-local tension* challenge.** (2) Then, on ResNet backbones, FleS also demonstrates clear improvements over the prevailing and SOTA activation functions. Although FleS requires additional parameters to capture meaningful statistical cues, it brings only negligible computational cost (measured by FLOPs). (3) FleS exhibits remarkable scalability. It not only works effectively when applied independently but is also capable of boosting other SOTA activation functions. For example, incorporating FleS's flexible scaling scheme into AdaShift (Cai, 2024a) improves Swin-Min. (4) The effectiveness of FleS's flexible scaling is consistently demonstrated across different network architectures and sizes. These results comprehensively validate our insights and practical designs for modeling discriminative neural activation methods. More experimental results and relevant analysis are provided in the appendix.

## 5.2 ABLATION STUDIES

We conduct extensive ablation studies on ImageNet to probe the theoretical and empirical insights underpinning FleS. Here, we present three representative studies and include more studies in Sec. E.

**On w/ or w/o feature statistics for flexible scaling.** We elucidate and demonstrate the significance of channel effective mean responses **(denoted by "channel indicators" in the following text for simplicity)** in driving adaptive scaling for discriminative self-gated activation. Here, we validate this insight by comparing our original FleS with a downgraded FleS variant (denoted as "FleS-DG") that omits the channel indicators. Specifically, the vertical and horizontal scaling coefficients of FleS-DG are defined as $\kappa_{ve} = \text{softplus}(\gamma_{ve})$ and $\kappa_{ho} = \text{softplus}(\gamma_{ho})$, respectively. The comparative results are presented in Tab. 4, leading to two key observations: (1) W/o leveraging statistical cues in channel indicators, FleS-DG exhibits significantly lower accuracy than FleS. (2) Despite this, FleS-DG still

Table 4: Ablation studies on **(left)** w/ or w/o the channel effective mean intensities $\{\bar{x}_c^+\}$ for modeling FleS coefficients; and **(right)** mining statistical cues within positive feature elements for FleS.

| Activation | Backbone | #Params. | FLOPs | Top-1(%)↑ | Activation | Backbone | #Params. | FLOPs | Top-1(%)↑ |
|---|---|---|---|---|---|---|---|---|---|
| GELU | | 11.8M | 1.6G | 68.7 | GELU | | 11.8M | 1.6G | 68.7 |
| FleS-DG | Swin-Min | 11.8M | 1.6G | 69.1 | FleS-P&N | Swin-Min | 13.0M | 1.6G | 69.8 |
| **FleS** | | 13.0M | 1.6G | **71.4** | **FleS** | | 13.0M | 1.6G | **71.4** |

* "FleS-DG" denotes the FleS variant omitting $\{\bar{x}_c^+\}$ in generating scaling coefficients.

* The FleS variant "FleS-P&N" averages positive and negative features for calculation of channel indicators.

outperforms the GELU baseline. These findings indicate that (1) statistical cues provided by channel indicators are critical guidance; and (2) scaling coefficients remain beneficial for self-gated activation, even in the absence of channel statistical cues. These validate our insights.

**On mining statistical cues in positive features.** We elucidate the necessity of treating positive and negative features differently in the modeling of non-local cues (in Sec. 4.1), aiming to prevent the neutralization effect induced by negative features on positive ones in adaptive scaling. Accordingly, we propose excluding negative responses when computing channel indicators as a simple yet effective design strategy. We validate this insight by comparing our original FleS with a tailored FleS variant (denoted as "FleS-P&N") that averages both positive and negative responses to calculate channel indicators. As reported in Tab. 4, FleS-P&N improves upon the GELU baseline but yields clearly inferior results to the original FleS. This supports our theoretical analysis.

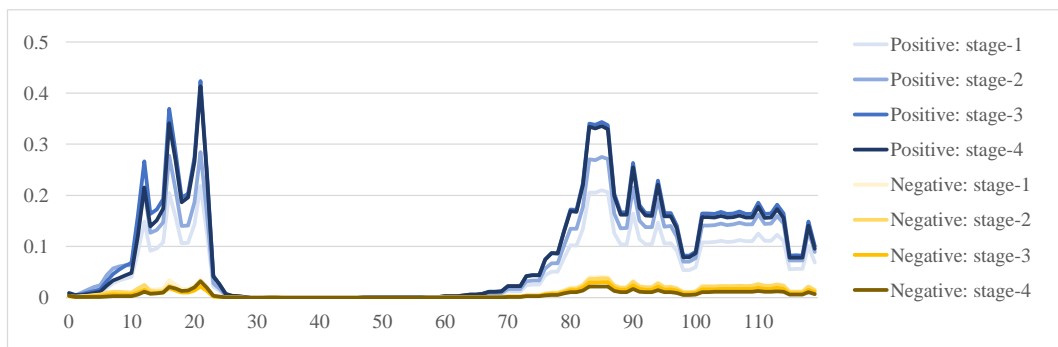

Figure 3: Illustration of positive versus negative gradient magnitudes over 120 training epochs.

**On empirical positive–negative gradient contributions.** We conduct a tailored study on ImageNet with Swin-Min + FleS and, for each epoch and stage, measuring the ratio between the mean gradient magnitudes at positions with positive versus negative FleS outputs. Averaged over training, gradients on positive responses are much larger than on negative ones: the mean positive-to-negative ratios are about $5.3\times$, $7.9\times$, $12.7\times$, and $13.8\times$ for stages 1-4, respectively, with deeper stages exhibiting stronger asymmetry. Over epochs, this gap widens from early to mid training and then remains high: ratios increase roughly from $3$–$9\times$ (epochs 0-9) to $6$–$15\times$ (epochs 40-79), and stay clearly above the early-phase levels in the late phase (epochs 80-119). Thus, as training proceeds, positive responses increasingly dominate the effective gradient budget, especially in deeper stages. These align with our sign-aware indicator design: the dominant optimization signal already lies on the positive side, so emphasizing it in the indicator helps avoid cancellation effects. Fig. 3 visualizes these trends.

## 6 CONCLUSION

In this work, we identified and formalized the *convergence limitation* inherent in self-gated neural activation, showing that it gives rise to the unstudied yet critical *non-local tension* challenge, which we found to hinder the potential of activation functions in enhancing modern neural networks. Grounded in decision-making principles and their encouraged insights, we derived the FleS-style dynamic scaling scheme that provided the first principled remedy to *non-local tension*. Comprehensive experiments on various popular benchmarks, together with targeted ablation studies, verified its effectiveness, generalizability, robustness, and extensibility, indicating strong potential to advance interpretable activation modeling for pattern recognition.

ACKNOWLEDGMENTS

This work was jointly supported by the Research Grants Council of Hong Kong (Nos. 25206524, 15212925); the National Natural Science Foundation of China (Nos. 42301520, 62302172); the Innovation and Technology Fund (No. PRP/068/23FX); the Platform Project of Unmanned Autonomous Systems Research Centre (No. P0049516); the Seed Projects of Smart Cities Research Institute (Nos. P0051028, P0054511); JSPS KAKENHI (No. JP25K21207); JST CREST (No. JPMJCR22M2); and the Guangdong Basic and Applied Basic Research Foundation (Nos. 2025A1515010225, 2025A1515060001).

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

# A DISCUSSION & PROOFS OF SECTION 3

## A.1 RATIONALE BEHIND FEATURE CONTRIBUTION RECALIBRATION FOR FILTER UPDATE IN NEURAL SELF-GATED ACTIVATION

In this appendix, we clarify the rationale behind treating the gating weights $\rho(\tilde{x})$ as a key factor in recalibrating the influence (*i.e.*, contribution) of features to the update of filters. Note that this appendix inherits the preliminary settings and assumptions proposed in the main paper.

*Motivation.* Grounded in the decision-making perspective, we revisit the physical meaning of neural activation from the viewpoint of back-propagation. Specifically, at a learning layer $\tau$, we consider the activation process, $\phi(\tilde{x})$, as a form of contribution recalibration: for a given input feature vector $\boldsymbol{x}$ and a filter vector $\boldsymbol{w}$, the pre-activation (*e.g.*, $\tilde{x} = \langle \boldsymbol{w}, \boldsymbol{x} \rangle$, where we omit the bias term $b$ to simplify the following analysis) can be viewed as a similarity-based score that estimates the relative importance of $\boldsymbol{x}$ with respect to $\boldsymbol{w}$. The activation function then accordingly assigns a gating weight $\rho(\tilde{x})$ to modulate this raw score, effectively emphasizing or suppressing the influence of $\boldsymbol{x}$ on the update of $\boldsymbol{w}$.

*Proxy objective.* Then, we consider a recognition scenario and build upon several key behavioral properties of the cross-entropy (CE) loss:

(1) The CE gradient is dominated by prediction error, producing a clear push-pull effect that penalizes incorrect classes and pulls toward the correct one;

(2) At the logit layer, the gradient can be simplified as the product of a prediction deviation term and the input feature, exhibiting a locally linear response;

(3) In early training or at shallower layers, the back-propagated CE gradient approximately retains a first-order structure;

(4) Each sample's contribution to parameter updates can be modeled independently, without involving global interactions or higher-order coupling.

Based on these properties, we adopt the following proxy objective to approximate the CE gradient behavior at layer $\tau$:

$$\mathcal{L}^{\mathrm{proxy}} = \frac{1}{2} \left( \phi(\tilde{x}) - \phi(\tilde{x}^*) \right)^2 . \tag{8}$$

Notably, we let

$$\tilde{x}^* = \langle \boldsymbol{w}^*, \boldsymbol{x} \rangle , \tag{9}$$

denoting the virtual ideal feature-filter response, as a reference to help simplify the representation of the effective loss. Here, $\boldsymbol{w}^*$ denotes the ideal pattern (*i.e.*, effective objective) with respect to the filter $\boldsymbol{w}$.

Then, the gradient of $\mathcal{L}^{\mathrm{proxy}}$ about $\boldsymbol{w}$ can be calculated as:

$$\begin{aligned}
\nabla_{\boldsymbol{w}} \mathcal{L}^{\mathrm{proxy}} &= \frac{\partial \mathcal{L}^{\mathrm{proxy}}}{\partial \left( \phi(\tilde{x}) - \phi(\tilde{x}^*) \right)} \cdot \frac{\partial \left( \phi(\tilde{x}) - \phi(\tilde{x}^*) \right)}{\partial \boldsymbol{w}} \\
&= \frac{\partial \mathcal{L}^{\mathrm{proxy}}}{\partial \left( \phi(\tilde{x}) - \phi(\tilde{x}^*) \right)} \cdot \frac{\partial \phi(\tilde{x})}{\partial \boldsymbol{w}} \\
&= \frac{\partial \mathcal{L}^{\mathrm{proxy}}}{\partial \left( \phi(\tilde{x}) - \phi(\tilde{x}^*) \right)} \cdot \frac{\partial \left( \rho(\tilde{x}) \cdot \tilde{x} \right)}{\partial \tilde{x}} \cdot \frac{\partial \tilde{x}}{\partial \boldsymbol{w}} \\
&= \left( \phi(\tilde{x}) - \phi(\tilde{x}^*) \right) \cdot \left( \rho'(\tilde{x}) \cdot \tilde{x} + \rho(\tilde{x}) \right) \cdot \boldsymbol{x} .
\end{aligned} \tag{10}$$

As the reference group, we consider the case where the activation is removed. In this case, the gradient is calculated as:

$$\nabla_{\boldsymbol{w}} \mathcal{L}^{\mathrm{proxy}} = (\tilde{x} - \tilde{x}^*) \cdot \boldsymbol{x} . \tag{11}$$

*Feature contribution recalibration.*

**Intuition A.1.** *In particular, for a significantly large $\tilde{x}$—that is, when the corresponding $x$ is of relatively high importance with respect to $w$ (or $w^*$ in a physical sense)—under the condition where we expect the potential occurrence of the* non-local tension *phenomenon, we identify that the proxy gradient $\nabla_w \mathcal{L}^{\mathrm{proxy}}$ can be further approximately simplified as follows:*

$$\nabla_w \hat{\mathcal{L}}^{\mathrm{proxy}} = (\phi(\tilde{x}) - \phi(\tilde{x}^*)) \cdot \rho(\tilde{x}) \cdot x, \tag{12}$$

i.e., *the term $\rho'(\tilde{x}) \cdot \tilde{x}$ is negligible.*

We formalize the conclusion in Intuition A.1 by Proposition A.1.

**Proposition A.1.** *For a self-gated activation function $\phi(\tilde{x}) = \rho(\tilde{x})\tilde{x}$ that satisfies:*

*(1) $\lim_{\tilde{x} \to -\infty} \rho(\tilde{x})\tilde{x} = 0$;*

*(2) and $\lim_{\tilde{x} \to +\infty} \rho(\tilde{x}) = \mathcal{M} > 0$.*

*we have:*

$$\lim_{\tilde{x} \to +\infty} \rho'(\tilde{x}) \cdot \tilde{x} = 0. \tag{13}$$

*Proof.* **core insight.** $\lim_{\tilde{x} \to +\infty} \rho'(\tilde{x}) \cdot \tilde{x} = 0 \longleftrightarrow \rho'(\tilde{x}) = o\left(\frac{1}{\tilde{x}}\right)$ ($\tilde{x} \to +\infty$) (*i.e.*, the term $\rho'(\tilde{x})$ is an infinitesimal of higher order than $\frac{1}{\tilde{x}}$). That is, if $\rho(\tilde{x})$, where $\lim_{\tilde{x} \to +\infty} \rho(\tilde{x}) = \mathcal{M}$, grows more slowly than $\ln(\tilde{x})$ by an order (or orders) of magnitude as $\tilde{x} \to +\infty$, then we have $\lim_{\tilde{x} \to +\infty} \rho'(\tilde{x}) \cdot \tilde{x} = 0$.

***Proof by contradiction.*** Our following proof is carried out using the Fundamental Theorem of Calculus, the Lagrange Mean Value Theorem, and contradiction.

Suppose $\exists \mathcal{G} > 0$ and an unbounded increasing sequence $\{\tilde{x}_n\}$ such that:

$$\left| \tilde{x}_n \cdot \rho'(\tilde{x}_n) \right| > \mathcal{G} \quad, \forall n, \tag{14}$$

*i.e.*, suppose that the convergence of $\rho'(\tilde{x}) \cdot \tilde{x}$ is not ensured (the contradictory case to Proposition A.1), thus, we have:

**Corollary A.2.** $|\rho'(\tilde{x}_n)| > \frac{\mathcal{G}}{\tilde{x}_n}$.

Further, assume that $\rho$ is differentiable on the interval $[\tilde{x}_n, 2\tilde{x}_n]$, by using Corollary A.2 and Lagrange Mean Value Theorem, we have: $\exists \xi_n \in [\tilde{x}_n, 2\tilde{x}_n] \quad, \tilde{x}_n > 0$ such that

$$\rho(2\tilde{x}_n) - \rho(\tilde{x}_n) = \rho'(\xi_n) \cdot (2\tilde{x}_n - \tilde{x}_n) = \rho'(\xi_n) \cdot \tilde{x}_n, \tag{15}$$

therefore,

$$\left| \rho(2\tilde{x}_n) - \rho(\tilde{x}_n) \right| = \left| \rho'(\xi_n) \right| \cdot \tilde{x}_n > \frac{\mathcal{G}}{\xi_n} \cdot \tilde{x}_n \geqslant \frac{\mathcal{G}}{2\tilde{x}_n} \cdot \tilde{x}_n = \frac{\mathcal{G}}{2}. \tag{16}$$

This leads to a contradictory conclusion to the pre-assumed condition (2) (*i.e.*, $|\rho(2\tilde{x}_n) - \rho(\tilde{x}_n)|$ is a convergent function).

In other words, for any given small value $\epsilon > 0$, there exists a threshold $\chi$ such that the inequality $\rho'(\tilde{x}) \cdot \tilde{x} < \epsilon$ holds for all $\tilde{x} > \chi$. This conclusion is equivalent to $\lim_{\tilde{x} \to +\infty} \rho'(\tilde{x}) \cdot \tilde{x} = 0$, and explains why we treat the term $\rho'(\tilde{x}) \cdot \tilde{x}$ as negligible when $\tilde{x}$ is sufficiently large.

This completes the proof of Proposition A.1. □

Then, under the assumption that $\tilde{x}$ is sufficiently large, we have:

$$\begin{aligned}
\nabla_w \hat{\mathcal{L}}^{\mathrm{proxy}} &= (\phi(\tilde{x}) - \phi(\tilde{x}^*)) \cdot \rho(\tilde{x}) \cdot x \\
&= (\rho(\tilde{x})\tilde{x} - \rho(\tilde{x}^*)\tilde{x}^*) \cdot \rho(\tilde{x}) \cdot x \\
&= \mathcal{M}\rho(\tilde{x})(\tilde{x} - \tilde{x}^*) \cdot x \\
&= \mathcal{M}\rho(\tilde{x}) \nabla_w \mathcal{L}^{\mathrm{proxy}}_{\mathrm{w/oAct}}.
\end{aligned} \tag{17}$$

That is, $\mathcal{M}\rho(\tilde{x})$ can be interpreted as the effect of recalibration. Furthermore, since $\mathcal{M}$ is a fixed value for each $\rho$, we posit that $\rho(\tilde{x})$ acts as the primary contributor to feature contribution recalibration through a self-gated activation process.

This supports the intuitions and insights we proposed based on feature contribution recalibration.

## A.2 PROOF OF THEOREM 3.1

In the main paper, we present Theorem 3.1 to formalize the problem of *convergence limitation*, which may underlie the *non-local tension* challenge and is inherently present in typical self-gated functions characterized by the general form:

$$\phi\left(\tilde{x}\right) = \rho\left(\tilde{x}\right)\tilde{x}, \tag{18}$$

where $\tilde{x} = \langle \boldsymbol{w}, \boldsymbol{x} \rangle + b \in \mathbb{R}$ is a given feature element (scalar), derived from the inner product of the filter $\boldsymbol{w}$ and feature vectors $\boldsymbol{x}$ along with a bias term $b$, and $\rho : \mathbb{R} \to \mathbb{R}$ assigns a score $\rho\left(\tilde{x}\right)$ to weight $\tilde{x}$. Typically, the weighting function $\rho$ is commonly required to satisfy that (Wu, 2022):

(1) $\lim_{\tilde{x}\to-\infty} \rho\left(\tilde{x}\right)\tilde{x} = 0$;

(2) and $\lim_{\tilde{x}\to+\infty} \rho\left(\tilde{x}\right) = \mathcal{M} > 0$.

**Retrospect.** For ease of reference, we restate Theorem 3.1 from the main text as Theorem A.3 here.

**Theorem A.3** (Convergence limitation: restatement of Theorem 3.1). *For any $\tilde{x}_i$ and $\tilde{x}_j$ corresponding respectively to $\boldsymbol{x}_i$ and $\boldsymbol{x}_j$ w.r.t. $\boldsymbol{w}$, if $\lim_{\tilde{x}\to+\infty} \rho\left(\tilde{x}\right) = \mathcal{M} > 0$, then, for any given $\epsilon > 0$, there must exist a threshold $\mathcal{X}$ such that for all $\tilde{x}_i, \tilde{x}_j > \mathcal{X}$, we have $|\rho(\tilde{x}_i) - \rho(\tilde{x}_j)| < \epsilon$.*

*Proof.* By the definition of limits, the given assumption:

$$\lim_{\tilde{x}\to+\infty} \rho\left(\tilde{x}\right) = \mathcal{M}, \tag{19}$$

where $\mathcal{M} > 0$, implies that: for any $\epsilon > 0$, there exists a sufficiently large scalar $\mathcal{X}$ such that for all $\tilde{x} > \mathcal{X}$, we have:

$$|\rho\left(\tilde{x}\right) - \mathcal{M}| < \frac{\epsilon}{2}. \tag{20}$$

Therefore, $\forall \tilde{x}_i, \tilde{x}_j > \mathcal{X}$, applying the above conclusion, we have:

$$|\rho\left(\tilde{x}_i\right) - \mathcal{M}| < \frac{\epsilon}{2}, \quad |\rho\left(\tilde{x}_j\right) - \mathcal{M}| < \frac{\epsilon}{2}. \tag{21}$$

Based on the triangle inequality, we have:

$$|\rho\left(\tilde{x}_i\right) - \rho\left(\tilde{x}_j\right)| < \frac{\epsilon}{2} + \frac{\epsilon}{2} = \epsilon. \tag{22}$$

This completes the proof. □

## B DISCUSSION & PROOFS OF SECTION 4

### B.1 DISCUSSION AND PROOFS OF INTUITION 4.1 AND PROPOSITION 4.1

*Overview.* We propose Proposition 4.1 to help clarify our intuition that inspires the modeling of the non-local indicators $\{\bar{x}_c^+\}$ (Eq. (5) in the main text), *i.e.*, we posit that positive and negative features should be used in a discriminative manner to produce the non-local indicators for inducing a FleS-style adaptive scaling scheme, so as to prevent the contributions of positive features from being neutralized by negative ones. In particular, we identify that positive features tend to have relatively higher accumulated contributions than negative features, and this advantage becomes more pronounced when the input distribution is relatively flat.

Before proving Proposition 4.1, we first discuss a more general case to clarify Intuition 4.1. All discussions and proofs in this appendix inherit the assumptions and pre-conditions established therein.

*Advantage of positive contribution after activation.* Through self-gated activation, the contributions of positive and negative features are relatively emphasized and suppressed, respectively, thereby giving positive features a relative advantage over negative ones in their overall influence.

This finding can be clarified as follows:

**Proposition B.1.** *Let*

(1) $\rho(\tilde{x}) = \beta_{ve}\text{sigmoid}(\beta_{ho}\tilde{x})$ *denote a sigmoid-like function, where* $\beta_{ve}, \beta_{ho} > 0$ *and* $\tilde{x} \in \mathbb{R}$;

(2) $\tilde{x} \sim \mathcal{N}(\mu, \sigma)$ *representing the random variable as a proxy for generating filter responses;*

(3) $\varphi(\tilde{x}; \mu, \sigma)$ *denote the Gaussian density with mean* $\mu$ *and standard deviation* $\sigma$;

(4) $f(\mu, \sigma) = \int_0^{+\infty} \rho(\tilde{x})\,\varphi(\tilde{x}; \mu, \sigma)\,\mathrm{d}\tilde{x} - \int_{-\infty}^0 \rho(\tilde{x})\,\varphi(\tilde{x}; \mu, \sigma)\,\mathrm{d}\tilde{x}$ *define the cumulative gap of gating weights (i.e., indicator of recalibrated contribution).*

*Then, for any given* $\sigma > 0$:

(i) $\forall \mu \geqslant 0$, *we have* $f(\mu, \sigma) > 0$;

(ii) $\exists \mu_0 < 0$ *such that* $\forall \mu \in (\mu_0, 0)$, $f(\mu, \sigma) > 0$.

*Proof.* First, Proposition B.1(i) clearly holds, we then omit its detailed proof, as $\varphi(\tilde{x}; \mu, \sigma)\,|_{\mu \geqslant 0}$ is symmetric about $\tilde{x} = \mu$ and $\rho(\tilde{x}) > \rho(-\tilde{x})$ for any $\tilde{x} > 0$. This ensures $\int_0^{+\infty} \rho(\tilde{x})\,\varphi(\tilde{x}; \mu, \sigma)\,\mathrm{d}\tilde{x} > \int_{-\infty}^0 \rho(\tilde{x})\,\varphi(\tilde{x}; \mu, \sigma)\,\mathrm{d}\tilde{x}$, *i.e.*, $f(\mu, \sigma)\,|_{\mu \geqslant 0} > 0$.

We thereby focus on Proposition B.1(ii) in the following.

Fix any $\sigma > 0$. By the continuity of $\rho(\tilde{x})$ and $\varphi(\tilde{x}; \mu, \sigma)$, and by dominated convergence on compact $\mu$-intervals, the mapping $\mu \mapsto f(\mu, \sigma)$ is continuous. Invoking Item (i), we have $f(0, \sigma) > 0$. Hence, by continuity at $\mu = 0$, there exists $\delta = \delta(\sigma) > 0$ such that $|\mu| < \delta$ implies $f(\mu, \sigma) > 0$. In particular, letting $\mu_0 := -\delta < 0$ yields $f(\mu, \sigma) > 0$ for all $\mu \in (\mu_0, 0)$.

This completes the proof. $\qquad\square$

**Remark B.1.** *Proposition B.1 indicates that, under a standard self-gated activation, the aggregate influence of positive features tends to be amplified relative to negatives, thus tending to yield higher contributions to filter updates during the activation process.*

Furthermore, we present an extension of Proposition B.1 as follows. This conclusion helps clarify how the relative advantage of the contributions of positive features becomes more pronounced when the input distribution is relatively flat:

**Proposition B.2.** *Consider* $f(\mu, \sigma)$ *that admits a negative interval (this condition is satisfiable,* e.g., *for* $\rho(\tilde{x}) = \text{sigmoid}(\tilde{x})$ *and* $\sigma = 1$, *then, numerically we can verify that* $f(\mu, \sigma)\,|_{\mu < -1} < 0$). *Let* $\mu_0(\sigma)$ *denote the unique negative root of* $f(\mu, \sigma) = 0$. *Then the function* $\mu_0(\sigma)$ *is strictly decreasing in* $\sigma > 0$.

*Proof.* We apply the implicit function theorem to the identity $f(\mu_0(\sigma), \sigma) = 0$. Then

$$\frac{d\mu_0}{d\sigma} = -\frac{\partial f/\partial \sigma}{\partial f/\partial \mu}. \tag{23}$$

We compute the partial derivatives of $f$. First, note that

$$\frac{\partial}{\partial \sigma}\varphi(\tilde{x}; \mu, \sigma) = \varphi(\tilde{x}; \mu, \sigma)\left(-\frac{1}{\sigma} + \frac{(\tilde{x} - \mu)^2}{\sigma^3}\right). \tag{24}$$

Hence,

$$\frac{\partial f}{\partial \sigma} = \int_0^\infty \rho(\tilde{x})\varphi(\tilde{x}; \mu, \sigma)\left(\frac{(\tilde{x} - \mu)^2}{\sigma^3} - \frac{1}{\sigma}\right)\mathrm{d}\tilde{x} - \int_{-\infty}^0 \rho(\tilde{x})\varphi(\tilde{x}; \mu, \sigma)\left(\frac{(\tilde{x} - \mu)^2}{\sigma^3} - \frac{1}{\sigma}\right)\mathrm{d}\tilde{x}. \tag{25}$$

Because $\rho(\tilde{x})$ is monotonic and much larger on $\tilde{x} > 0$ than on $\tilde{x} < 0$, and $(\tilde{x} - \mu)^2$ grows quickly on $\tilde{x} > 0$, the first integral dominates. Thus, $\frac{\partial f}{\partial \sigma} > 0$.

Next, we compute the derivative with respect to $\mu$ as:

$$\frac{\partial}{\partial \mu}\varphi(\tilde{x}; \mu, \sigma) = \frac{(\tilde{x} - \mu)}{\sigma^2}\varphi(\tilde{x}; \mu, \sigma). \tag{26}$$

Therefore, we have:

$$\frac{\partial f}{\partial \mu} = \frac{1}{\sigma^2} \left[ \int_0^\infty \rho(\tilde{x})(\tilde{x} - \mu)\varphi(\tilde{x}; \mu, \sigma) \, \mathrm{d}\tilde{x} - \int_{-\infty}^0 \rho(\tilde{x})(\tilde{x} - \mu)\varphi(\tilde{x}; \mu, \sigma) \, \mathrm{d}\tilde{x} \right] . \tag{27}$$

When $\mu < 0$, both integrals involve $(\tilde{x} - \mu) > 0$, but the second is weighted by smaller $\rho(\tilde{x})$. Hence the second term dominates, and $\frac{\partial f}{\partial \mu} < 0$.

Combining the above, we conclude that:

$$\frac{d\mu_0}{d\sigma} = -\frac{\partial f/\partial \sigma}{\partial f/\partial \mu} < 0 \, , \tag{28}$$

and therefore $\mu_0(\sigma)$ is strictly decreasing in $\sigma$.

This completes the proof. $\qquad\square$

Notably, Propositions B.1 and B.2 underpin our intuition for Proposition 4.1, which can be regarded as an extreme case of Proposition B.1. We now address Proposition 4.1 and also clarify two related conclusions (*i.e.*, Lemma B.4 and Proposition B.5) to further support our intuition.

**Retrospect.** For ease of reference, we restate Proposition 4.1 from the main text as Proposition B.3 here.

**Proposition B.3** (Relative contribution recalibration bias: restatement of Proposition 4.1). *For conditions assumed in Proposition B.1, for any given $\mu \in \mathbb{R}$, the conditional expectation ratio:* $\mathcal{R}(\mu, \sigma) = \frac{\mathbb{E}(\rho(\tilde{x})|\tilde{x}<0)}{\mathbb{E}(\rho(\tilde{x})|\tilde{x}>0)}$ *satisfies* $\lim_{\sigma \to \infty} \mathcal{R}(\mu, \sigma) = 0$.

*Proof.* We first consider $\rho(\tilde{x}) = \text{sigmoid}(\tilde{x})$, *i.e.*, $\beta_{ve} = 1$ and $\beta_{ho} = 1$ for simplicity, and then generalize the conclusion to general sigmoid-like functions.

Let $\mathcal{C}_+(\mu, \sigma) = \int_0^{+\infty} \rho(\tilde{x})\, \varphi(\tilde{x}; \mu, \sigma) \, \mathrm{d}\tilde{x}$ and $\mathcal{C}_-(\mu, \sigma) = \int_{-\infty}^0 \rho(\tilde{x})\, \varphi(\tilde{x}; \mu, \sigma) \, \mathrm{d}\tilde{x}$ indicate the cumulative recalibrated contributions of positive and negative features, respectively.

***core insight.*** As $\sigma$ increases, the Gaussian distribution $\varphi(\tilde{x}; \mu, \sigma)$ becomes increasingly flat. In the limit $\sigma \to \infty$, the contributions restricted to any finite interval $[a, b]$ vanish:

$$C_+\big|_{\tilde{x} \in [a,b]} \to 0, \quad C_-\big|_{\tilde{x} \in [a,b]} \to 0 \, . \tag{29}$$

Thus, the total contributions $C_+$ and $C_-$ are entirely determined by the behavior over the tails of the distribution.

We focus on two semi-infinite regions: $(M_-, 0)$ with $M_- \ll 0$, and $(0, M_+)$ with $M_+ \gg 0$, where

$$C_- \approx C_-\big|_{\tilde{x} \in (M_-,0)}, \quad C_+ \approx C_+\big|_{\tilde{x} \in (0,M_+)} \, . \tag{30}$$

In this regime, the gating function satisfies:

$$\rho(\tilde{x}) \to 1 \quad \text{for } \tilde{x} \in (0, M_+), \qquad \rho(\tilde{x}) \to 0 \quad \text{for } \tilde{x} \in (M_-, 0) \, . \tag{31}$$

As a result, the ratio of negative to positive cumulative contributions tends to zero:

$$\frac{C_-\big|_{\tilde{x} \in (M_-,0)}}{C_+\big|_{\tilde{x} \in (0,M_+)}} \to 0 \, , \tag{32}$$

which implies:

$$\frac{C_-}{C_+} \to 0 \, , \tag{33}$$

since the excluded portions $[M_-, 0]$ and $[0, M_+]$ are finite intervals whose contribution becomes negligible in the limit $\sigma \to \infty$.

***Proof based on the core insight.*** Fix any $\mu > 0$. To analyze the ratio $C_-/C_+$ as $\sigma \to \infty$, we partition each integral into contributions over finite and infinite intervals.

Let $M_- < 0 < M_+$ be two constants. We split:

$$C_- = \underbrace{\int_{-\infty}^{M_-} \rho(\tilde{x})\varphi(\tilde{x};\mu,\sigma)\,\mathrm{d}\tilde{x}}_{I_-^{\infty}} + \underbrace{\int_{M_-}^{0} \rho(\tilde{x})\varphi(\tilde{x};\mu,\sigma)\,\mathrm{d}\tilde{x}}_{I_-^{\text{finite}}}, \tag{34}$$

$$C_+ = \underbrace{\int_{0}^{M_+} \rho(\tilde{x})\varphi(\tilde{x};\mu,\sigma)\,\mathrm{d}\tilde{x}}_{I_+^{\text{finite}}} + \underbrace{\int_{M_+}^{\infty} \rho(\tilde{x})\varphi(\tilde{x};\mu,\sigma)\,\mathrm{d}\tilde{x}}_{I_+^{\infty}}. \tag{35}$$

*Finite region vanishing.* For any $\varepsilon > 0$, since $\varphi(\tilde{x};\mu,\sigma)$ converges uniformly to 0 on any compact interval as $\sigma \to \infty$, we can find $\Sigma > 0$ such that for all $\sigma > \Sigma$,

$$I_-^{\text{finite}} < \varepsilon, \quad I_+^{\text{finite}} < \varepsilon. \tag{36}$$

*Asymptotic behavior on infinite tails.* Note that on $\tilde{x} \in (M_-, 0)$, $\rho(\tilde{x}) \to 0$, and on $\tilde{x} \in (M_+, \infty)$, $\rho(\tilde{x}) \to 1$. Thus, for large enough $\sigma$,

$$I_-^{\infty} \leqslant \sup_{\tilde{x}\in(-\infty,M_-)} \rho(\tilde{x}) \cdot \int_{-\infty}^{M_-} \varphi(\tilde{x};\mu,\sigma)\,\mathrm{d}\tilde{x} \leqslant \rho(M_-) \cdot \int_{-\infty}^{M_-} \varphi(\tilde{x};\mu,\sigma)\,\mathrm{d}\tilde{x}. \tag{37}$$

Similarly,

$$I_+^{\infty} \geqslant \inf_{\tilde{x}\in(M_+,\infty)} \rho(\tilde{x}) \cdot \int_{M_+}^{\infty} \varphi(\tilde{x};\mu,\sigma)\,\mathrm{d}\tilde{x} \geqslant \rho(M_+) \cdot \int_{M_+}^{\infty} \varphi(\tilde{x};\mu,\sigma)\,\mathrm{d}\tilde{x}. \tag{38}$$

Since $\rho(M_-) \ll 1$ and $\rho(M_+) \approx 1$, and since $\varphi(\tilde{x};\mu,\sigma)$ is normalized, we can always choose $M_-$ and $M_+$ so that:

$$\int_{M_+}^{\infty} \varphi(\tilde{x};\mu,\sigma)\,\mathrm{d}\tilde{x} \gg \int_{-\infty}^{M_-} \varphi(\tilde{x};\mu,\sigma)\,\mathrm{d}\tilde{x} \quad \text{for all large } \sigma. \tag{39}$$

*Conclude the ratio vanishes.* For all large $\sigma$,

$$\frac{C_-(\mu,\sigma)}{C_+(\mu,\sigma)} = \frac{I_-^{\infty} + I_-^{\text{finite}}}{I_+^{\infty} + I_+^{\text{finite}}} \leqslant \frac{\rho(M_-) \cdot \int_{-\infty}^{M_-} \varphi(\tilde{x})\,\mathrm{d}\tilde{x} + \varepsilon}{\rho(M_+) \cdot \int_{M_+}^{\infty} \varphi(\tilde{x})\,\mathrm{d}\tilde{x} - \varepsilon} \to 0. \tag{40}$$

Hence, we have:

**Lemma B.4.** *For the conditions assumed in Proposition B.3, we have:* $\lim_{\sigma\to\infty} \frac{C_-(\mu,\sigma)}{C_+(\mu,\sigma)} = 0$.

Then, by using Lemma B.4, we have the following conclusion:

$$
\begin{aligned}
\lim_{\sigma \to +\infty} \mathcal{R}\left(\mu, \sigma\right) &= \lim_{\sigma \to +\infty} \frac{\mathbb{E}\left(\rho\left(\tilde{x}\right) \mid \tilde{x} < 0\right)}{\mathbb{E}\left(\rho\left(\tilde{x}\right) \mid \tilde{x} > 0\right)} \\
&= \lim_{\sigma \to +\infty} \frac{\frac{1}{\mathbb{P}(\tilde{x}<0)} \int_{-\infty}^{0} \rho(\tilde{x})\, \varphi(\tilde{x}; \mu, \sigma)\, \mathrm{d}\tilde{x}}{\frac{1}{\mathbb{P}(\tilde{x}>0)} \int_{0}^{+\infty} \rho(\tilde{x})\, \varphi(\tilde{x}; \mu, \sigma)\, \mathrm{d}\tilde{x}} \\
&= \lim_{\sigma \to +\infty} \frac{\frac{1}{\Phi\left(\frac{-\mu}{\sigma}\right)} \int_{-\infty}^{0} \rho(\tilde{x})\, \varphi(\tilde{x}; \mu, \sigma)\, \mathrm{d}\tilde{x}}{\frac{1}{1-\Phi\left(\frac{-\mu}{\sigma}\right)} \int_{0}^{+\infty} \rho(\tilde{x})\, \varphi(\tilde{x}; \mu, \sigma)\, \mathrm{d}\tilde{x}} \\
&= \lim_{\sigma \to +\infty} \frac{\frac{1}{\Phi(0)} \int_{-\infty}^{0} \rho(\tilde{x})\, \varphi(\tilde{x}; \mu, \sigma)\, \mathrm{d}\tilde{x}}{\frac{1}{1-\Phi(0)} \int_{0}^{+\infty} \rho(\tilde{x})\, \varphi(\tilde{x}; \mu, \sigma)\, \mathrm{d}\tilde{x}} \\
&= \lim_{\sigma \to +\infty} \frac{2 \int_{-\infty}^{0} \rho(\tilde{x})\, \varphi(\tilde{x}; \mu, \sigma)\, \mathrm{d}\tilde{x}}{2 \int_{0}^{+\infty} \rho(\tilde{x})\, \varphi(\tilde{x}; \mu, \sigma)\, \mathrm{d}\tilde{x}} \\
&= \lim_{\sigma \to +\infty} \frac{\int_{-\infty}^{0} \rho(\tilde{x})\, \varphi(\tilde{x}; \mu, \sigma)\, \mathrm{d}\tilde{x}}{\int_{0}^{+\infty} \rho(\tilde{x})\, \varphi(\tilde{x}; \mu, \sigma)\, \mathrm{d}\tilde{x}} \\
&= 0\,, \tag{41}
\end{aligned}
$$

where

$$
\Phi\left(z\right) = \frac{1}{\sqrt{2\pi}} \int_{-\infty}^{z} \mathrm{e}^{-\frac{t^2}{2}}\, \mathrm{d}t \tag{42}
$$

denotes the standard normal cumulative distribution function (CDF).

Note that the above conclusions are applicable to general sigmoid-like functions, since simultaneously changing $\beta_{ve}$ and $\beta_{ho}$ in both $\mathcal{C}_+\left(\mu, \sigma\right)$ and $\mathcal{C}_-\left(\mu, \sigma\right)$ does not affect the results.

This completes the proof. $\qquad \square$

***A variant case: for uniformly distributed inputs.*** Here, we generalize our Intuition 4 to the case where the inputs $\tilde{x}$ obey a uniform distribution.

**Proposition B.5.** *For $\tilde{x} \sim \mathcal{U}\left(\delta^-, \delta^+\right)$, where $\delta^- < 0$ and $\delta^+ > 0$, we have the following conclusion:* $\lim_{\delta^- \to -\infty, \delta^+ \to +\infty} \frac{\int_{\delta^-}^{0} \rho(\tilde{x})\mathrm{d}\tilde{x}}{\int_{0}^{\delta^+} \rho(\tilde{x})\mathrm{d}\tilde{x}} = 0.$

*Proof.* Because $\lim_{\tilde{x} \to -\infty} \rho\left(\tilde{x}\right)\tilde{x} = 0$, we have: $\lim_{\tilde{x} \to -\infty} \rho\left(\tilde{x}\right) = 0$. Further, since $\lim_{\tilde{x} \to +\infty} \rho\left(\tilde{x}\right) = \mathcal{M} > 0 = \lim_{\tilde{x} \to -\infty} \rho\left(\tilde{x}\right)$, we have two derived conclusions:

(1) $\rho\left(\tilde{x}\right)$ is monotonically non-decreasing and $\rho\left(\tilde{x}\right) \geqslant 0$;

(2) $\exists \Delta^+$ such that $\rho\left(\tilde{x}\right) > 0$ for any $\tilde{x} > \Delta^+$.

Without loss of generality, suppose $\rho\left(\tilde{x}\right) = K \mid_{\tilde{x}>\Delta^+}$, where $K > 0$ is a constant, we have:

$$
\begin{aligned}
\lim_{\delta^+ \to +\infty} \int_{0}^{\delta^+} \rho\left(\tilde{x}\right) \mathrm{d}\tilde{x} &= \int_{0}^{\Delta^+} \rho\left(\tilde{x}\right) \mathrm{d}\tilde{x} + \lim_{\delta^+ \to +\infty} \int_{\Delta^+}^{\delta^+} \rho\left(\tilde{x}\right) \mathrm{d}\tilde{x} \\
&\geqslant \lim_{\delta^+ \to +\infty} \int_{\Delta^+}^{\delta^+} \rho\left(\tilde{x}\right) \mathrm{d}\tilde{x} \\
&> \lim_{\delta^+ \to +\infty} \int_{\Delta^+}^{\delta^+} K \mathrm{d}\tilde{x} \\
&= +\infty\,. \tag{43}
\end{aligned}
$$

Therefore, we have: $\lim_{\delta^+ \to +\infty} \int_{0}^{\delta^+} \rho\left(\tilde{x}\right) \mathrm{d}\tilde{x} = +\infty.$

Then, we prove $\lim_{\delta^- \to -\infty} \int_{\delta^-}^0 \rho(\tilde{x}) \, \mathrm{d}\tilde{x}$ is upper-bounded.

Because $\lim_{\tilde{x} \to -\infty} \rho(\tilde{x}) \tilde{x} = 0$, we have:

$$\lim_{\tilde{x} \to -\infty} \rho(\tilde{x}) \tilde{x} = \lim_{\tilde{x} \to -\infty} \frac{\rho(\tilde{x})}{\frac{1}{\tilde{x}}} = 0 \implies \lim_{\tilde{x} \to +\infty} \frac{\rho(-\tilde{x})}{\left|\frac{1}{\tilde{x}}\right|} = \lim_{\tilde{x} \to +\infty} \frac{\rho(-\tilde{x})}{\frac{1}{\tilde{x}}} = 0 \,. \tag{44}$$

Without loss of generality, let $g(\tilde{x}) = \rho(-\tilde{x})$, we have: $g(\tilde{x}) = o\left(\frac{1}{\tilde{x}}\right)$ as $\tilde{x} \to +\infty$. That is, $\exists \Delta^- > 0$ such that $\forall \tilde{x} > \Delta^-$ we have:

$$g(\tilde{x}) \leqslant C \frac{1}{\tilde{x}^{1+\epsilon}} \,, \tag{45}$$

where $C > 0$ and $\epsilon > 0$ are constants.

Based on the derived conclusions above, we have:

$$\lim_{\delta^+ \to +\infty} \int_0^{\delta^+} g(\tilde{x}) \, \mathrm{d}\tilde{x} = \int_0^{\Delta^-} g(\tilde{x}) \, \mathrm{d}\tilde{x} + \lim_{\delta^+ \to +\infty} \int_{\Delta^-}^{\delta^+} g(\tilde{x}) \, \mathrm{d}\tilde{x}$$
$$= C_1 + \lim_{\delta^+ \to +\infty} \int_{\Delta^-}^{\delta^+} g(\tilde{x}) \, \mathrm{d}\tilde{x} \,, \tag{46}$$

where $C_1 > 0$ is a constant, and:

$$\lim_{\delta^+ \to +\infty} \int_{\Delta^-}^{\delta^+} g(\tilde{x}) \, \mathrm{d}\tilde{x} \leqslant C \lim_{\delta^+ \to +\infty} \int_{\Delta^-}^{\delta^+} \frac{1}{\tilde{x}^{1+\epsilon}} \mathrm{d}\tilde{x} = C \left[ \frac{\tilde{x}^{-\epsilon}}{\epsilon} \right]_{+\infty}^{\Delta^-} = C \cdot C_2 - C \cdot 0 = C \cdot C_2 < +\infty \,, \tag{47}$$

where $C_2 > 0$ is a constant. This proves that $\lim_{\delta^- \to -\infty} \int_{\delta^-}^0 \rho(\tilde{x}) \, \mathrm{d}\tilde{x}$ is upper-bounded.

Therefore, we have the conclusion: $\lim_{\delta^- \to -\infty, \delta^+ \to +\infty} \mathcal{R}_\rho(\delta^-, \delta^+) = 0$

This completes the proof. $\square$

## C  GENERALIZATION BEYOND VISION: GLUE EVALUATION

To further assess the generalizability of FleS beyond vision, we validate it on the **GLUE** benchmark (Wang et al., 2018).

*Adapting vision FleS to NLP.* FleS aims to *extract, encode, and inject* task-relevant commonalities across a semantically meaningful group of inputs ("reference feature group"). In vision, class labels provide a natural grouping; in NLP, although token semantics are highly context-sensitive, task-dependent regularities still emerge (*e.g.*, sentiment). Thus, when adapting FleS to NLP, the critical step is the construction of the *indicator* (class-/group-relevant statistics) under contextual volatility, while the modulation mechanism (e.g., MLP-based scaling) can remain largely unchanged. We adapt the practical FleS from the vision domain to NLP based on these heuristic insights.

*Practical models.* Here, we introduce two NLP FleS variants:

- **FleS-NLP**: direct adaptation of vision FleS to sequences; replace hard positive selection with `Softplus` to stably extract indicators on short sequences. Moreover, because each token already encodes condensed semantics and token meanings can vary substantially within a sentence, we compute a token-level "class" indicator rather than a sentence-level mean.

- **FleS-SeqGate**: lightweight enhancement that mimics state evolution via a depthwise 1D conv inside a simple Sigmoid-based gate-MLP (with a channel reduction ratio of 8 by default). This adds only $\sim$6% parameters/FLOPs over the baseline model with GELU activation. Notably, this *SeqGate* indicator provides sequence-aware, content-adaptive smoothing akin to a lightweight state evolution, while remaining permutation-equivariant to batch ordering.

*Training protocol.*

- **Backbone.** BERT-Tiny.
- **Pretraining.** BookCorpus + Wikipedia from scratch; tokenizer=`bert-base-uncased`, max len=128, batch=4096, lr=$6\times10^{-4}$, epochs=50, warmup=0.1, wd=0.01, mixed precision.
- **Fine-tuning.** GLUE (9 tasks): batch=128, lr=$3\times10^{-4}$; epochs: CoLA/STS-B=100, MRPC=10, others=4.

Table 5: Comparative results on GLUE benchmark (full-task evaluation).

| Backbone | | BERT-Tiny | | | | BERT-Mini | BERT-Small |
|---|---|---|---|---|---|---|---|
| Activation | | GELU | **FleS-NLP** | **FleS-SeqGate** | | GELU | GELU |
| #Params. | | 4.38M | 4.51M | 4.66M | 4.66M | 11.20M | 28.80M |
| #Num-Attn-Blocks | | 2 | 2 | 2 | 2 | 4 | 4 |
| #Pretrained-Ep. | | 50 | 50 | 14 | 50 | 50 | 50 |
| CoLA | MCC | 12.13 | 20.68 | 23.48 | 22.04 | 17.38 | 27.24 |
| SST-2 | Acc. | 81.54 | 82.45 | 83.72 | 84.17 | 83.65 | 89.79 |
| MRPC | F1 | 77.78 | 81.75 | 83.20 | 83.56 | 76.35 | 87.10 |
| | Acc. | 69.61 | 71.32 | 73.77 | 75.98 | 76.68 | 81.13 |
| QQP | F1 | 78.00 | 78.57 | 81.99 | 82.21 | 80.55 | 84.86 |
| | Acc. | 83.21 | 83.25 | 86.50 | 86.48 | 85.80 | 88.73 |
| STS-B | Pearson | 23.65 | 23.72 | 77.03 | 79.68 | 84.25 | 85.20 |
| | Spearman | 23.72 | 22.88 | 76.37 | 79.14 | 84.33 | 85.13 |
| MNLI | -M (Acc.) | 65.58 | 67.25 | 71.77 | 72.44 | 67.47 | 77.42 |
| | -MM (Acc.) | 65.66 | 68.31 | 72.25 | 72.94 | 67.72 | 77.97 |
| QNLI | Acc. | 63.77 | 65.37 | 81.59 | 81.37 | 73.84 | 83.87 |
| RTE | Acc. | 51.26 | 55.60 | 57.04 | 55.96 | 57.04 | 58.84 |
| WNLI* | Acc. | 50.70 | 56.34 | 56.34 | 56.34 | 49.30 | 45.07 |
| | SCORE | 56.72 | **59.44** | **69.97** | **70.18** | **67.59** | **74.32** |

\* BERT-Tiny enhanced with FleS-NLP and FleS-SeqGate are compared against the GELU baseline and substantially larger models, *i.e.*, BERT-Mini and BERT-Small. Note that "WNLI" (marked by "⋆") is excluded from the final score as recommended (Wang et al., 2018).

*Findings.* Our observations are fourfold:

- **FleS-NLP** enjoys clear improvements over the GELU baseline with minimal overhead, aligning with the vision findings.
- **FleS-SeqGate** delivers remarkably powerful gains over both GELU and FleS-NLP with only marginal additional cost.
- **Well-suited for fast pretraining:** FleS-SeqGate achieves near–50-epoch performance after just 14 epochs, suggesting fast convergence and favorable optimization behavior.
- **Challenging significantly larger models:** *FleS-SeqGate enables BERT-Tiny to surpass much larger models, i.e.*, BERT-Mini ($\sim \mathbf{2.5}\times$ in parameters and computational cost) by a notable margin, while remaining competitive with BERT-Small ($> \mathbf{6}\times$ in parameters and computational cost).

These demonstrate the effectiveness, generalizability, and extensibility of our theoretical and heuristic insights, as well as the strong potential of the FleS activation methodology inspired by them.

## D  FURTHER ELABORATION ON THE USE OF POSITIVE AND NEGATIVE PRE-ACTIVATIONS IN FLES INDICATOR MODELING

A central design principle in FleS is to construct channel-wise indicators via a *sign-aware recalibration* that monotonically emphasizes positive features and suppresses negative ones before

summarizing pre-activations. The positive-only indicator used in our default configuration is the simplest instantiation of this idea: it explicitly distinguishes positive and negative responses and focuses the statistics on positively activated features.

We suggest choosing between the positive-only indicator and softer alternatives based on the application scenario. For ImageNet-style vision benchmarks, a positive-only design is typically sufficient: these datasets exhibit relatively mild semantic variation in appearance, and a hard separation between positive and negative responses already provides reliable per-channel statistics at very low computational cost. In contrast, for NLP tasks with more abrupt token-level semantic changes, we adopt a Softplus-based indicator (Sec. C), which offers numerically smoother and more stable behavior. More generally, in applications that demand numerically safer behavior—for example, when negative values carry informative semantic meaning—we recommend using the two additional instantiations described in Sec. E.12, with the Softplus-based variant being generally preferred.

To clarify how we interpret positive and negative pre-activation responses, consider a self-gated activation of the form (revisiting Eq. (1))

$$\phi(\tilde{x}) = \rho(\tilde{x}) \cdot \tilde{x}, \quad \rho(\tilde{x}) \geqslant 0, \tag{48}$$

where $\rho(\tilde{x})$ is viewed as a soft importance score indicating how strongly a response $\tilde{x}$ should be retained. In this perspective, *positive* and *negative* responses are not globally absolute concepts, but are defined relative to a given filter. Suppose a layer includes two filters $\boldsymbol{w}_1, \boldsymbol{w}_2 \in \mathbb{R}^C$ with $C \geqslant 2$, and let $\boldsymbol{x} \in \mathbb{R}^C$ be an input feature vector. The corresponding output $\tilde{\boldsymbol{x}} \in \mathbb{R}^2$ has two channels, where $\tilde{x}_1 = \langle \boldsymbol{x}, \boldsymbol{w}_1 \rangle$ and $\tilde{x}_2 = \langle \boldsymbol{x}, \boldsymbol{w}_2 \rangle$ (omitting bias terms for simplicity). The same $\boldsymbol{x}$ may be strongly suppressed by $\boldsymbol{w}_1$ (e.g., $\tilde{x}_1 < 0$) while still being informative for $\boldsymbol{w}_2$ (e.g., $\tilde{x}_2 > 0$). Being down-weighted in one channel does not imply that a feature is discarded by the layer; it may be emphasized in other channels whose filters are better aligned with it.

Based on this understanding, we propose that an effective principle for constructing channel-wise indicators in FleS is to introduce *differential feature rectification* that mitigates confusion caused by positive–negative cancellation. Concretely, we apply a monotonic, sign-aware recalibration that emphasizes positive features and attenuates negative ones before summarizing pre-activation responses. This makes the resulting statistics more faithfully reflect genuine per-channel intensity, rather than confusing truly weak channels with channels that have strong but mixed-sign responses. In contrast, aggregating positive and negative responses in a symmetric manner may incur "$-1+1$"-style cancellation effects, which increases the burden on the MLP that predicts scaling factors from the indicator, because its input no longer reliably encodes filter-level importance statistics (Tab. 4 (right) empirically supports this observation). Importantly, even if a response is negative for the dominant filter of one channel, the same feature can still be positively emphasized in other channels; FleS only rectifies the statistics used for scaling, not the global availability of features.

From a decision-making perspective, the scenario where negative values carry informative semantic meaning can be viewed as follows: in sufficiently complex applications, before filters have been updated enough to reliably indicate feature importance, the sign of a pre-activation response (positive versus negative) may not reliably reflect utility. Under this mechanistic view, it is often preferable to use a numerically safer indicator (e.g., the Softplus-based variant) rather than aggressively maximizing efficiency via a strictly positive-only design.

# E COMPLEMENTARY ABLATION STUDIES FOR SECTION 5.2

We present further key ablation studies examining the insights underpinning FleS, complementing Sec. 5.2. Unless otherwise specified, each ablation experiment is conducted on ImageNet using the Swin-Min (Liu et al., 2021) or PoolFormer-S12 (Yu et al., 2022) backbones.

## E.1 ON THE SATURATION REGIME

We quantify saturation on ImageNet using the original Swin-Micro and PoolFormer-S12 trained for 120 epochs. For each epoch and for every MLP block, we measure the fraction of activations whose self-gating weight $\rho(\tilde{x})$ exceeds 0.9 (for GELU, $0 < \rho(\tilde{x}) < 1$), and aggregate these fractions per hierarchical stage.

For Swin-Micro, saturation is clearly depth-dependent in early training: in the first few epochs, only about 0.1%–0.2% of activations are saturated in stages 1–2, compared to roughly 0.6% in stage 3 and 2.5%–3% in stage 4. As training progresses, stages 2–4 stabilize around 1%–2%, while stage 1 increases to about 4%–5%, rising from almost zero to a higher steady level. On average, stages 2–4 remain in a moderate range (around 1%–2%), whereas stage 1 eventually accounts for a larger saturated fraction.

PoolFormer-S12 exhibits stronger saturation under the same protocol, especially in deeper stages. In early epochs, stages 3–4 already reach roughly 3% and 6%–7%, respectively. During mid training, saturation becomes more evenly distributed across depth: stage 1 rises to about 4%–5%, while stages 2–4 settle around 2%–3%. After the middle stage, these ratios remain relatively steady. Overall, except for the very first few epochs, a non-trivial fraction of features stays in the saturation regime throughout training.

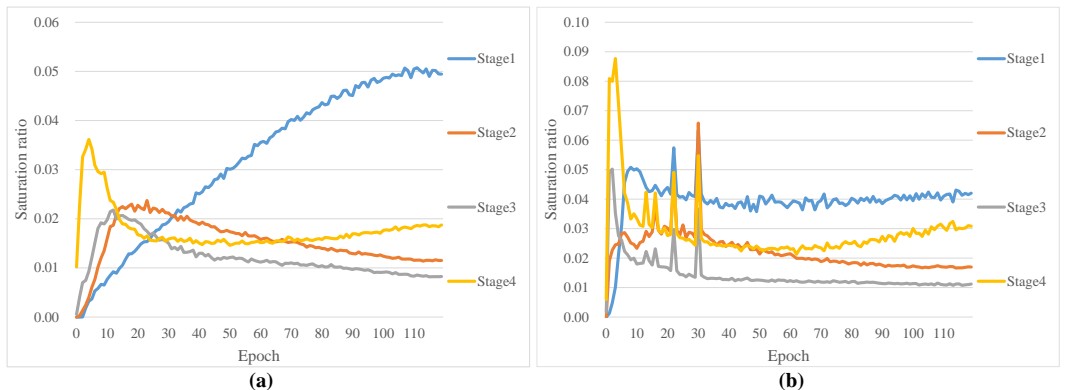

Figure 4: Fraction of activations in the positive-side saturation regime ($\rho(\tilde{x}) > 0.9$) for **(a)** Swin-Micro and **(b)** PoolFormer-S12 on ImageNet. For each epoch and each hierarchical stage, we compute the proportion of MLP activations whose self-gating weight exceeds 0.9 and aggregate per stage. The curves show that (i) saturation is initially stronger in deeper stages, (ii) saturation levels gradually stabilize as training proceeds, and (iii) except for the very first few epochs, a non-trivial fraction of features remains in the saturation regime throughout training.

From the non-local training perspective, these patterns suggest that non-local effects tend to accumulate in deeper blocks at the beginning of training: before the model has acquired stable knowledge, deeper blocks operate at a higher semantic level, so non-local information aggregates there and drives a subset of features into the high-activation (and saturated) region. As training enters the mid–late phase, the model converges to a more effective allocation of activation mass across stages: saturation ratios stabilize, fluctuations shrink, and each stage maintains a characteristic saturation level that reflects its role in the hierarchy. Figure 4 visualizes the saturation dynamics for both architectures.

## E.2 INITIALIZATION OF $\gamma_{ve}$ AND $\gamma_{ho}$

FleS introduces two learnable log-scale parameters, $\gamma_{ve}$ and $\gamma_{ho}$, which are mapped to non-negative scaling factors $\kappa_{ve}$ and $\kappa_{ho}$ through a Softplus transform. In all main experiments, we set $\gamma_{ve} = \gamma_{ho} = 0.6$, which yields an "identity-safe" configuration where $\kappa_{ve}$ and $\kappa_{ho}$ are initialized close to 1.0. This keeps FleS near the baseline activation at initialization and avoids aggressive changes to the optimization landscape.

To quantify sensitivity to this initialization, we perform an ablation on ImageNet using the Swin-Min backbone, jointly varying the initialization of $\gamma_{ve}$ and $\gamma_{ho}$ in the range $[-1.0, 2.0]$. Table 6 summarizes the results. For values around 0–2, FleS is not sensitive to the exact initialization of $\gamma$: all settings consistently yield substantial gains over the GELU baseline (from 68.7% to $\approx 71\%$ top-1), suggesting that AdamW with warmup can absorb moderate changes in the initial gate steepness and range. A more aggressive negative initialization ($\gamma = -1.0$, corresponding to $\kappa \approx 0.31$) leads to a small but noticeable drop, likely because the very small initial scaling reduces effective gradients and mimics an overly small base learning rate in early training; even with AdamW and warmup, the optimizer then struggles to reach the best trajectory. Relatively large positive values slightly increase

the initial steepness but only induce minor, non-systematic fluctuations in accuracy, indicating that the optimizer tolerates moderately over-scaled initial gates. Based on these observations, we adopt $\gamma_{ve} = \gamma_{ho} = 0.6$ as the default setting.

Table 6: Sensitivity of FleS to the initialization of $\gamma_{ve}$ and $\gamma_{ho}$ on ImageNet with the Swin-Min backbone. The parameters $\kappa_{ve}$ and $\kappa_{ho}$ are derived from $\gamma_{ve}$ and $\gamma_{ho}$ via a Softplus transform.

| Activation | Backbone | Init. $\gamma_{ve}, \gamma_{ho}$ | Init. $\kappa_{ve}, \kappa_{ho}$ | Top-1 (%) |
|---|---|---|---|---|
| GELU | Swin-Min | — | — | 68.7 |
| **FleS** | Swin-Min | 0.6 (default) | $\approx 1.0375$ | **71.4** |
| **FleS** | Swin-Min | 0.0 | $\approx 0.6931$ | 71.2 |
| | | $-1.0$ | $\approx 0.3133$ | *70.8* |
| | | 1.0 | $\approx 1.3133$ | **71.4** |
| | | 1.4 | $\approx 1.6204$ | 71.2 |
| | | 1.7 | $\approx 1.8678$ | 71.3 |
| | | 2.0 | $\approx 2.1269$ | 71.1 |

### E.3 EVOLUTION OF $\kappa_{ve}$ AND $\kappa_{ho}$

We study how the FleS scaling factors evolve during training by running a 120-epoch ImageNet experiment with Swin-Min + FleS and, at each epoch, recording for each of the four stages (i) the stage-wise mean of $\kappa_{ho}$ and (ii) the mean of the top 10% $\kappa_{ve}$ values.

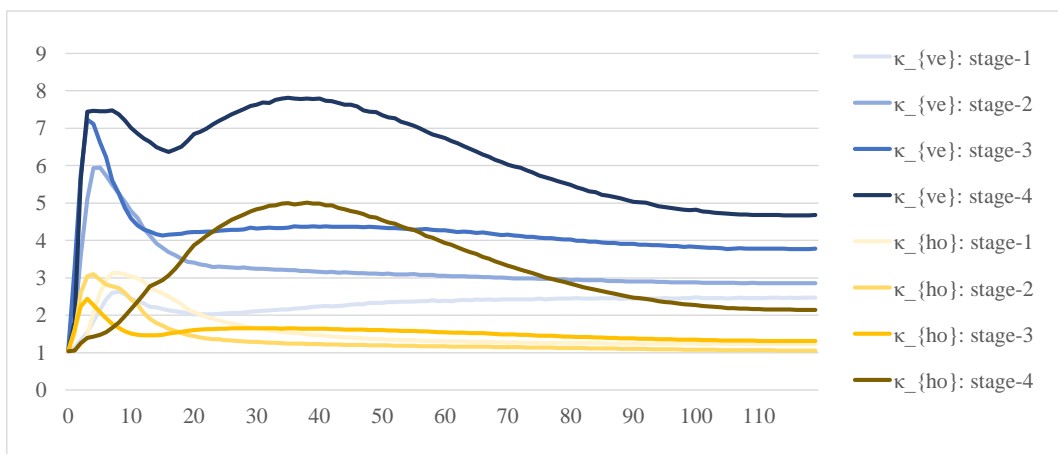

Figure 5: Stage-wise evolution of FleS scaling factors on ImageNet with the Swin-Min backbone. For each epoch and each stage, we record **(1)** the stage-wise mean of the horizontal scaling $\kappa_{ho}$; and **(2)** the mean of the top 10% vertical scaling values $\kappa_{ve}$. Both factors start near 1, rapidly become depth-dependent in early training, and eventually settle into a regime where shallow and mid stages exhibit mild scaling, whereas the deepest stage maintains the strongest horizontal and vertical scaling, reflecting its role in non-local evidence aggregation.

At initialization (epoch 0), both $\kappa_{ho}$ and $\kappa_{ve}$ are close to 1 across all stages, i.e., FleS starts from an approximately identity-like scaling. Within the first few epochs, they quickly become depth-dependent. In stages 2–3, the vertical scaling factor $\kappa_{ve}$ rises from $\approx 1$ to about 2.4–3.1, and the horizontal factor $\kappa_{ho}$ rises to around 5–7. Stage 1 also strengthens but more moderately, with $\kappa_{ve}$ peaking at $\approx 3.1$ and $\kappa_{ho}$ at $\approx 2.6$. Stage 4 shows the most aggressive early horizontal scaling: $\kappa_{ho}$ grows rapidly from $\approx 1.1$ to about 7.4, while $\kappa_{ve}$ increases more gradually over a longer period, eventually peaking around 5.0.

After this early phase, stages 1–3 gradually relax to stable, mid-range values. By the end of training (e.g., epoch 119), stage-1 $\kappa_{ve}$ has decreased from its peak $\approx 3.1$ to $\approx 1.23$, and $\kappa_{ho}$ stabilizes around $\approx 2.47$. Stages 2–3 similarly converge to milder scaling, with $\kappa_{ve}$ around $\approx 1.06$–1.31 and $\kappa_{ho}$ around $\approx 2.85$–3.78. In contrast, the deepest stage 4 consistently maintains the strongest scaling

throughout training: even at epoch 119, it still has the largest values, with $\kappa_{ve} \approx 2.14$ and $\kappa_{ho} \approx 4.68$. The variances exhibit a similar depth-dependent pattern: deeper stages show larger dispersion in the early epochs (especially stage 4), which gradually contracts as training converges.

Overall, this 120-epoch run reveals a clear depth-dependent gating pattern: mid–shallow stages converge to mild scaling, while the last stage maintains the most aggressive horizontal and vertical scaling. This is consistent with our non-local training perspective, where deeper blocks bear more of the non-local evidence aggregation and thus benefit from stronger, persistent gating. Figure 5 visualizes the per-stage $\kappa_{ve}$ and $\kappa_{ho}$ dynamics over training.

### E.4 STABILITY IN IRREGULAR SMALL-BATCH REGIMES

To assess the stability of FleS under non-standard small-batch constructions, we consider an irregular two-stage training schedule on ImageNet with Swin-Min. In the standard setting, we use a batch size of 1024. In the irregular setting, we train for 60 epochs with batch size 132 and learning rate $3 \times 10^{-4}$, followed by 60 epochs with batch size 92 and learning rate $2 \times 10^{-4}$. The results are summarized in Table 7.

Table 7: Performance of FleS and GELU under a standard large-batch regime and an irregular small-batch schedule on ImageNet with the Swin-Min backbone.

| Activation | Backbone | Batch Setting | Top-1 (%)↑ |
|---|---|---|---|
| GELU | Swin-Min | 1024 | 68.7 |
| **FleS** | | | **71.4** |
| GELU | Swin-Min | 132 + 92 | 68.2 |
| **FleS** | | | **70.3** |

Both the GELU baseline and FleS exhibit a small drop in top-1 accuracy under the irregular batch schedule (from 68.7% to 68.2% for GELU, and from 71.4% to 70.3% for FleS), with a comparable degree of degradation. Importantly, FleS still delivers clear improvements over GELU in both regimes, indicating that FleS maintains similar tolerance to irregular/small-batch training while achieving consistently higher accuracy.

### E.5 INTERACTION WITH BATCH NORMALIZATION

We further examine how FleS interacts with batch normalization by replacing layer normalization (LN) with batch normalization (BN) in each Transformer block of Swin-Min and re-training on ImageNet with the same recipe for all variants.

Table 8: Ablation study on interaction between FleS and normalization.

| Activation | Backbone | Norm. | #Params | FLOPs | Top-1 (%)↑ |
|---|---|---|---|---|---|
| GELU | Swin-Min | LN | 11.8M | 1.6G | 68.7 |
| **FleS** | Swin-Min | LN | 13.0M | 1.6G | **71.4** |
| GELU | Swin-Min | BN | 11.8M | 1.6G | *69.3* |
| **FleS** | Swin-Min | BN | 13.0M | 1.6G | **73.8** |

\* Replacing layer normalization (LN) with batch normalization (BN) brings a modest gain for GELU, but a much larger gain when combined with FleS.

As shown in Table 8, BN yields a modest but noticeable improvement for the GELU baseline (top-1: $68.7 \rightarrow 69.3$). When combined with FleS, however, the gain from BN is substantially larger (top-1: $71.4 \rightarrow 73.8$), which is notable given that FleS already operates in a higher-accuracy regime where improvements are typically harder to obtain. This suggests a non-trivial interaction between FleS's

activation scaling and batch-normalized feature statistics. A detailed analysis of this phenomenon is beyond the scope of this work, but it points to an interesting direction for future investigation.

### E.6 LINEAR LAYER VERSUS MLP IN THE INDICATOR HEAD

To assess the effect of replacing MLPs with linear layers, we conduct an ablation on ImageNet with the Swin-Min and Swin-Micro backbones. Specifically, we replace each FleS MLP with a low-rank linear module (i.e., an MLP with reduction ratio but without nonlinearity), denoted as *FleS-LRL*, under a comparable parameter budget.

Table 9: Ablation of the indicator design: MLP vs. a low-rank linear (FleS-LRL) variant.

| Activation | Backbone | Ratio $r$ | #Params | FLOPs (G) | Throughput images/sec. | Top-1↑ (%) |
|---|---|---|---|---|---|---|
| GELU | Swin-Min | — | 11.8M | 1.6 | 4207.2 | 68.7 |
| FleS-LRL | Swin-Min | 32 | 13.0M | 1.6 | 4016.7 | *71.0* |
| **FleS (original)** | | 32 | 13.0M | 1.6 | 4011.3 | **71.4** |
| FleS-LRL | | 24 | 13.4M | 1.6 | 4013.6 | *71.2* |
| **FleS (original)** | | 24 | 13.4M | 1.6 | 4001.0 | **71.5** |
| GELU | Swin-Micro | — | 21.1M | 2.6 | 2775.6 | 78.7 |
| FleS-LRL | Swin-Micro | 32 | 23.5M | 2.6 | 2633.8 | *79.7* |
| **FleS (original)** | | 32 | 23.5M | 2.6 | 2616.8 | **80.3** |

\* FleS-LRL replaces the MLP with two linear layers of the same reduction ratio $r$. FleS outperforms FleS-LRL at close computational cost.

On the Swin-Min backbone, we evaluate two reduction ratios ($r = 32$, our default, and $r = 24$ with slightly more parameters). In both cases, FleS-LRL yields a small but consistent drop in top-1 accuracy compared to the original FleS, while still clearly outperforming the GELU baseline. On the deeper Swin-Micro backbone, the gap between FleS-LRL and FleS becomes more pronounced, in a regime where marginal gains in top-1 are harder to obtain. In terms of cost, FleS-LRL and FleS have almost identical FLOPs and very similar throughput (single RTX 3090, `torch.compile`), and both introduce only a light overhead over the baselines. Given the higher accuracy gains at close cost, we adopt the MLP-with-reduction version as the default configuration (Table 9).

*Intuitive elaboration on using MLPs.* Our decision-making interpretation suggests two key intuitions that guide the practical design of FleS indicators. First, extracting and translating importance descriptors is central: a main challenge in mitigating non-local tension is to construct descriptors that numerically reflect the inter-channel importance of pre-activations and then translate these descriptors into activation scales. In other words, the interpretive view reduces the problem to how to build and exploit channel-wise statistics that genuinely encode feature importance for scaling. Second, sign-aware recalibration matters: in the affine–activation pipeline, the affine projection provides an initial signal of how strongly each input feature should influence the update direction of a given filter, but this signal alone is insufficient. When features with positive and negative affine projections induce gradients of comparable magnitude but opposite effect, their contributions can partially cancel out, making it harder for the filter to follow an update trajectory dominated by informative features. This motivates a sign-aware recalibration step before summarizing channel statistics for scaling.

To probe these intuitions, we first constructed a prototype variant, FleS-Proto. On ImageNet, when very clean, class-aligned channel-importance statistics are available at test time (Tab. 1 in Sec. 4.1), simple linear mappings from these statistics to activation scales already yield very strong gains. However, when the batch is fully shuffled or the batch size is reduced, these gains shrink markedly or even disappear. This suggests that in realistic regimes—where clean and sufficiently rich channel statistics are rarely accessible—*refining* and *translating* noisy channel-importance cues becomes the central practical challenge. We therefore hypothesize that the mapping from channel-wise rectified statistics to effective activation scales is generally more complex than a single affine transform can capture. In FleS, the small MLP head acts as a lightweight universal approximator that refines these rectified statistics within its receptive field and transforms them into scales that better adapt to the

current activation process. Empirically, this design yields a consistent accuracy benefit over the low-rank linear variant FleS-LRL at essentially the same computational cost (Tab. 9), supporting the use of an MLP-based indicator in the final algorithm.

### E.7 ABLATION ON NEIGHBORHOOD SIZE FOR OBJECT DETECTION

In the vision instantiation of FleS for dense recognition tasks, a spatial neighborhood (window) size needs to be chosen to compute the indicator statistics. In practice, we select this window size empirically based on the input resolution and the typical pixel extent of objects. For example, on COCO we adopt a $9 \times 15$ window: COCO images are typically resized to $800 \times 1333$, and small objects usually occupy more than $10^2$ pixels, so a $9 \times 15$ window can capture relatively clean class-relevant statistics. We then use an MLP to aggregate cross-window cues by scanning over the image to construct the indicators.

Table 10: Ablation study of neighborhood size (realized by window) on COCO object detection using PoolFormer-S12 encoder with RetinaNet.

| Activation | Window | $mAP$ (%)↑ | $AP_{50}$(%)↑ | $AP_{75}$(%)↑ | $AP_S$(%)↑ | $AP_M$(%)↑ | $AP_L$(%)↑ |
|---|---|---|---|---|---|---|---|
| GELU | — | 35.5 | 55.5 | 37.5 | 19.5 | 38.7 | 46.3 |
| **FleS** | $9 \times 15$ | **36.2** | 57.0 | 38.1 | 20.7 | 40.1 | 46.8 |
| **FleS** | $9 \times 9$ | 36.1 | 56.6 | 37.9 | 20.5 | 40.0 | 46.8 |
| **FleS** | $5 \times 5$ | 35.8 | 57.0 | 37.5 | 20.0 | 39.6 | 46.7 |

\* FleS remains robust when reducing the window from $9 \times 15$ to $9 \times 9$, while a smaller $5 \times 5$ window leads to a modest performance drop but still improves over the GELU baseline.

To investigate appropriate neighborhood settings, we vary the window size in COCO object detection with PoolFormer-S12 as the encoder and RetinaNet as the detection head. As summarized in Tab. 10, FleS is reasonably robust when reducing the window from $9 \times 15$ to $9 \times 9$ (mAP $36.2 \to 36.1$), while a smaller $5 \times 5$ window leads to a modest drop, yet still outperforms the GELU baseline. These results suggest that using a window covering roughly 50 or more pixels is preferable, while still focusing on a compact local range to facilitate the extraction of relatively simple class-relevant cues.

### E.8 SCOPE OF NON-LOCAL TENSION ACROSS ARCHITECTURES

On classical CNNs (e.g., ResNet), we observe that the performance gains of FleS are mainly driven by the *general benefits of its adaptive scaling mechanism*, rather than by a non-local tension (NLT) effect of the same strength as in token-mixer–based architectures (e.g., Swin and PoolFormer). In relatively deep blocks, a classical CNN can also capture non-local information to some extent through its enlarged effective receptive field, but the resulting non-local signals are more diffuse and less explicit than those in token-mixer–based blocks, so the opportunities for strongly triggering NLT are comparatively rarer. This picture, where FleS primarily boosts CNN blocks via general adaptive scaling rather than a fully developed NLT regime, is consistent with our empirical observations.

As shown in Tab. 11, on ResNet-50 (as a representative classical CNN), two state-of-the-art activation functions, IIEU (Top-1: 79.7%; parameters: 25.6M; FLOPs: 4.2G) and AdaShift (Top-1: 79.9%; parameters: 25.6M; FLOPs: 4.1G), achieve slightly lower accuracy than FleS (Top-1: 80.1%; parameters: 28.1M; FLOPs: 4.1G), while using fewer parameters and similar FLOPs. In contrast, on Swin and PoolFormer, FleS exhibits a substantially larger margin over all competitive activation methods. This contrast suggests that, for classical CNNs, the main benefit of FleS stems from its adaptive scaling mechanism, whereas the NLT phenomenon is more pronounced in architectures that explicitly model non-local/token-mixing interactions.

Notably, we view the channel-wise indicator design—which models interpretable, lightweight non-local cues—as a critical factor underlying the gains of FleS on both CNNs and Transformers. To probe this, we conduct an additional ablation on ResNet-50 where we remove the key component implied by our decision-making interpretation, namely the *monotonic sign-aware recalibration that emphasizes positive and suppresses negative pre-activation responses*. Without this component,

Table 11: Investigation of non-local tension in ResNet.

| Activation | Backbone | #Params | FLOPs | Top-1 (%)↑ |
|---|---|---|---|---|
| ReLU | ResNet-50 | 25.6M | 4.1G | 77.2 |
| IIEU | ResNet-50 | 25.6M | 4.2G | 79.7 |
| AdaS | | 25.6M | 4.1G | 79.9 |
| **FleS** | ResNet-50 | 28.1M | 4.1G | **80.1** |
| **FleS-P&N** | | 28.1M | 4.1G | *79.4* |

\* "FleS-P&N" denotes a FleS variant that aggregates positive and negative pre-activations in a balanced (symmetric) manner.

the Top-1 accuracy of FleS on ResNet-50 drops from 80.1% to 79.4%, making it inferior to IIEU and AdaShift. This supports the view that, on classical CNNs without explicit block-wise non-local modeling, the gains mainly arise from FleS's principled adaptive scaling mechanism, with the decision-making–guided, sign-aware indicator modeling serving as a key driver of the observed improvements.

### E.9 ON POSITIVE CONSTRAINTS FOR SCALING COEFFICIENTS

We posit that positive constraints are decisive for FleS-style adaptive scaling in self-gated activation, based on our analysis in Sec. 3, as each pre-activation $\tilde{x}$ is a sign-sensitive importance measure of a feature. That is, negative values of $\kappa_{ve}$ and $\kappa_{ho}$ will reverse $\rho(\tilde{x})$ and $\tilde{x}$, thereby compromising their physical meaning in decision-making.

Table 12: Ablation study on positive constraints for vertical and horizontal scaling coefficients. "BSL" denotes "Baseline."

| Activation | Backbone | Variant | #Params. | FLOPs | Top-1(%)↑ |
|---|---|---|---|---|---|
| GELU (BSL) | | — | 11.8M | 1.6G | 68.7 |
| FleS Variant | Swin-Min | $-\mathrm{pc}_{\kappa_{ve}}$ | 13.0M | 1.6G | 68.1 |
| | | $-\mathrm{pc}_{\kappa_{ho}}$ | 13.0M | 1.6G | 71.1 |
| | | $-\mathrm{pc}_{\{\kappa_{ve},\kappa_{ho}\}}$ | 13.0M | 1.6G | 67.9 |
| **FleS** | | Original | 13.0M | 1.6G | **71.4** |

We validate this heuristic intuition by comparing the original FleS with ablated FleS variants that omit the positive constraints on:

(1) the vertical scaling coefficient $\kappa_{ve}$ (denoted as "$-\mathrm{pc}_{\kappa_{ve}}$"),

(2) the horizontal scaling coefficient $\kappa_{ho}$ (denoted as "$-\mathrm{pc}_{\kappa_{ho}}$"),

and (3) both $\kappa_{ve}$ and $\kappa_{ho}$ (denoted as "$-\mathrm{pc}_{\{\kappa_{ve},\kappa_{ho}\}}$"). Based on the comparative results in Tab. 12, we observe the following: (1) Omitting the positive constraints on $\kappa_{ve}$ results in a significant drop in accuracy. (2) Omitting the positive constraints on $\kappa_{ho}$ leads to a clear decrease in accuracy, but not as severe as (1), because the base function (*i.e.*, Sigmoid) itself provides a positive constraint on the overall scaled inputs. (3) Omitting the positive constraints on both $\kappa_{ve}$ and $\kappa_{ho}$ results in the worst performance. These phenomena support our intuition.

### E.10 ON INDEPENDENT USE OF THE VERTICAL/HORIZONTAL SCALING COEFFICIENT

We model $\kappa_{ve}$ and $\kappa_{ho}$ as scaling coefficients for activation functions, responsible for controlling the bounds and steepness, respectively, thus enabling targeted modulation of different aspects of the activation shape.

We compare the original FleS with two ablated variants, omitting (1) the vertical scaling coefficient $\kappa_{ve}$ (denoted as "$-\kappa_{ve}$") and (2) the horizontal scaling coefficient $\kappa_{ho}$ (denoted as "$-\kappa_{ho}$"), to

Table 13: Ablation study on the independent use of vertical or horizontal scaling.

| Activation | Backbone | Variant | #Params. | FLOPs | Top-1(%)↑ |
|---|---|---|---|---|---|
| GELU (BSL) | | — | 11.8M | 1.6G | 68.7 |
| FleS Variant | Swin-Min | $-\kappa_{ve}$ | 12.6M | 1.6G | 69.6 |
| | | $-\kappa_{ho}$ | 12.6M | 1.6G | 70.8 |
| | | $-\{\kappa_{ve}, \kappa_{ho}\}$ | 12.6M | 1.6G | 68.9 |
| **FleS** | | Original | 13.0M | 1.6G | **71.4** |

\* Note that the FleS variant "$-\{\kappa_{ve}, \kappa_{ho}\}$" is equivalent to SiLU (Elfwing et al., 2018) activation function. "BSL" denotes "Baseline."

examine their individual contributions. As shown in Tab. 13, our key observations are as follows: (1) Both control groups, "$-\kappa_{ve}$" and "$-\kappa_{ho}$", lead to a decrease in accuracy, with "$-\kappa_{ve}$" demonstrating a more severe decrease than "$-\kappa_{ho}$". (2) Despite the performance degradation, both "$-\kappa_{ve}$" and "$-\kappa_{ho}$" still enjoy significant improvements over the GELU baseline. These observations support our insights.

### E.11 GENERALIZABILITY ACROSS VARIOUS WEIGHTING FUNCTIONS

In this work, we apply the Sigmoid function as the default weighting function $\rho$. Since our assumptions generalize across different forms of self-gated functions, we posit that the FleS-style adaptive scaling scheme is applicable to various choices of $\rho$.

Table 14: Ablation study on the generalizability of FleS scaling scheme using different $\rho$.

| Activation | Backbone | weighting $\rho(\tilde{x})$ | Prototype $\phi(\tilde{x})$ | | Top-1(%)↑ |
|---|---|---|---|---|---|
| | | | $\rho(\tilde{x})\tilde{x}$ | $\kappa_{ve}\rho(\kappa_{ho}\tilde{x})\tilde{x}$ | |
| SiLU (Elfwing et al., 2018) | Swin-Min | sigmoid $(\tilde{x})$ | ✓ | | 68.9 |
| **FleS (Original)** | | | | ✓ | **71.4** |
| GELU (Hendrycks et al., 2016) | Swin-Min | $0.5(1 + \mathrm{erf}(\tilde{x}/\sqrt{2}))$ | ✓ | | 68.7 |
| **GELU-FleS** | | | | ✓ | **71.2** |
| Mish (Misra, 2020) | Swin-Min | $\tanh(\mathrm{softplus}(\tilde{x}))$ | ✓ | | 68.6 |
| **Mish-FleS** | | | | ✓ | **70.9** |
| TanhGate (Cai, 2024a) | Swin-Min | $0.5(\tanh(\tilde{x}) + 1)$ | ✓ | | 68.7 |
| **TanhGate-FleS** | | | | ✓ | **71.1** |

\* Each activation function with the suffix "-FleS" refers to a FleS-augmented variant, where the corresponding $\rho$ is applied with FleS-style scaling.

To investigate the generalizability of FleS across various weighting functions, we conduct a tailored ablation study. Specifically, we validate four FleS variants, each of which employs a different $\rho$ function. Each FleS variant is compared with its baseline counterpart (the reference group), which uses the same $\rho$ but excludes the FleS-style scaling augmentation. The references and their corresponding FleS-augmented variants include: (1) baseline SiLU (Elfwing et al., 2018) and FleS, using Sigmoid $\rho$; (2) baseline GELU (Hendrycks & Gimpel, 2016) and GELU-FleS, using ERF-based $\rho$; (3) baseline Mish (Misra, 2020) and Mish-FleS with $\rho(\cdot) = \tanh(\mathrm{softplus}(\cdot))$; and (4) TanhGate (a simple Tanh-based function suggested in (Cai, 2024a)) and TanhGate-FleS, where $\rho(\cdot) = 0.5(\tanh(\cdot) + 1) = 1/1+e^{-2(\cdot)}$.

As reported in Tab. 14, each FleS variant demonstrates significant improvements over its baseline counterpart. This validates the generalizability of FleS's methodology.

### E.12 On polarity-selective indicator: default & alternatives

In Sec. 4.1 (see Eq. (5) in the main paper and the corresponding clarification), we introduce our default method for implementing the discriminative use of positive and negative features in modeling the non-local indicator that drives FleS-style scaling scheme. Notably, Eq. (5) (main paper) is motivated by Intuition 4.2, and alternative technical formulations can be used in its place.

Table 15: Ablation study on technical choices for implementing selective use of positive and negative features in FleS. "BSL" denotes "Baseline."

| Activation | Backbone | Variant | #Params. | FLOPs | Top-1(%)↑ |
|---|---|---|---|---|---|
| GELU (BSL) | | — | 11.8M | 1.6G | 68.7 |
| FleS Variant | Swin-Min | Softplus-based | 13.0M | 1.6G | **71.4** |
| | | Quasi-linear | 13.0M | 1.6G | 71.3 |
| **FleS** | | Original | 13.0M | 1.6G | **71.4** |

To investigate the modeling of non-local indicator (denoted as $\bar{x}_c^+$), we conduct a targeted ablation study comparing our default method with two functionally similar variants, used as reference groups:

(1) The Softplus-based variant:

$$\bar{x}_c^+ = \mathrm{softplus}\left(\gamma_c \tilde{x} + \beta_c\right)\big|_{\tilde{x} \in \mathbb{X}_c}, \tag{49}$$

where $\gamma_c$ and $\beta_c$ are learnable factors to dynamically adjust the shape of Softplus function;

(2) The quasi-linear variant:

$$\bar{x}_c^+ = \begin{cases} \tilde{x}, & \text{if } \tilde{x} \geqslant \eta, \\ \eta \cdot \exp\left(\dfrac{\tilde{x}}{\eta} - 1\right), & \text{if } \tilde{x} < \eta, \end{cases} \tag{50}$$

where $\eta$ is a small threshold (*e.g.*, 0.1 in this ablation study). Note that Eq. (50) defines a differentiable function on $\mathbb{R}$.

As reported in Tab. 15, these three variants achieve similar improvements over the baseline, while the default method attains these improvements with relatively minimal technical effort.

### E.13 From FleS-Proto to practical FleS: what batch-class coherence reveals

**Setup.** Here we present an extended ablation study that helps clarify our findings on FleS-Proto and justifies the FleS design they inspire.

To this end, we use a *Category-Block Shuffled Evaluation Protocol* on ImageNet:

(1) group validation images by ground-truth labels (50 per class);

(2) form each class group into a contiguous block;

(3) randomly permute the order of class blocks;

(4) uniformly shuffle image order *within* each block. Sequential sampling thus yields evaluation batches that typically contain same-class samples, while the global class order is disrupted.

We denote this as "#Class-Block-Shuffle" and vary the evaluation batch size. We also consider "#Total Batch Shuffle," which mixes classes across all batches.

**Observations.** (1) When batches remain class-coherent (CBS) and the evaluation batch size is moderate ($\geqslant 8$), **FleS-Proto** shows stable, strong performance. (2) As the batch size shrinks (e.g., $\leqslant 4$), **FleS-Proto** degrades, indicating under-representative per-batch statistics. (3) **FleS** and **GELU** are insensitive to both class-order and global shuffling, as they do not rely on inter-sample batch statistics. These phenomena support the view that the drop with FleS-Proto arises from insufficiently representative statistics in small or mixed batches.

Table 16: Effect of evaluation batch size (BS) under the Class-Block Shuffle protocol (ImageNet).

| Activation | Backbone | #CB-Shuffle | #TB-Shuffle | #Eval. BS | #Params. | FLOPs | Top-1(%)↑ |
|---|---|---|---|---|---|---|---|
| FleS-Proto | Swin-Micro | — | — | 256 | 21.1M | 2.6G | **85.2** |
| | | ✓ | — | 256 | 21.1M | 2.6G | **85.2** |
| | | ✓ | — | 128 | 21.1M | 2.6G | **85.4** |
| | | ✓ | — | 64 | 21.1M | 2.6G | **85.5** |
| | | ✓ | — | 32 | 21.1M | 2.6G | 84.9 |
| | | ✓ | — | 16 | 21.1M | 2.6G | 83.1 |
| | | ✓ | — | 8 | 21.1M | 2.6G | 80.1 |
| | | ✓ | — | 4 | 21.1M | 2.6G | *74.9* |
| | | ✓ | — | 2 | 21.1M | 2.6G | *66.7* |
| | | ✓ | — | 1 | 21.1M | 2.6G | *54.1* |
| | | — | ✓ | 256 | 21.1M | 2.6G | *77.3* |
| GELU (BSL) | Swin-Micro | — | — | 256 | 21.1M | 2.6G | 78.7 |
| | | — | ✓ | 256 | 21.1M | 2.6G | 78.7 |
| | | ✓ | — | 1 | 21.1M | 2.6G | 78.7 |
| **FleS** | Swin-Micro | — | — | 256 | 23.5M | 2.6G | **80.3** |
| | | — | ✓ | 256 | 23.5M | 2.6G | **80.3** |
| | | ✓ | — | 1 | 23.5M | 2.6G | **80.3** |

\* "#Class-Block (CB)-Shuffle" indicates whether the block protocol is applied; "#Total-Batch (TB)-Shuffle" shuffles samples globally across batches.

**From prototype to practice.** **FleS-Proto** is a proof of concept serving to expose modeling cues on how to extract and aggregate class-specific statistics. Guided by these cues, **FleS** replaces hand-crafted batch statistics with *learned* transformations that infer robust class-relevant indicators even under shuffled or noisy batches.

**On the role of MLPs in FleS (capturing shared regularities through translation equivariance).**
*Core principle.* Effective *adaptive activation scaling*—designed to mitigate *non-local tension* that hampers neural selectivity—should *extract, encode, and inject* common statistical characteristics among samples under a task-relevant grouping rule (cf. the "reference feature group"). Such shared characteristics obey statistical regularities: the more samples, the more representative the statistics; in image classification, the class identity naturally yields a meaningful grouping.

*Why MLPs? Generalizing to shuffled batches.* In fully or heavily shuffled batches, *clean* class-specific statistics are unavailable; when the number of same-class samples drops to $\leqslant 4$, direct aggregation becomes under-representative due to large intra-class variance in $C$-dimensional descriptors. In the extreme case of full shuffling, we cannot infer any grouping beyond a single image.

*Hypothesis.* Although descriptors from the same class may differ in the original space, their *channel-wise distributions* may follow a latent pattern learnable via transformation.

*Goal.* Learn a transformation that maps noisy, sample-level descriptors to meaningful scaling coefficients, recovering shared structure even when class-coherent batches are absent.

*Properties needed.*

(i) **Universal approximability**: MLPs can approximate continuous functions, making them suitable for mapping noisy statistics to scaling coefficients with compact parameterization.

(ii) **Permutation equivariance across tokens and batches**: MLPs apply the same pointwise transformation to every token, independent of position. As a result, moving a token to a different position yields the same value for that token; changing the token order merely reorders the outputs one-to-one. This position-agnostic, tokenwise permutation equivariance makes the layer robust to token and batch ordering.

*Instantiation.* We (a) extract preliminary descriptors (e.g., token/patch-level mean responses per sample); (b) feed them to an MLP that projects them into a shared space where class-commonalities

become salient; (c) use the outputs to guide *adaptive activation scaling*, restoring discriminative signals lost under batch shuffling.

**Beyond shuffled classification: object detection.** The design extends to detection, where an image contains multiple object classes and background. Semantic objects are typically *spatially continuous*; for a non-boundary pixel $I_x$, a small neighborhood $B(I_x, \epsilon)$ contains same-class pixels. One can first compute fine-grained local statistics (*e.g.*, pooled or masked aggregation) and then apply the same MLP transformation to drive adaptive scaling in mixed-class settings.

**Takeaway.** Batch-class coherence diagnostics reveal *why* FleS-Proto succeeds under class-coherent batches yet drops under heavy shuffling, and *how* these phenomena motivate FleS: learn transformations that infer robust, class-relevant indicators without relying on inter-sample batch statistics.

### E.14    ON ROBUSTNESS TO BATCH SIZE

**Setup.** We conduct a controlled ablation on ImageNet with the Swin-Micro backbone, varying the global batch size $B \in \{256, 1024, 2048\}$ for FleS while keeping all other training settings fixed; the learning rate is scaled approximately linearly with $B$. A GELU baseline at $B=1024$ is included for reference.

Table 17: Ablation of training batch size on ImageNet.

| Activation | Backbone | #Params. | FLOPs | Batch Size | LR | Top-1(%)↑ |
|---|---|---|---|---|---|---|
| GELU | Swin-Micro | 21.1M | 2.6G | 1024 | $1 \times 10^{-3}$ | 78.7 |
| **FleS** | Swin-Micro | 23.5M | 2.6G | 256 | $2.5 \times 10^{-4}$ | 80.1 |
| | | | | 1024 | $1 \times 10^{-3}$ | **80.3** |
| | | | | 2048 | $2 \times 10^{-3}$ | **80.3** |

\* With LR scaled approximately linearly with batch size by following (Goyal et al., 2017). FleS maintains stable Top-1 across a wide range of batch sizes.

**Results.** As shown in Tab. 17, **FleS** exhibits strong stability across an $8\times$ range of batch sizes (80.1–80.3% Top-1). At *test time*, FleS is batch-size invariant; evaluating with batch size $= 1$ yields 80.1% Top-1, consistent with the design that avoids inter-sample batch statistics at inference.

### E.15    CHANNEL REDUCTION RATIO OF LIGHTWEIGHT MLP IN FLES

**Setup.** We analyze the sensitivity of **FleS** to the *channel reduction ratio* $r$, which directly controls the hidden size of the two lightweight MLP heads: $H = \lfloor D/r \rfloor$ for input/output scaling (all other settings fixed). Unless stated otherwise, the backbone is **Swin-Micro** on **ImageNet**; we set $r = 32$ by default.

Table 18: Ablation on the channel reduction ratio $r$.

| Activation | Backbone | #Params. | Ratio $r$ | Top-1(%)↑ |
|---|---|---|---|---|
| GELU | Swin-Micro | 21.1M | — | 78.7 |
| **FleS** | Swin-Micro | 22.8M | 48 | 79.7 |
| | | 23.5M | 32 | **80.3** |
| | | 25.9M | 16 | **80.4** |

\* Larger MLPs (smaller $r$) improve accuracy but with diminishing returns relative to parameter growth.

**Findings.** As $r$ decreases (*i.e.*, the MLP size increases), accuracy improves, but the *marginal gain* becomes small compared with the considerable additional parameters. Overly small MLPs ($r=48$) lead to noticeable performance degradation. Overall, $r=32$ strikes a robust balance between accuracy and model size; we therefore adopt it as the default.

### E.16 ROBUSTNESS TO INITIALIZATION

**Setup.** We assess initialization sensitivity on PoolFormer-S12, training with identical settings while varying the random seed $\{42, 0, 31415, 2025\}$. A GELU baseline (seed = 42) is included for reference.

Table 19: Robustness to random initialization.

| Activation | Backbone | #Seed | Top-1(%)↑ |
|---|---|---|---|
| GELU | PoolFormer-S12 | 42 | 77.2 |
| **FleS** | PoolFormer-S12 | 42 | 79.4 |
| | | 0 | 79.4 |
| | | 31415 | 79.3 |
| | | 2025 | **79.5** |

\* FleS demonstrates robustness across seeds and consistently outperforms GELU.

**Findings.** Across seeds, FleS remains stable (79.3–79.5% Top-1 acc.) and consistently exceeds the GELU baseline by a significant margin, indicating robustness to initialization.

## F ROBUSTNESS UNDER LONG-TAILED DISTRIBUTIONS

**Setup.** We evaluate FleS on ImageNet-LT (long-tailed distribution) using PoolFormer-S12. To isolate the intrinsic effect of activation strategies, we *do not* apply any specialized long-tail techniques (*e.g.*, re-weighting/re-sampling, deferred re-balancing). Training follows common long-tail practice: 300 epochs, 20 warm-up epochs, cosine LR, base LR $5 \times 10^{-4}$, AdamW, weight decay 0.05.

Table 20: Comparative evaluation on ImageNet-LT.

| Activation | Backbone | #params. | FLOPs | Top-1(%)↑ |
|---|---|---|---|---|
| GELU | PoolFormer-S12 | 11.9M | 1.8G | 37.1 |
| **FleS** | PoolFormer-S12 | 13.8M | 1.8G | **40.6** |

\* No specialized long-tail mitigating techniques applied.

**Findings.** **FleS** substantially outperforms GELU under class-imbalance, indicating stronger adaptability to long-tailed distributions even without any imbalance-specific heuristics.

## G IMPLEMENTATION RECIPES FOR IMAGENET EXPERIMENTS

For fair comparisons, (1) we adopt the standard training-evaluation recipe (Touvron et al., 2021; Liu et al., 2021) for Vision Transformers, except for (2) Swin-Min, where we reduce the 300-epoch training to 120 epochs (due to time and resource constraints); (3) For ResNets, we adopt the standard CNN training-evaluation recipe (Zhou et al., 2021; Ma et al., 2021). The implementation protocols are detailed as follows:

1. For Swin-Micro (Liu et al., 2021), Swin-T (Liu et al., 2021), and PoolFormer-S12 (Yu et al., 2022), we adopt the standard data augmentation suggested in (Touvron et al., 2021; Liu et al., 2021) and widely used AdamW optimizer (Loshchilov & Hutter, 2019) to train each implemented model with the standard cosine scheduler through 300 epochs (including 20 linear warm-up epochs). The learning rate starts from $1 \times 10^{-3}$ with an effective batch size of 1024 by default and decays to $1 \times 10^{-6}$, smoothly. The weight decay is set to 0.05 and label-smoothing of 0.1. We follow the common practice to stabilize the model weights by 10 cool-down epochs with the minimum learning rate $1 \times 10^{-6}$ after the main epochs.

2. For Swin-Min, we retain most of the recipes from the above configuration 1, except that we reduce the training epochs from 300 to 120 to shorten the training duration, as we train our model and the competing models from scratch.

3. For ResNets (He et al., 2016), we adopt the common data augmentation strategy (Zhou et al., 2021) and the standard SGD optimizer to train each model for 120 epochs, including 5 linearly increasing warm-up epochs. The learning rate starts at 0.1 with a batch size of 256 and decays to $1 \times 10^{-5}$. The momentum and weight decay are set to 0.9 and $1 \times 10^{-4}$, respectively. We follow the common practice to stabilize the model weights by 10 cool-down epochs with the minimum learning rate $1 \times 10^{-5}$ after the main epochs.

Following the common practice, we (1) train and test all models with an image size of $224 \times 224$; (2) report the results of our models and the official results for the baseline methods in terms of Top-1 Accuracy, rounded to one decimal place.

## H  CONVERGENCE ATTRIBUTE

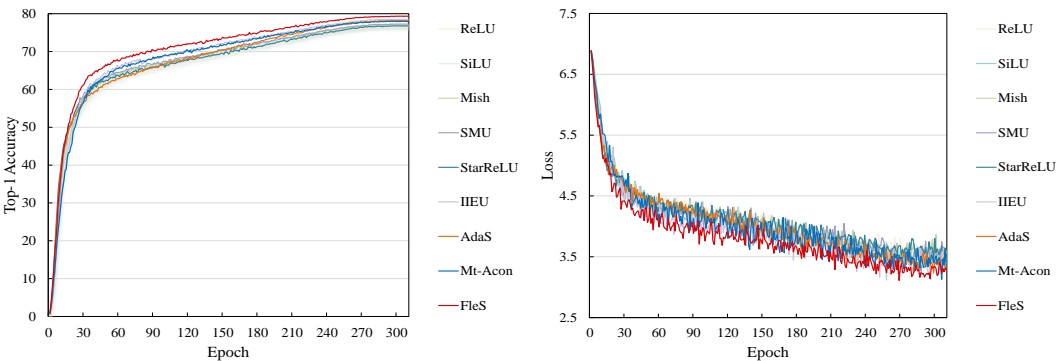

Figure 6: The accuracy curve (left) and loss curve (right) of PoolFormer-S12 (Yu et al., 2022) backbone with different activation models.

We show the convergence curves of models built on the PoolFormer-S12 backbone (Yu et al., 2022) and equipped with FleS and other competing baseline/popular/SOTA activation functions. Note that each model is trained by the standard 300-epoch recipe (suggested in (Touvron et al., 2021; Liu et al., 2021) and introduced in the training configuration 1) from scratch to convergence.

Fig. 6 illustrates the convergence trends in Top-1 accuracy (higher is better) and training loss (lower is better) of the PoolFormer-S12(s) models equipped with FleS and other activation functions. The baseline PoolFormer-S12 employs the GELU activation function. FleS achieves the highest Top-1 accuracy and the lowest loss across epochs, demonstrating its favorable convergence properties.

## I  CIFAR-100 CLASSIFICATION

**Implementation details.** We further compare our FleS with other SOTA/popular activation functions on CIFAR-100 (Krizhevsky, 2009), using CIFAR-Swin-T as the backbone. CIFAR-Swin-T is a modified version of the original Swin-T (Liu et al., 2021), which was designed for ImageNet (Deng et al., 2009) and downstream tasks. Specifically, we reduce the base embedding dimension from 96 to 24 to prevent redundant parameters, as CIFAR-100 contains far fewer images than ImageNet and has significantly lower image resolution.

To ensure fair comparisons, all models are trained from scratch using the same standard training-evaluation recipe. To construct this recipe, we adopt most of the training protocols and data augmentations suggested in (Li et al., 2019), with slight modifications to fit the Transformer-based backbone. Specifically, each model is trained for 350 epochs with a batch size of 256, by an AdamW optimizer with a weight decay of 0.05. The learning rate starts from $1 \times 10^{-3}$ and decreases to $1 \times 10^{-6}$ by following the standard cosine learning rate schedule. All the input images are fixed to the size of $32 \times 32$ by following the common practice.

**Experimental results.** Comparative results are shown in Tab. 21, where our FleS improves upon the SOTA and popular competing methods by a significant margin. This observation is consistent with our observations on ImageNet, further supporting the adaptability of FleS across datasets of different scales.

Table 21: Comparison of different activation functions on CIFAR-100 benchmark dataset.

| Backbone | Method | GELU | SiLU | Mish | Pserf | IIEU | AdaS | FleS |
|---|---|---|---|---|---|---|---|---|
| CIFAR-Swin-T | Top-1(%)↑ | 66.7±0.3 | 65.8±0.2 | 65.7±0.3 | 66.0±0.3 | 66.7±0.2 | 67.0±0.2 | **68.9**±0.3 |
| | #Params. | 1.8M | 1.8M | 1.8M | 1.8M | 2.0M | 2.0M | 2.0M |

* Each model is trained 8 times, and the mean and standard deviation of its Top-1 accuracy are reported.

## J  MS COCO OBJECT DETECTION

**Implementation details**. In this Appendix, we further validate the versatility and generalizability of our activation function FleS on MS COCO (Lin et al., 2014) object detection. We evaluate FleS by comparing it with a series of popular and SOTA self-gated activation functions, including (1) GELU (Hendrycks & Gimpel, 2016) (the most widely used activation function in Transformers), (2) SMU (Biswas et al., 2022b), and (3) Meta-ACON (Ma et al., 2021). Meta-ACON generalizes lightweight channel attention (Hu et al., 2020) to perform context-aware dynamic scaling on input features. Thus, it is functionally relevant to FleS and has a similar number of additional parameters, but with a fundamentally different motivation, philosophy, and methodological insights. We conduct the experiment using the popular PoolFormer-S12 (Yu et al., 2022) backbone and RetinaNet detector (Lin et al., 2017).

For fair comparisons, we train each RetinaNet equipped with different activation models from scratch using the same implementation configuration constructed on the $1\times$ schedule in MMDetection toolbox (Chen et al., 2019). To better suit Transformer layers, we replace the default SGD optimizer in the $1\times$ schedule with AdamW optimizer. The learning rate starts at $2 \times 10^{-3}$ and decays to $1 \times 10^{-5}$, gradually. Following common practice, the weight decay is set to 0.05. We report the results using standard evaluation metrics, *i.e.*, mAP as the primary metric for average precision and $AP_{50}$, $AP_{75}$, $AP_S$, $AP_M$, $AP_L$ as specific APs for different scales. Each PoolFormer-S12 (Yu et al., 2022) backbone using a different activation function is initialized with its corresponding ImageNet pre-trained weights. To ensure reproducibility, we maintain deterministic mode in each implementation.

Table 22: Comparative evaluation on MS COCO (Lin et al., 2014) object detection. The PoolFormer-S12 (Yu et al., 2022) is applied as the encoder with the popular RetinaNet detector (Lin et al., 2017).

| Activation | Encoder | $mAP$ (%)↑ | $AP_{50}$(%)↑ | $AP_{75}$(%)↑ | $AP_S$(%)↑ | $AP_M$(%)↑ | $AP_L$(%)↑ |
|---|---|---|---|---|---|---|---|
| GELU | | 35.5 | 55.5 | 37.5 | 19.5 | 38.7 | 46.3 |
| SMU | PoolFormer-S12 | 35.4 | 55.2 | 37.1 | 20.5 | 38.6 | 47.2 |
| Mt-ACON | | 35.5 | 55.7 | 37.6 | 19.7 | 39.1 | **47.6** |
| **FleS (Ours)** | | **36.2** | **57.0** | **38.1** | **20.7** | **40.1** | 46.8 |

**Experimental results**. The comparative results are reported in Tab. 22, where FleS achieves clear improvements over all the competing popular and SOTA activation models across almost all evaluation metrics, especially in $AP_S$, which measures performance on challenging small objects. This further validates the generalizability of FleS.

Note that as discussed in Section 4.2 of our main paper, for dense recognition tasks such as object detection, each image contains multiple semantic classes of objects, which requires computing the channel indicators $\bar{x}^+$ with finer ranges to mitigate class information confusion (*e.g.*, image patches). Due to time and resource constraints, we use a brute-force approach to calculate channel indicators for each feature patch of size $9 \times 15$, and leave the investigation of the optimal patch size for future exploration.

## K    ELABORATION ON INTUITION BEHIND FLES-NLP & FLES-SEQGATE

For NLP tasks, our design is motivated by the fact that token-level semantics are highly context-dependent and can change abruptly along the sequence. In this setting, a naïve sequence-level mean tends to wash out local, context-specific cues. Therefore, instead of using a global indicator as in the vision setting, FleS-NLP and FleS-SeqGate construct *token-level indicators*, so that each position is modulated by statistics adapted to its own semantic neighborhood; the integration of channel-wise cues is largely delegated to the FleS MLP in this design.

In FleS-SeqGate, to make these indicators expressive yet lightweight, we further use a *depthwise 1D convolution along the sequence* as a low-cost way for each channel to aggregate information from nearby tokens. This provides a more suitable mechanism than a simple mean for estimating context-aware per-channel importance, while keeping the computational overhead small. The design is partially inspired by scan-style state-space models (e.g., Mamba), but we adopt 1D depthwise convolutions as a much cheaper proxy, while preserving the core FleS principle of *monotonic, sign-aware recalibration of pre-activations*.

More broadly, presenting both FleS-NLP and FleS-SeqGate serves to illustrate that the decision-making perspective is not only useful for interpreting existing nonlinear mechanisms, but can also guide the design of new ones. In particular, viewing nonlinear activations through this lens naturally leads to token-level indicators and lightweight sequence-wise aggregation as principled ways to handle context-dependent semantics in NLP, analogous to how attention has inspired a variety of subsequent token-mixing architectures.

## L    POTENTIAL LIMITATIONS

We note that one practical limitation of the current vision instantiation of FleS for dense recognition tasks is that a spatial window size needs to be chosen to compute the indicator statistics. In practice, we select this neighborhood (window) size empirically based on the input resolution and the typical pixel extent of objects. For example, on COCO we adopt a $9 \times 15$ window, since images are typically resized to $800 \times 1333$ and small objects usually occupy more than $10^2$ pixels, so such a window can capture relatively valid class-relevant statistics. We then use MLPs to integrate cross-window cues by scanning over the image to construct the indicators. This additional design choice slightly increases the deployment burden.

## M    POTENTIAL EXTENSIONS

As a possible direction for future work, we plan to explore FleS variants with improved adaptability, in particular unified and adaptive schemes for extracting valid, fine-grained rectified statistics on pre-activations to construct more effective indicators.

## LLM USAGE

ChatGPT was used to aid in polishing the writing. Specifically, it was employed to correct grammar, improve readability, and refine the clarity of sentences.

