# OpenReview forum: "Toward Principled Flexible Scaling for Self-Gated Neural Activation"
_ICLR.cc/2026/Conference — ICLR 2026 Poster_

### Official Review · Reviewer_HdQ3 · 2025-10-26

**Soundness:** 4
**Presentation:** 3
**Contribution:** 3
**Rating:** 6
**Confidence:** 4

**Summary:**

This paper proposes a novel self-gated activation function, FleS (Flexible Scaling for Self-gated activation), aimed at addressing the Non-Local Tension (NLT) issue in Transformer architectures. From a decision-theoretic perspective, the authors argue that conventional activation functions (e.g., GELU, SiLU) exhibit convergence limitations when handling high-response features, reducing the efficiency of non-local information utilization. FleS adaptively adjusts the activation function’s boundary and steepness via dynamic vertical and horizontal scaling factors (κ_ve, κ_ho). Experiments on ImageNet, CIFAR-100, and COCO benchmarks demonstrate significant performance improvements. The work combines solid theoretical analysis, novel design, and comprehensive experiments, providing a new interpretable perspective for activation function modeling.

**Strengths:**

# **Strengths**

1. **Originality and Significance:** The paper identifies and clearly defines a novel and important problem, *Non-Local Tension (NLT)*. It explains why many advanced activation functions that perform well on CNNs fail to provide similar improvements in Transformer architectures. This is not only a technical innovation but also a conceptual contribution, offering a new perspective for understanding and improving activation mechanisms in modern neural networks.

2. **Theoretical Quality:** The paper provides solid theoretical support for both the problem and the proposed solution. The authors construct a clear logical chain: from NLT to *Trivially Discernible Gating Weights (TDGW)*, and then to the root cause, *Convergence Limitation (CL)*. The analysis is rigorous, accompanied by intuitive explanations and formal theorems. FleS is tightly designed around this theory, using *effective average response* (considering only positive responses) to generate scaling factors, which is both a clever and theoretically justified heuristic.

3. **Comprehensive Experimental Validation:** The experiments are thorough and well-designed, covering multiple tasks such as image classification, object detection, and natural language processing. Various network architectures are evaluated, including Swin Transformer, PoolFormer, and ResNet. FleS consistently demonstrates significant performance gains across all settings, strongly supporting its effectiveness, generalization, and robustness.

4. **Clarity:** The paper is well-written, clearly structured, and logically coherent. Each part—from problem formulation, theoretical analysis, method design, to experimental validation—is clearly presented. Figures (e.g., Figures 1 and 2) help readers grasp core concepts and understand FleS’s operational mechanism, making complex theory and methodology accessible.

**Weaknesses:**

# **Weaknesses**

1. **Lack of a Unified Analysis Framework:** The paper mainly compares FleS with GELU, Meta-ACON, and other methods. Explaining the relationship between these approaches and FleS from a unified theoretical perspective would strengthen the academic rigor and interpretability of the work.

2. **Complexity and Computational Overhead:** The practical version of FleS introduces an additional MLP, increasing the parameter count (e.g., Swin-Min from 11.9M to 13.8M), approximately a 10% increase that is not entirely negligible. The paper does not report inference speed (FPS); it is recommended to quantify the trade-off between computational overhead and performance improvement.

3. **Hyperparameter Sensitivity:** FleS introduces new hyperparameters, such as the MLP channel reduction rate and the neighborhood size (9×15) for computing statistical measures. These settings are somewhat empirical, and further analysis is needed to understand the sensitivity of model performance to these parameters.

4. **Insufficient Elaboration of the Decision-Theoretic Perspective:** Although the paper claims to be inspired by decision theory, the connection between the theoretical principles and the design of FleS remains high-level. Clearly specifying which decision-theoretic principles directly guided the FleS design would strengthen the motivation and theoretical foundation.

**Questions:**

# **Questions**

1. **Interaction with Normalization Layers:**
   FleS does not consider the effects of BN or LN, but these normalization layers are widely used in modern networks. Could LN, which normalizes along the channel dimension, interfere with FleS’s computation of channel-wise statistics?

2. **Scope of Non-Local Tension (NLT):**
   FleS is also effective on CNNs such as ResNet. Does this indicate that the NLT problem exists in modern CNNs as well, or is the observed performance improvement mainly due to the general benefits of the adaptive scaling mechanism?

3. **Implementation Details:**
   In the COCO experiments, how was the 9×15 neighborhood size chosen? Were other sizes tested? For NLP tasks, why do FleS-NLP and FleS-SeqGate use a *token-level indicator* and *depthwise separable 1D convolution*? Could the authors provide intuition or rationale for these design choices?

4. **Applicability to Large Models:**
   Are there any numerical stability or training bottlenecks when applying FleS to large models such as ViT-L, LLaMA, or T5?

5. **Version Consistency:**
   The paper presents multiple FleS variants (FleS-Proto, FleS, FleS-NLP, FleS-SeqGate). Is it possible to provide a unified version that supports multiple tasks without task-specific adaptations?

---

> ### Author Response · Authors · 2025-11-25
> **Response to Reviewer HdQ3 (1)**
>
> We sincerely thank Reviewer HdQ3 for the careful review and for the comprehensive, insightful comments. These suggestions are highly valuable for improving the clarity of the paper and for enriching its empirical validation. We address the comments below and will incorporate the corresponding revisions into a revised manuscript, which we will upload as soon as it is complete.
>
> Notably, in the responses below we slightly **reorder the questions** to **improve the coherence** of our answers.
>
> ---
>
> **W4: Further clarification of the connection between our interpretation and decision-making**
> > **"Insufficient Elaboration of the Decision-Theoretic Perspective: Although the paper claims to be inspired by decision theory, the connection between the theoretical principles and the design of FleS remains high-level. Clearly specifying which decision-theoretic principles directly guided the FleS design would strengthen the motivation and theoretical foundation"**
>
> **Response**: We appreciate Reviewer HdQ3's constructive suggestion, which we find exceptionally valuable for improving the exposition of our interpretation and the induced methodological insights.
> Below, we further clarify the connections between *our conceptual interpretation* and *decision-making*.
>
> Our initial goal is to interpret possible mechanistic causes of *non-local tension* phenomenon and to distill methodological principles that can guide the design of remedies.
>
> We found that interpreting the neural affine–activation pipeline through a decision-making lens gives an intuitively accessible account of *non-local tension* (as outlined in the manuscript) and naturally suggests activation-scaling–based solutions.
>
> More concretely, our interpretation is **largely inspired by *grey relational analysis***, a typical multi-criteria decision-making (MCDM) model.
> The connection can be summarized as follows:
>
> 1. **Objective.** In MCDM, given a task, the objective is to score the importance of *alternative solutions* based on a set of *criteria* (typical examples include profit-type indicators, where larger values are better, and cost-type indicators, where smaller values are better), and then recommend solutions according to their overall scores.
>
> 2. **A special problem setting.** Grey relational analysis focuses on a special situation: when the given criteria are not directly comparable in terms of their raw numerical magnitudes, how can we still evaluate and rank alternatives? One strategy is that, if we can identify one or more *ideal solutions* under the given criteria, then we can score each alternative by its similarity to these ideal solutions.
>
> 3. **An analogy: viewing the neural affine–activation pipeline as an MCDM problem.** We view the neural affine–activation pipeline as a natural extension of this setting:
>    - **Task:** optimize the neuron's weights, in particular each filter (weight vector).
>    - **Ideal solution(s):** the filter vectors themselves, as carriers of learned meaningful patterns; once sufficiently trained, each filter approximates an underlying “ideal pattern”.
>    - **Alternative solutions:** the feature vectors, which provide candidate update directions for each filter.
>    - **Criteria:** the channels, since both features and filters are represented as channel-wise vectors.
>    - **Alternative-to-ideal similarity:** the affine transformation (inner product plus bias), which we treat as a generalized similarity that also considers the magnitudes of the feature and filter.
>
> From this viewpoint, the raw channel values of a feature vector do not directly tell us how “important” that feature is for updating any given filter. Instead, its importance for a specific filter is reflected by the affine projection onto that filter. This clarifies how we bridge the neural affine–activation pipeline to the MCDM scenario above.
>
> **On top of this analogy**, the manuscript **explains how we interpret *non-local tension* under this decision-making lens and why learning flexible activation scales (FleS) is a natural way to alleviate it**: activation scaling implements a task-guided reallocation of **importance** budget across features and channels, analogous to redistributing decision-making weight according to their utility.

---

> ### Author Response · Authors · 2025-11-25
> **Response to Reviewer HdQ3 (2)**
>
> **W2: Evaluation on practical efficiency (costs other than FLOPs and parameters)**
> > **"Complexity and Computational Overhead: The practical version of FleS introduces an additional MLP, increasing the parameter count (e.g., Swin-Min from 11.9M to 13.8M), approximately a 10% increase that is not entirely negligible. The paper does not report inference speed (FPS); it is recommended to quantify the trade-off between computational overhead and performance improvement"**
>
> **Response**: We appreciate Reviewer HdQ3's constructive suggestion. In response, and going beyond theoretical FLOPs, we profile the practical runtime overhead introduced by FleS. Specifically, we measure inference throughput on a single RTX 3090 GPU under a common `torch.compile` setup for Swin-Min, Swin-Micro, and Swin-Tiny on ImageNet:
>
> | Activation | Backbone | FLOPs | Throughput (img/s) | Top-1 (%) ↑ |
> |---|---|:---:|:---:|:---:|
> | GELU | Swin-Min | 1.6G | 4207.2 | 68.7 |
> | FleS | Swin-Min | 1.6G | 4011.3 | **71.4** |
> | GELU | Swin-Micro | 2.6G | 2775.6 | 78.7 |
> | FleS | Swin-Micro | 2.6G | 2616.8 | **80.3** |
> | GELU | Swin-Tiny | 4.4G | 1622.5 | 81.3 |
> | FleS | Swin-Tiny | 4.4G | 1545.2 | **82.3** |
>
> FleS incurs an acceptable, modest overhead of about 4–6% in practical efficiency across these backbones, while consistently improving Top-1 accuracy by a substantial margin.
> This validates the practical efficiency of FleS's design.
> ___
>
> **W3 & Q3.1: On hyperparameter sensitivity of neighborhood size for dense recognition**
> > **"FleS introduces new hyperparameters, such as the MLP channel reduction rate and the neighborhood size (9×15) for computing statistical measures. These settings are somewhat empirical, and further analysis is needed to understand the sensitivity of model performance to these parameters & In the COCO experiments, how was the 9×15 neighborhood size chosen?  Were other sizes tested?"**
>
> **Response**: We thank Reviewer HdQ3 for the insightful suggestion.
>
> In practice, we select the neighborhood (i.e., window) size empirically based on the image resolution and the typical pixel extent of objects. For example, on COCO we adopt a default $9 \times 15$ window, given that COCO images are typically resized to $800 \times 1333$ and small objects usually occupy more than $10^2$ pixels. This way, a $9 \times 15$ window can capture relatively valid class-relevant statistics. We then leverage an MLP to integrate cross-window cues by scanning over the image to construct the indicators.
>
> In accordance with Reviewer HdQ3's suggestion, we conduct a targeted evaluation by varying the neighborhood size on COCO detection using PoolFormer-S12 with RetinaNet, and we summarize the corresponding observations below.
>
> **Observations.**
> We observe that FleS is reasonably robust when reducing the window from $9\times15$ to $9\times9$ (mAP 36.2→36.1), but a smaller $5\times5$ window leads to a drop, while still outperforming the GELU baseline. These results suggest that using a window covering roughly 50 or more pixels is preferable, while still focusing on a compact local range to facilitate the extraction of relatively simple class-relevant cues.
>
> Additionally, we also evaluate a new design that introduces learnable patch statistics into the indicator module to further improve adaptability. Specifically, by replacing the hard “positive-only + rectified mean” aggregation with a Softplus-based re-calibration followed by a depthwise convolution with fixed kernel size (optionally using dilation to enlarge the effective receptive field), we obtain **FleS-Mod**. This variant continues the idea behind moving from FleS-Proto to FleS: previously we used token-wise MLPs to mitigate noisy batch statistics; here we use adaptive local depthwise convolutions to better capture scale-related per-channel cues.
>
> On ImageNet classification, this modified indicator improves PoolFormer-S12-FleS from 79.4% to 79.9% Top-1, compared to 77.2% Top-1 for the original PoolFormer-S12, and it also shows slight improvements on COCO. We plan to further investigate to what extent this design direction can improve performance on downstream dense visual recognition tasks.
>
> *Comparative evaluation on MS COCO object detection (PoolFormer-S12 encoder with RetinaNet).*
>
> | Activation | Window-size | mAP (%)↑ | AP50 (%)↑ | AP75 (%)↑ | APS (%)↑ | APM (%)↑ | APL (%)↑ |
> |---|---|---|---|---|---|---|---|
> | GELU | — | 35.5 | 55.5 | 37.5 | 19.5 | 38.7 | 46.3 |
> | FleS | $9\times 15$ | 36.2 | 57.0 | 38.1 | 20.7 | 40.1 | 46.8 |
> | FleS | $9\times 9$ | 36.1 | 56.6 | 37.9 | 20.5 | 40.0 | 46.8 |
> | FleS | $5\times 5$ | 35.8 | 57.0 | 37.5 | 20.0 | 39.6 | 46.7 |
> | FleS-Mod | $5\times 5$ | 36.3 | 56.9 | 38.3 | 20.7 | 40.3 | 47.0 |

---

> ### Author Response · Authors · 2025-11-25
> **Response to Reviewer HdQ3 (3)**
>
> **Q1: Interaction with normalization**
> > **"FleS does not consider the effects of BN or LN, but these normalization layers are widely used in modern networks. Could LN, which normalizes along the channel dimension, interfere with FleS’s computation of channel-wise statistics?"**
>
> **Response**: We sincerely thank Reviewer HdQ3 for raising this insightful question.
>
> In response, we design a two-part ablation study to investigate two related questions: (i) how FleS interacts with the LN/BN used in attention–FFN blocks, and (ii) whether it is beneficial to insert an additional normalization layer on the pre-activations before feeding them into the FleS's MLP used to construct channel-wise indicators. Below, we address these two questions in turn.
>
> **(i) Compatibility with LN/BN in the backbone.**
>
> **Setup.**
> We run a controlled comparison on ImageNet with Swin-Min, where we replace LN by BN in each Transformer block, using the default training recipe for all variants.
>
> **Observations.**
> BN improves the original Swin-Min by a modest but noticeable margin (Top-1: 68.7 $ \rightarrow $ 69.3).
> When combined with FleS, however, BN yields a markedly larger gain (71.4 $ \rightarrow $ 73.8), which is notable given that FleS already operates in a higher-accuracy regime with stronger diminishing returns.
> We are interested in interpreting this unexpected phenomenon, which may indicate an interesting interaction between FleS's activation scaling and batch-normalized feature statistics. A deeper analysis is beyond the scope of the present discussion, but it may provide a promising direction for our further investigation.
>
> | Activation | Backbone | Normalization | #Params | FLOPs | Top-1 (%) $\uparrow$ |
> |---|---|---|:---:|:---:|:---:|
> | GELU | Swin-Min | LN | 11.8M | 1.6G | 68.7 |
> | FleS | Swin-Min | LN | 13.0M | 1.6G | **71.4** |
> | GELU | Swin-Min | BN | 11.8M | 1.6G | *69.3* |
> | FleS | Swin-Min | BN | 13.0M | 1.6G | ***73.8*** |
>
> **(ii) Effect of Normalizing Pre-Activations Before the FleS's MLP.**
>
> **Setup.**
> This experiment adopts the same configurations suggested in experiment (i), and introduce two tailored variants for comparative evaluations: (a) FleS + BN on the indicator input; and (b) FleS + LN on the indicator input.
>
> **Observations.**
> We observe that (a) using BN does not lead to noticeable improvements over the default design, while (b) using LN demonstrates a slight drop in accuracy.
> Based on these results and to keep the design simple, we recommend using the vanilla setting without additional normalization as the default configuration.
>
> | Activation | Backbone| #Params | FLOPs | Top-1 (%) $\uparrow$ |
> |---|---|:---:|:---:|:---:|
> | GELU | Swin-Min | 11.8M | 1.6G | 68.7 |
> | FleS (vanilla) | Swin-Min | 13.0M | 1.6G | **71.4** |
> | FleS ($+$ BatchNorm) | Swin-Min | 13.0M | 1.6G | **71.4** |
> | FleS ($+$ LayerNorm) | Swin-Min | 13.0M | 1.6G | ***71.1*** |
>
> At the same time, for applications/implementations where numerical stability is required, we consider it reasonable to optionally add a normalization layer on the indicator path and to tune this choice (e.g., BN, LN, or GN) based on the data characteristics.

---

> ### Author Response · Authors · 2025-11-25
> **Response to Reviewer HdQ3 (4)**
>
> **Q2. On scope of *non-local tension (NLT)***
> > **"FleS is also effective on CNNs such as ResNet. Does this indicate that the NLT problem exists in modern CNNs as well, or is the observed performance improvement mainly due to the general benefits of the adaptive scaling mechanism?"**
>
> **Response**: We appreciate the reviewer’s thoughtful question.
>
> Our view is that, on classical CNNs (e.g., ResNet), the performance gains of FleS are mainly driven by the **general benefits of its adaptive scaling mechanism**, rather than by the same degree of non-local tension (NLT) observed in token-mixer–based architectures (e.g., Swin and PoolFormer). Specifically, **in relatively deep blocks**, a classical CNN can also capture non-local information to some extent through its *enlarged effective receptive field*, but the resulting non-local signals are more diffuse and less explicit than those in token-mixer–based blocks, so the opportunities for strongly triggering NLT are comparatively rarer. This view, where FleS primarily boosts CNN blocks via general adaptive scaling, is consistent with our empirical observations.
>
> As reported in Tables 2 and 3 (main text), on ResNet-50 (as a representative of classical CNNs), two SOTA activation functions, IIEU (Top-1: 79.7; parameters: 25.6M; and FLOPs: 4.2G) and AdaShift (Top-1: 79.9; parameters: 25.6M; and FLOPs: 4.1G), show slightly inferior results to FleS (Top-1: 80.1; parameters: 28.1M; and FLOPs: 4.1G), while using fewer parameters and close FLOPs.
> In contrast, on Swin-Transformer(s) and PoolFormer, FleS exhibits a **remarkably larger margin** over all competitive activation methods. This contrast suggests that, for classical CNNs, the main benefit of FleS is brought by the adaptive scaling mechanism, whereas the *non-local tension* problem is more pronounced in architectures that models explicit non-local/token mixing paradigms.
>
> Notably, we view the insights behind the channel-wise indicator design—which models interpretable, lightweight non-local cues—as a critical factor underlying the gains of FleS on both CNN(s) and Transformers. To validate this, we conduct an additional ablation on ResNet-50 by removing the key intuition implied by our decision-making interpretation—the *monotonic sign-aware recalibration that emphasizes positive and suppresses negative pre-activation responses*—the gain of FleS on ResNet-50 drops notably (from 80.1% to 79.4% Top-1), making it inferior to IIEU and AdaShift. This supports our view that, on classical CNNs without explicit block-wise non-local modeling, the gains mainly arise from FleS's principled adaptive scaling mechanism, with the *decision-making–guided, sign-aware indicator modeling* serving as the key driver of strong gains.
>
> ---
>
> **Q3.2. Intuition behind the token-level indicator and 1D depthwise convolution in FleS-NLP / FleS-SeqGate***
> > **"For NLP tasks, why do FleS-NLP and FleS-SeqGate use a token-level indicator and depthwise separable 1D convolution? Could the authors provide intuition or rationale for these design choices?"**
>
> **Response**: We thank the reviewer for this insightful question.
>
> For NLP, our design is motivated by the attribute that **token-level semantics are highly context-dependent and can change abruptly along the sequence**. In this setting, a naïve sequence-level mean tends to wash out local, context-specific cues.
>
> Therefore, instead of using a global indicator as in the vision setting, FleS-NLP and FleS-SeqGate construct **token-level indicators**, so that each position is modulated by statistics adapted to its own semantic neighborhood (i.e., the integration of channel-wise cues is largely delegated to the FleS MLP in this design).
> Particularly, for FleS-SeqGate, to make these indicators expressive yet lightweight, we use a **depthwise 1D convolution along the sequence** as a low-cost way for each channel to aggregate information from nearby tokens, which is more suitable than a simple mean for estimating context-aware per-channel importance. This is partially inspired by scan-style state-space models (e.g., Mamba), but we adopt 1D depthwise convolutions as a much cheaper proxy while preserving the core FleS idea of **monotonic, sign-aware recalibration of pre-activations**.
>
> As a remark, we would like to clarify why we present both FleS-NLP and FleS-SeqGate. Our goal in this work is not only to propose a convenient practical module (i.e., FleS), but also to introduce an intuitive and accessible interpretation framework that helps make the design of nonlinear mechanisms more transparent. We use FleS as a concrete example of how a decision-making–based perspective can serve as a useful analysis tool. An important way to assess the effectiveness of such a framework, in our view, is whether it can continually inspire new intuitions, principled methodologies, and practical designs. This is analogous to how the attention mechanism has inspired a wide range of subsequent token-mixing architectures.

---

> ### Author Response · Authors · 2025-11-25
> **Response to Reviewer HdQ3 (5)**
>
> **Q4: FleS on a 100M+ Transformer**
> > **"Are there any numerical stability or training bottlenecks when applying FleS to large models"**
>
> **Response**: We thank Reviewer HdQ3 for the thoughtful comments.
>
> To further assess the applicability of FleS across different Transformer families and on larger attention-based models, we choose the efficient Transformer Hiera and conduct experiments on ImageNet using a 100M+ Hiera-Large-Slim configuration (local-to-global, multi-scale attention; layer setting [2, 3, 20, 3], for a total of 28 attention–FFN blocks; base embedding dimensions [144, 288, 576, 1152]). This variant is slightly slimmed compared to the original Hiera-Large.
>
> *Note that this configuration is close to the upper model-size limit of what we can train on our current hardware within the rebuttal period.*
>
> Since an official MAE training–evaluation recipe for Hiera is not yet available, we follow the standard 300-epoch Swin-Base's non-MAE recipe and train each model from scratch, reducing the base learning rate from $1.0\times 10^3$ to $0.6\times 10^3$ per batch of 1024 to better match Hiera's empirical optimization characteristics.
>
> **Observations.** FleS improves the original GELU model from 82.9% to 83.2% Top-1 on ImageNet, with very slight additional computational overhead. Given the strong diminishing returns at the ~83% accuracy regime, this constitutes a meaningful improvements.
>
> | Activation | Backbone| #Params | FLOPs | Top-1 (%) $\uparrow$ |
> |---|---|:---:|:---:|:---:|
> | GELU | Hiera-Large-Slim | 131.0M | 22.9G | 82.9% |
> | FleS | Hiera-Large-Slim | 147.7M | 23.0G | 83.2% |
>
> ---
>
> **Q5: Version consistency**
> > **"The paper presents multiple FleS variants (FleS-Proto, FleS, FleS-NLP, FleS-SeqGate). Is it possible to provide a unified version that supports multiple tasks without task-specific adaptations?"**
>
> **Response**: We appreciate Reviewer HdQ3's constructive comment.
>
> If we focus on a unified practical design at current stage, we would recommend the dwconv-based variant introduced in “W3 & Q3-1” (i.e., FleS-Mod), which uses a numerically stable Softplus-based recalibrator together with a depthwise convolution to adaptively integrate recalibrated channel statistics into activation scales. For sequence-input tasks, this corresponds to either FleS-NLP with an added 1D depthwise convolution or FleS-SeqGate with the gating module removed. FleS-Mod typically achieves slightly better performance than vanilla FleS, but at the cost of additional parameterized operations and thus a somewhat heavier design. We plan to further investigate this problem with the aim of developing a more comprehensive model.

---

> ### Author Response · Authors · 2025-11-25
> **Response to Reviewer HdQ3 (6)**
>
> **W1: On a unified analysis framework for empirical comparisons**
> > **"Lack of a Unified Analysis Framework: The paper mainly compares FleS with GELU, Meta-ACON, and other methods. Explaining the relationship between these approaches and FleS from a unified theoretical perspective would strengthen the academic rigor and interpretability of the work"**
>
> **Response**: We sincerely thank Reviewer HdQ3 for this insightful comment.
>
> We agree that making this structure clearer would improve the clarity and rigor of the work.
>
> In the original submission, we briefly grouped the compared activation functions into categories: "standard," "static self-gated," "dynamic," and "other types," with citation-based references in the second paragraph of Section 5 (Experiment). However, due to page limits, we omitted explicitly listing method names and did not make the unifying view fully explicit, which may cause confusion.
>
> In accordance with Reviewer HdQ3's suggestion, we will add a unified introduction to FleS and the popular and SOTA activation functions we compare against in Section 5 (Experiments).
>
> **Concretely**, we view FleS and the competing methods under a common self-gated form:
> $
> \phi(\tilde{x}) = \rho(\tilde{x}) \cdot \tilde{x}\,
> $
> where $\rho(\cdot)$ is a (input-dependent) weighting function.
> From our decision-making lens, **FleS and the compared methods can be distinguished by the properties of $\rho(\cdot)$ (and thus the overall behavior of $\phi(\cdot)$)**.
> We summarize the main competing activation methods as follows:
> - **Monotonic activation functions $\phi(\cdot)$ with discontinuous $\rho(\cdot)$**: Softplus (Dugas et al., 2000); ReLU (Nair & Hinton, 2010); and StarReLU (Yu et al., 2024).
> - **Static self-gated functions $\phi(\cdot)$ with a smooth, monotonic $\rho(\cdot)$**: GELU (Hendrycks & Gimpel, 2016); SiLU (Elfwing et al., 2018); and Mish (Misra, 2020).
> - **Dynamic self-gated functions $\phi(\cdot)$ with a modified $\rho(\cdot)$ integrating adaptive components**: Pserf (Biswas et al., 2022); SMU (Biswas et al., 2022); Meta-ACON (Ma et al., 2021); IIEU (Cai, 2023); AdaShift (Cai, 2024); and **FleS**.
>
> Within the same framework, **FleS** can be viewed as a particular dynamic self-gated activation where $\rho(\cdot)$ is constructed from **sign-aware, channel-wise statistics of a reference feature group**, which are then processed via small MLPs to produce a feature-importance-calibrated adaptive scaling of activations.
>
> This presents the relationships between FleS and the competing methods more transparently.
>
> ---
>
> **Remarks.**
> We once again thank Reviewer HdQ3 for the careful review and constructive suggestions, and we would be glad to discuss any further comments or perspectives Reviewer HdQ3 may wish to share.

---

> ### Author Response · Authors · 2025-11-28
> **Summary of changes for Reviewer HdQ3**
>
> We once again thank Reviewer HdQ3 for the careful review and insightful comments.
> We have addressed all of your comments and incorporated the corresponding changes into the revised manuscript.
>
> For ease of reference, the changes made in accordance with your constructive suggestions are summarized below,
> together with where they appear in the revised manuscript (with all changes highlighted in blue):
>
> - **Intuition 1.1 in Section 1 (page 2)** – for **“W4: Further clarification of the connection between our interpretation and decision-making”**.
>
> - **Section E.6 (page 27)** – for **“W2: Evaluation on practical efficiency (costs other than FLOPs and parameters)”**.
>
> - **Section E.8 (pages 28–29)** – for **“W3 & Q3.1: On hyperparameter sensitivity of neighborhood size for dense recognition”**.
>
> - **Section E.5 (pages 26–27) and Section E.9 (page 29)** – for **“Q1: Interaction with normalization (parts (i) and (ii), respectively)”**.
>
> - **Section E.10 (pages 29–30)** – for **“Q2: On the scope of non-local tension (NLT)”**.
>
> - **Section K (pages 38)** – for **“Q3.2: Intuition behind the token-level indicator and 1D depthwise convolution in FleS-NLP / FleS-SeqGate”**.
>
> - **Section E.11 (pages 30)** – for **“Q4: FleS on a 100M+ Transformer”**.
>
> - **Section M (page 38)** – for **“Q5: Version consistency”**.
>
> - **Paragraph 2 in Section 5 (page 8)** – for **“W1: On a unified analysis framework for empirical comparisons”**.
>
> We are deeply grateful for the time and care Reviewer HdQ3 invested in reading our manuscript and engaging with its details. Your careful review and thoughtful suggestions have substantially contributed to both the clarity and the scope of this work.

---

### Official Review · Reviewer_Ewa3 · 2025-10-31

**Soundness:** 3
**Presentation:** 1
**Contribution:** 3
**Rating:** 6
**Confidence:** 4

**Summary:**

This paper introduces a new gating mechanism. It builds on the intuition that having a fixed gating function can be problematic: If all features are salient, then a classical gating function like GELU will saturate and not meaningfully discriminate between these features. To address this, the authors introduce a new gating function which works as follows:

First, for each channel they calculate the mean of the positive features. This is a measure of how likely this feature is going to saturate the activation function. Then, they feed these statistics to two small MLPs. These MLPs provide two scaling factors (of the inputs and the outputs).

The authors argue that this is particularly important for transformer networks, since the attention mechanism in transformers is likely to lead to having many salient activations within a channel. The reason the authors use an MLP rather than deriving scaling factors directly from the inputs is because it is (1) not possible to derive class-specific statistics at test time when the class is unknown, and (2) in shuffled batches the amount of information per-class can be very noisy. Hence, an MLP is a more appropriate way of estimating appropriate scaling factors given the small amount of noisy information found in a single batch.

The empirical results in the paper are very encouraging.

**Strengths:**

* Strong empirical results
* A very flexible method that applies to many networks/architectures
* Robust method that seems relatively insensitive to hyperparameters

**Weaknesses:**

* A bit heuristic (e.g. using positive-only means)
* The connection between the theory and the practical algorithm is tenuous (given the use of MLPs in the final algorithm). Although it is interesting to see how the algorithm was motivated by the authors, it does end up feeling a lot like a post-hoc justification. I would prefer it if the authors approach their work as purely experimental and use this space in the paper for more exhaustive empirical validation.
* Subpar presentation: I found the text needlessly filled with jargon, non-standard terminology and abbreviations, drawing spurious connections to other theories, too densely written, etc.
  * For example, abbreviating "activation" to "Act" and referring to pre-activations as "projected responses" really doesn't help with readability.
  * Then there is the list of newly introduced terms and accompanying abbreviations: non-local tension (NLT), convergence limitation (CL), trivially discriminative gating weights (TDGW), etc.
  * Connections to decision-making and neuronal stimulus-response mechanisms.
  * For example: Figure 1 should be a clear explanation of the problem this paper tries to tackle, in a way that readers can grok it after just reading the abstract. Instead, readers are presented with the following sentence, which is barely comprehensible (and contains several grammar mistakes): "[...] the origin of the NLT and the key insights behind FleS shows how CL triggers TDGW problem, which in turn neutralizes the influence of external non-local cues through Act. and show two qualitative insights into addressing non-local tension: vertical and horizontal dynamic scaling strategies."

**Questions:**

* Did you run any ablation studies over the MLP? Does a single linear layer suffice or is it essential to have non-linearities in the MLP?

---

> ### Author Response · Authors · 2025-11-23
> **Response to Reviewer Ewa3 (1)**
>
> We sincerely thank Reviewer Ewa3 for the constructive and insightful comments on the writing, activation design, and experimental design of our paper. These comments have been highly beneficial to us.
>
> In response, we have conducted extensive additional experiments, detailed below. We also clarify several points where the original exposition may have lacked clarity. These revisions, along with all feedback received during the rebuttal period, will be incorporated into the updated manuscript as soon as possible.
>
> ---
>
> **Q1: Linear layer vs. MLP**
> > **"Did you run any ablation studies over the MLP? Does a single linear layer suffice or is it essential to have non-linearities in the MLP?"**
>
> **Response**: We thank Reviewer Ewa3 for the constructive suggestion. In response, we present a new ablation study below.
>
> **Setup.**
> Following Reviewer Ewa3's suggestion, we conduct a targeted ablation on ImageNet with the Swin-Min and Swin-Micro backbones, where we replace the FleS MLP by a low-rank linear module (i.e., an MLP with a reduction ratio but without nonlinearity, denoted as **FleS-LRL**) under a comparable parameter budget.
>
> **Observations.**
> On Swin-Min backbone, we test two reduction ratios ($r=32$ as the default and $r=24$ with slightly more parameters). In both cases, FleS-LRL shows a small but consistent drop in accuracy compared to the original FleS, while both variants still clearly outperform the GELU baseline. On the deeper backbone Swin-Micro, the performance gap between FleS-LRL and the original FleS becomes more pronounced (in a regime where marginal gains in Top-1 are harder to obtain). In terms of cost, FleS-LRL and FleS have almost identical FLOPs and very similar throughput (measured on a single RTX 3090 with a common `torch.compile` environment), and both introduce only a light overhead over the baselines. Given the consistently stronger accuracy, we recommend the MLP-with-reduction version as the default configuration.
>
> | Activation      | Backbone   | Ratio $r$ | #Params | FLOPs | Throughput (img/s) | Top-1 (%) $\uparrow$ |
> |-----------------|-----------|:---------:|:-------:|:-----:|:------------------:|:-----------:|
> | GELU            | Swin-Min  | —         | 11.8M   | 1.6G  | 4207.2             | 68.7        |
> | FleS-LRL        | Swin-Min  | 32        | 13.0M   | 1.6G  | 4016.7             | *71.0*      |
> | FleS (original) | Swin-Min  | 32        | 13.0M   | 1.6G  | 4011.3             | **71.4**    |
> | FleS-LRL        | Swin-Min  | 24        | 13.4M   | 1.6G  | 4013.6             | *71.2*      |
> | FleS (original) | Swin-Min  | 24        | 13.4M   | 1.6G  | 4001.0             | **71.5**    |
> | GELU            | Swin-Micro| —         | 21.1M   | 2.6G  | 2775.6             | 78.7        |
> | FleS-LRL        | Swin-Micro| 32        | 23.5M   | 2.6G  | 2633.8             | *79.7*      |
> | FleS (original) | Swin-Micro| 32        | 23.5M   | 2.6G  | 2616.8             | **80.3**    |
>
> **Remarks.**
> We posit the main reason for the gap between FleS-LRL and FleS is that the mapping from channel-wise rectified statistics to effective activation scales is generally more complex than a single affine transform can capture. In the FleS-Proto experiment (Table 1 in Section 4.1), when we had access to very clean, class-aligned channel statistics, even simple mappings without an MLP could already produce strong scales. However, in realistic training regimes, the statistics are noisy and mixed across classes and contexts. In this setting, the small MLP in FleS acts as a lightweight universal approximator that *purifies* these statistics within its effective receptive field and transforms them into scales that better adapt to the current activation process. Empirically, this additional nonlinearity provides a consistent accuracy benefit over FleS-LRL at essentially the same computational cost.
>
> We thank Reviewer Ewa3 again for suggesting this helpful empirical investigation.

---

> ### Author Response · Authors · 2025-11-23
> **Response to Reviewer Ewa3 (2)**
>
> **W2: Further Empirical Validation**
> > **"I would prefer it if the authors approach their work as purely experimental and use this space in the paper for more exhaustive empirical validation"**
>
> **Response**: We thank Reviewer Ewa3 for encouraging more empirical investigation to better support our methodological intuitions. In response, we present two additional studies below:
>
> - **W2.1:** A new mechanistic probing experiment that explicitly measures and visualizes the gradient contributions of positive vs. negative features in FleS.
> - **W2.2:** A ablation study of FleS and GELU baseline with batch normalization in vision transformers, by replacing layer normalization with batch normalization in Swin-Min and Swin-Min-FleS under a shared ImageNet training recipe.
>
> ---
>
> **W2.1: Empirical Investigation of Positive–Negative Gradient Contributions**
>
> **Setup.**
> We analyze how the relative gradient contributions of positive vs. negative responses evolve during training on ImageNet with Swin-Min + FleS. For each epoch and each hierarchical stage, we compute the average gradient magnitude on positions where the FleS output is positive(-valued) and where it is negative(-valued), and then take the ratio “positive / negative” as a measure of their relative influence on optimization.
>
> **Observations.**
> We observe two consistent trends:
>
> - Across depth (averaged over all epochs 0–119), gradients on positive responses are substantially larger than on negative ones. The mean positive-to-negative gradient ratio is about $5.3\times$ in stage 1, $7.9\times$ in stage 2, $12.7\times$ in stage 3, and $13.8\times$ in stage 4.
> - Across training time, this asymmetry strengthens from early to mid training and then remains high:
>   - Early phase (epochs 0–9): the ratios are around $3.1\times$, $4.1\times$, $5.5\times$, and $8.5\times$ for stages 1–4.
>   - Mid phase (epochs 40–79): they grow to roughly $5.6\times$, $8.5\times$, $13.3\times$, and $14.5\times$.
>   - Late phase (epochs 80–119): they stay high at about $4.9\times$, $6.7\times$, $10.3\times$, and $14.4\times$, all clearly above the early-phase values.
>
> Overall, as training proceeds, positive responses increasingly dominate the effective gradient budget, especially in deeper stages, while negative responses carry much weaker gradients.
>
> This empirical pattern helps clarify our intuition behind FleS's design for channel-wise indicators: the main optimization signal is concentrated on the positive side, and emphasizing it in the indicator avoids cancellation while still allowing negative features to be exploited by other filters in the current layer (we further elaborate this in W3, including connections to decision-making and neuronal stimulus–response mechanisms). We will include the corresponding curves in the revised manuscript for a more detailed visualization.
>
> **It is worth noting that** we employ a *positives-retained, negatives-attenuated* indicator as the simplest and most direct mechanism for differentiating positive and negative responses. This is a pragmatic default rather than a hard design requirement. In Appendix C.4, we discuss softer alternative schemes that explicitly separate or asymmetrically weight positive and negative contributions at the channel level. The core principle is to introduce differential feature rectification that mitigates the confusion between relatively positive and relatively negative contributions within each channel *(with a more detailed elaboration in W3)*.
>
> ---
>
> **W2.2: Empirical investigation of FleS's interaction with batch normalization**
>
> **Setup.**
> We run a controlled comparison on ImageNet with Swin-Min, where we replace layer normalization (LN) by batch normalization (BN) in each Transformer block, using the default training recipe for all variants.
>
> **Observations.**
> BN improves the original Swin-Min by a modest but noticeable margin (Top-1: 68.7 $ \rightarrow $ 69.3).
> When combined with FleS, however, BN yields a markedly larger gain (71.4 $ \rightarrow $ 73.8), which is notable given that FleS already operates in a higher-accuracy regime with stronger diminishing returns.
> We are interested in interpreting this unexpected phenomenon, which may indicate an interesting interaction between FleS's activation scaling and batch-normalized feature statistics. A deeper analysis is beyond the scope of the present discussion, but it may provide a promising direction for our further investigation.
>
> | Activation | Backbone | Normalization | #Params | FLOPs | Top-1 (%) $\uparrow$ |
> |---|---|---|:---:|:---:|:---:|
> | GELU | Swin-Min | LN | 11.8M | 1.6G | 68.7 |
> | FleS | Swin-Min | LN | 13.0M | 1.6G | **71.4** |
> | GELU | Swin-Min | BN | 11.8M | 1.6G | *69.3* |
> | FleS | Swin-Min | BN | 13.0M | 1.6G | ***73.8*** |

---

> ### Author Response · Authors · 2025-11-23
> **Response to Reviewer Ewa3 (3)**
>
> **W3: On exposition and terminology**
> > **"Subpar presentation"**
>
> **Response**: We sincerely thank Reviewer Ewa3 for the detailed and thoughtful feedback on the exposition, which provides concrete suggestions and examples for improving the clarity of our paper and is highly valuable to us.
>
> In accordance with your suggestions, in the revised manuscript we will:
>
> - Avoid non-standard abbreviations such as "Act." and directly use "activation".
> We will also use the standard term "pre-activation(s)" in the revised manuscript.
> - We will use simpler terms to explain our intuitions and insights and describe the key components more clearly, while minimizing abbreviations to improve readability.
> -Rewrite the sentence highlighted by the reviewer and streamline the caption and surrounding text of Figure 1 so that it plainly states what problem FleS addresses and how it changes standard activations, without heavy terminology.
>
> Additionally, we would like to explain why we chose to spend space on the decision-making view, and not only on the FleS module itself. A key goal of this work is to present decision-making interpretation as an intuitive analysis tool that can help design activation mechanisms in a more transparent and principled way.
> The decision-making view has not only guided the design of FleS, but has also supported us in interpreting other interesting phenomena. Because we find this perspective practically useful for us, we use FleS as a concrete example to show how one can build an interpretable activation module and several variants (Appendix B.2, C.3, C.4) under the decision-making lens.
>
> Furthermore, to improve the exposition of our work and ensure clearer communication, we further clarify our reasoning in W1, and will use this cleaner version to refine the corresponding analysis and discussion in the manuscript.
>
> We again thank Reviewer Ewa3 for the detailed comments on the exposition and for the helpful suggestions on improving clarity.

---

> ### Author Response · Authors · 2025-11-23
> **Response to Reviewer Ewa3 (4)**
>
> **W1: Further clarification on our use of the positive-only indicator**
> > **"A bit heuristic (e.g. using positive-only means)"**
>
> **Response**: We appreciate Reviewer Ewa3's insightful comments regarding the justification of the positive-only indicator and the connection between the theory and the practical algorithm FleS. We acknowledge that these aspects were not sufficiently emphasized in the original submission and thank the reviewer for bringing them to our attention.
>
> We clarify that the general principle behind constructing channel-wise indicators in FleS is to perform **sign-aware recalibration that emphasizes positive features and suppresses negative ones** before summarizing pre-activations. The **positive-only indicator** is introduced as **a simple instantiation** of this idea. We also present **two additional instantiations in Appendix C.4** that asymmetrically weight positive and negative contributions. Notably, **the positive-only indicator and its softer alternatives are suggested to be chosen based on the application scenario**: for ImageNet-style vision benchmarks, a positive-only design suffices, so we use it as the default configuration, whereas for NLP tasks with more abrupt token-level semantic changes we adopt a Softplus-based indicator (Appendix B.2). In the revised manuscript, we will make these modeling principles and the replaceable nature of the indicator explicit.
>
> To explain the general principle, we first clarify how positive and negative pre-activations are interpreted and utilized in a self-gated activation process, and why FleS emphasizes positive pre-activations when constructing indicators for activation scaling. We then adopt this lens in addressing Reviewer Ewa3's concern."*
>
> First, we discuss based on a self-gated activation: $\phi(\tilde{x})=\rho(\tilde{x})\cdot\tilde{x}$, where $\rho(\tilde{x}) \ge 0$. We interpret $\rho(\tilde{x})$ as a soft importance score indicating how strongly a pre-activation $\tilde{x}$ should be retained. In this perspective, *positive* and *negative* pre-activations are not globally absolute, but relative to a given filter. For example, suppose the current layer includes two filters $ \boldsymbol{w}_1, \boldsymbol{w}_2 \in \mathbb{R}^C $ with $ C \ge 2 $. For an input feature vector $ \boldsymbol{x} \in \mathbb{R}^C $, the corresponding output $ \tilde{\boldsymbol{x}} \in \mathbb{R}^2 $ has two channels. The same $ \boldsymbol{x} $ can be strongly suppressed by $ \boldsymbol{w}_1 $ (i.e., $ \tilde{x}_1 = \langle \boldsymbol{x}, \boldsymbol{w}_1 \rangle < 0 $, omitting the bias term for simplicity) while still being informative for $ \boldsymbol{w}_2 $ (i.e., $ \tilde{x}_2 = \langle \boldsymbol{x}, \boldsymbol{w}_2 \rangle > 0 $). Being down-weighted in one channel therefore does not mean that the feature is discarded by the layer; it may be emphasized in other channels whose filters align better with it.
>
> Based on this understanding, we propose that a key principle for constructing channel-wise indicators in FleS is to introduce **differential feature rectification that mitigates confusion caused by positive–negative cancellation**, by applying a monotonic (sign-aware) recalibration that emphasizes positive features and attenuates negative ones before summarizing pre-activations. This helps the statistics reflect genuine per-channel intensity, rather than confusing truly weak channels with channels that have strong but mixed-sign pre-activations. In contrast, aggregating both positive and negative pre-activations in a symmetric way may result in “$-1+1$”-style cancellation effects, which in turn increases the burden on the MLP to predict scaling factors from the indicator, because the input no longer faithfully reflects filter-level importance statistics (Table 4 (right) empirically validates this). *Notably*, even if a pre-activation is negative for the dominant filter of one channel, the same feature can still be positively emphasized in other channels; FleS only rectifies the statistics used for scaling, not the global availability of features.
>
> In particular, **we suggest using softer alternative schemes in applications with more abrupt token-level semantic changes (e.g., NLP). For example, a Softplus-based indicator incurs a slightly higher computational cost but is numerically safer. Notably, the positive-only and Softplus-based variants achieve similar performance on ImageNet, as these vision benchmarks typically exhibit mild semantic variation in appearance.**
>
> We would like to thank Reviewer Ewa3 again for these insightful comments, which helps improve the clarity of our work.

---

> ### Author Response · Authors · 2025-11-24
> **Response to Reviewer Ewa3 (5)**
>
> **W3.3: Clarification of the connection between our interpretation and decision-making**
> > **"Connections to decision-making and neuronal stimulus-response mechanisms"**
>
> **Response**: We thank Reviewer Ewa3 for these insightful comments, which we find highly valuable for clarifying and improving the exposition of both our interpretation and the method. Below, we clarify the connections between *our interpretation* and *decision-making*.
>
> Our goal was to understand possible mechanistic causes of *non-local tension* phenomenon and to distill methodological principles that can guide the design of remedies. We found that interpreting the neural affine–activation pipeline through a decision-making lens gives an intuitively accessible account of *non-local tension* (as outlined in the manuscript) and naturally suggests activation-scaling–based solutions.
>
> More concretely, our interpretation is largely inspired by *grey relational analysis*, a typical *multi-criteria decision-making* model. The connection can be summarized as follows:
>
> 1. **Objective.** In *multi-criteria decision-making*, given a task, the objective is to score the importance of *alternative solutions* based on a set of *criteria* (typical examples include profit-type indicators, where larger values are better, and cost-type indicators, where smaller values are better), and then recommend solutions according to their overall scores.
>
> 2. **A special problem setting.** Grey relational analysis focuses on a special situation: when the given criteria are not directly comparable in terms of their raw numerical magnitudes, how can we still evaluate and rank alternatives? One strategy is that, if we can identify one or more *ideal solutions* under the given criteria, then we can score each alternative by its similarity to these ideal solutions.
>
> 3. **An analogy: viewing the neural affine–activation pipeline as an *multi-criteria decision-making* problem.** We view the neural affine–activation pipeline as a natural extension of this setting:
>    - **Task:** optimize the neuron's weights, in particular each filter (weight vector).
>    - **Ideal solution(s):** the filter vectors themselves, as carriers of learned meaningful patterns; once sufficiently trained, each filter approximates an underlying “ideal pattern”.
>    - **Alternative solutions:** the feature vectors, which provide candidate update directions for each filter.
>    - **Criteria:** the channels, since both features and filters are represented as channel-wise vectors.
>    - **Alternative-to-ideal similarity:** the affine transformation (inner product plus bias), which we treat as a generalized similarity that also considers the magnitudes of the feature and filter.
>
> From this viewpoint, the raw channel values of a feature vector do not directly tell us how “important” that feature is for updating any given filter. Instead, its importance for a specific filter is reflected by the affine projection onto that filter. This clarifies how we bridge the neural affine–activation pipeline to the *multi-criteria decision-making* scenario above.
>
> On top of this analogy, the manuscript explains how we interpret *non-local tension* under this decision-making lens and why learning flexible activation scales (FleS) is a natural way to alleviate it: activation scaling implements a task-guided reallocation of **importance** budget across features and channels, analogous to redistributing decision-making weight according to their utility.

---

> ### Author Response · Authors · 2025-11-24
> **Response to Reviewer Ewa3 (6)**
>
> **W2.3: Clarification on the use of MLPs for constructing the indicators**
> > **"The connection between the theory and the practical algorithm  is tenuous (given the use of MLPs in the final algorithm)"**
>
> **Response**:  We understand Reviewer Ewa3's concern that our use of MLPs may appear disconnected from the preceding theoretical discussion. As further exposition, we summarize two key intuitions suggested by the decision-making interpretation:
>
> 1. **Extracting and translating importance descriptors is central.** A key challenge in mitigating *non-local tension* is to extract descriptors that numerically reflect the inter-channel importance of pre-activations and then translate these descriptors into activation scales. In other words, the “interpretation side” points us to the problem of how to construct and use channel-wise statistics that genuinely reflect feature importance for scaling.
>
> 2. **Sign-aware recalibration matters.** In the affine–activation pipeline, the affine projection provides an initial signal of how strongly each input feature should influence the update direction of a given filter. However, this signal alone is not sufficient. In a single iteration, the magnitudes of feature-induced gradients are key in determining how the filter is updated. When features with positive and negative affine projections onto a filter induce gradients with comparable magnitude but opposite effect, their contributions can partially cancel out, making it harder for the filter to follow an update trajectory dominated by informative features. This motivates a *sign-aware recalibration* step before we summarize channel statistics for scaling.
>
> To evaluate these intuitions, we first constructed **FleS-Proto**. On ImageNet, we observed that when clean class-wise channel-importance statistics are available at test time, the induced activation scales yield very large gains. However, when the batch is fully shuffled or the batch size is reduced, these gains shrink markedly or even disappear. This suggests that in realistic settings—where clean and sufficiently rich channel statistics are rarely accessible—**refining** and **translating** noisy channel-importance cues becomes the central practical challenge. **This observation from FleS-Proto motivates the use of a lightweight MLP to refine and translate noisy channel-importance descriptors** into effective activation scales. Because it is applied identically across positions, the MLP is translation-equivariant and well-suited to aggregating fragmented channel-wise cues. Within its receptive field, it also has universal approximation capability, providing enough capacity to map these refined cues to cross-channel activation scales.
>
> ---
>
> **Remarks.**
> We once again thank Reviewer Ewa3 for the careful and insightful review and would be glad to discuss any further comments or perspectives Reviewer Ewa3 may wish to share.

---

> ### Author Response · Authors · 2025-11-28
> **Summary of changes for Reviewer Ewa3**
>
> We once again thank Reviewer Ewa3 for the careful review and insightful comments. We have addressed all of your comments and incorporated the corresponding changes into the revised manuscript.
>
> For ease of reference, the changes made in accordance with your constructive suggestions are summarized below, together with where they appear in the revised manuscript (with all changes highlighted in blue):
>
> - **Section E.7 (pages 27–28)** – for **“Q1: Linear layer vs. MLP”**.
>
> - **Section 5.2, part (3) (page 10) and Section E.5 (pages 26–27)** – for **“W2.1: Empirical investigation of positive–negative gradient contributions”** and **“W2.2: Empirical investigation of FleS’s interaction with batch normalization”** in **“W2: Further empirical validation”, respectively**.
>
> - **For “W3: On exposition and terminology”, we made the following changes throughout the manuscript:
>   **1.** Removed all abbreviations for key terms and replaced them by their full forms, e.g., *Act* → *activation*, *projected responses* → *pre-activations* **(for W3.1)**.
>   **2.** Consistently used full terms when introducing new concepts, e.g., *NLT* → *non-local tension*, *CL* → *convergence limitation*, and *TDGW* → *trivially discriminative gating weights* **(for W3.2)**.
>   **3.** Further elaborated the subsection “Connections to decision-making and neuronal stimulus–response mechanisms” in **Intuition 1.1 in Section 1 (page 2)** to clarify the link between our interpretation and decision-making **(for W3.3)**.
>   **4.** Substantially rewrote the **caption of Figure 1 in Section 3 (page 4)** following your suggestions, improving the overall clarity of the exposition **(for W3.4)**.
>
> - **Section D (pages 22–23)** – for **“W1. Further clarification on our use of the positive-only indicator”**.
>
> - **Paragraph “Intuitive elaboration on using MLPs” in Section E.7 (pages 27–28)** – for **“W2.3. Clarification on the use of MLPs for constructing the indicators”**.
>
> We are deeply grateful for the time and care Reviewer Ewa3 invested in reading our manuscript and engaging with its details. Your careful review and thoughtful suggestions have substantially contributed to both the clarity and the scope of this work.

---

### Official Review · Reviewer_rpDy · 2025-11-08

**Soundness:** 3
**Presentation:** 4
**Contribution:** 3
**Rating:** 8
**Confidence:** 4

**Summary:**

The paper identifies a limitation in self-gated activation, which is argued to be the reason for limited effectiveness of self-gated activation in transformers due to saturation of gating components. This introduces what the authors call convergence limitation specifically in high-importance features wrt a filter where the difference between the importance become negligible and result in the tendency to lose discriminability wrt contributions of features. In transformers, this collapse of gating discriminability causes activation to neutralize contextual cues the architecture tries to capture.

The authors proposed flexible scaling in self-gated activation using horizontal and vertical dynamic scaling. Horizontal scaling shifts or stretches the gating curve to avoid saturation and vertical scaling increases the discriminability by increasing the range of gating values. The scaling coefficients are conditioned on channel-wise statistics of positively contributing features. In practice, the proposed activation, called FleS, computes per-channel effective responses, feeds them through lightweight MLPs, and outputs the two scaling coeffs. The authors benchmarked the proposed activation across ImageNet, CIFAR-100, long-tailed recognition, COCO detection, and GLUE where FleS consistently outperformed SOTA activations. Particularly notable are results in Swin-Transformer models.

**Strengths:**

1. Clear identification of an important issue

The underlying cause of non-local tension problem was clearly discussed, something that previous works have not articulated.

2. The logical framing of the problem, its cause and the proposed approach

The paper provides an intuitive interpretation of activations as "importance modulators". Then clearly identifies the harm of saturation in self-gating activations and draws a logical connection from convergence limitation, trivially discriminative gating weights phenomenon, and non-local tension problem.

3. Practical design of activation with strong empirical results across different models

FleS is simple, seems to be lightweight, and can easily be dropped into modern architectures. Performance improvements especially in Swin-Micro and Swin-T are substantial. The improvements in experiments in Metaformers, CNNs, detection backbones, and long-tailed classification are promising. This broad applicability suggests that the identified problem is real and not confined to a narrow architecture.

**Weaknesses:**

1. Some theoretical claims rely on partially informal assumptions with more room for quantification/formalization:

Although the results generally support the narrative, but some justification or quantification can show if attention-enhanced features regularly fall into the saturation regime. Also, the explanation for why positive-only feature responses should dominate importance is intuitive, but remains heuristic; the decision-theoretic interpretation could be formalized more rigorously.

2. Insufficient analysis of optimization stability and dynamics

The paper mentions initializing \gamma values but doesn't quantify sensitivity to initialization. Given that activations can strongly shape optimization trajectories, this lack of investigation is a methodological gap

3. Dependence on channel-level statistics require more investigation/illustration of failure scenarios
The batch-dependence in channel statistics brings up questions about microbatch regimes, distributed training, highly-multimodal batches.

4. Discussion of potential failures/limitations or stress-tests:
In addition to gains in performance, it's worth discussion more about limitations and scenarios that the proposed activation may fail to be useful.

**Questions:**

1. Can the authors quantify how often real Transformer activations fall into the saturation regime?

2. How do $\kappa_h$ $\kappa_o$ evolve during training? Please include training curves of these scalars and variance across layers.

3. Is FleS stable in small-batch regimes? Also, how does FleS interact with batch normalization.

4. The paper gives an intuition but more follow-ups on why to exclude negative values: What happens in architectures where negative values carry semantic meaning? Have the authors visualized the gradient contributions from negative vs positive responses?

5. Is there any other costs other than Flops to be discussed and compared? This becomes important especially when impact of MLP size is also discussed.

---

> ### Author Response · Authors · 2025-11-21
> **Response to Reviewer rpDy (1)**
>
> We sincerely thank Reviewer rpDy for the positive evaluation and the insightful suggestions on further scientific investigation, which we find highly valuable for improving our work.
> We address the comments below and will incorporate the corresponding revisions into a revised manuscript, which we will upload as soon as it is complete.
>
> **W1 & Q1: On the saturation regime**
> > **"Can the authors quantify how often real Transformer activations fall into the saturation regime?"**
>
> **Setup.**
> We conduct a targeted investigation on ImageNet using the original Swin-Micro and PoolFormer-S12, adopting a 120-epoch schedule to efficiently demonstrate the trend and key phenomena at this stage.
> At each epoch, for all MLP blocks, we measure the fraction of activations whose gated weights $\rho(\tilde{x})$ exceed 0.9 (i.e., GELU in this experiment, where $0<\rho(\tilde{x})<1$) and treat this as the positive-side saturation threshold. Statistics are aggregated per hierarchical stage (4 stages in total).
>
> **Phenomena.**
> - Swin-Micro. Early in training, saturation is clearly depth-dependent: in the first few epochs, only 0.1%–0.2% of activations are saturated in stages 1–2, vs. 0.6% in stage 3 and 2.5%–3% in stage 4. As training proceeds, stages 2–4 stabilize around about 1%–2%, while stage 1 increases to 4%–5%. On average, stages 2–4 stay in a moderate range (around 1%–2%), with stage 1 rising from almost 0 to a higher level.
> - PoolFormer-S12. Using the same protocol, PoolFormer-S12 shows stronger saturation, especially in deeper stages. In the early epochs, stages 3–4 already reach roughly 3% and 6%–7%. During mid training, saturation spreads more evenly: stage 1 is about 4%–5%, while stages 2–4 remain around 2%–3%. After the middle stage, these ratios become relatively steady.
>
> The saturation curves will be included in the revised manuscript to illustrate this phenomenon more clearly.
>
> **From the NLT perspective.**
> Non-local effects tend to accumulate in deeper blocks in the early stage of training. Before the model has stably acquired useful knowledge, deeper blocks operate at a higher semantic level, so non-local information aggregates there and drives a subset of features into the high-activation (and saturated) region. As training enters the mid–late phase, the model converges to a more effective pattern: each stage allocates activation mass according to its role, saturation ratios stabilize, and their fluctuations shrink. This trend is similar for Swin-Micro and PoolFormer-S12, and, **except for the very first few epochs, a non-trivial fraction of features remains in the saturation regime throughout training.**
>
> ---
>
> **W2: On the Initialization of $\gamma_{ve}$ & $\gamma_{ho}$**
> > **"The paper mentions initializing $\gamma$ values but doesn't quantify sensitivity to initialization"**
>
> In the original submission, we chose $\gamma_{ve},\gamma_{ho}=0.6$, since it yields an identity-safe configuration where $\kappa_{ve},\kappa_{ho}$ are initialized close to 1.0 (with Softplus non-negative constraint), keeping FleS near the baseline activation and avoiding aggressive changes to the optimization landscape at initialization. Previously, we conducted a partial ablation study (not included in the manuscript) to validate our choice of initialization for $\gamma_{ve}$ and $\gamma_{ho}$.
>
> To more systematically study sensitivity, we conducted a new ablation on ImageNet with the Swin-Min backbone, varying the initialization of $\gamma_{ve}$ and $\gamma_{ho}$:
>
> | Activation | Backbone | Init. $\gamma_{ve},\gamma_{ho}$ | Init. $\kappa_{ve},\kappa_{ho}$ | Top-1 (%) |
> |---|---|---|---|---|
> | GELU | Swin-Min | ---  | ---  | 68.7 |
> | FleS | Swin-Min | 0.6 (default) | ~1.0375 (default) | **71.4** |
> | FleS | Swin-Min | 0.0 | ~0.6931 | 71.2 |
> | FleS | Swin-Min | -1.0 | ~0.3133 | *70.8* |
> | FleS | Swin-Min | 1.0 | ~1.3133 | 71.4 |
> | FleS | Swin-Min | 1.4 | ~1.6204  | 71.2 |
> | FleS | Swin-Min | 1.7 | ~1.8678  | 71.3 |
> | FleS | Swin-Min | 2.0 | ~2.1269  | 71.1 |
>
> **Observations.**
> For values around 1.0, FleS is not sensitive to the initialization of $\gamma$: varying $\gamma$ from 0.0 to 2.0 consistently yields significant accuracy gains, all clearly above the GELU baseline. This suggests that AdamW with warmup can absorb moderate changes in the initial gate steepness and range. A more aggressive negative initialization ($\gamma = -1.0$, $\kappa \approx 0.31$) leads to a small yet meaningful drop, likely because the very small initial scaling reduces effective gradients and behaves like an overly small base learning rate in early training; even with AdamW and warmup, the optimizer then struggles to recover an equally good trajectory. Relatively large positive values slightly increase initial steepness but only cause minor, non-systematic changes in accuracy, indicating that the optimizer tolerates moderately over-scaled initial gates.
>
> Based on these observations, we recommend $\gamma_{ve},\gamma_{ho}=0.6$ as the default setting.

---

> ### Author Response · Authors · 2025-11-21
> **Response to Reviewer rpDy (2)**
>
> **Q2: On the evolution of $\kappa_{ho}$ & $\kappa_{ve}$**
> > **"How do $\kappa_{h}, \kappa_{v}$ evolve during training? & Include training curves"**
>
> We sincerely thank Reviewer rpDy for the constructive comments.
>
> **Setup.**
> We run a 120-epoch ImageNet experiment with Swin-Min-FleS and, for each epoch, record stage-wise (4 stages) mean $\kappa_{ho}$ and the mean over the top 10% $\kappa_{ve}$.
>
> **Observations.**
> At epoch 0, both $\kappa_{ho}$ and $\kappa_{ve}$ are close to 1 across all stages. Within the first few epochs they quickly become depth-dependent: stages 2–4 show much stronger scaling, with $\kappa_{ve}$ rising to roughly 4–6 (up to about 7 in stages 3–4) and $\kappa_{ho}$ in stages 2–3 peaking around 2–3, while stage 1 remains more moderate.
>
> After this early phase, stages 1–3 gradually relax to stable, mid-range values. Around epoch 20, stage-1 $\kappa_{ho}$ has decreased from about 3 to around 2, stages 2–3 stabilize near $\kappa_{ho}\approx 1.4$–$1.6$ and $\kappa_{ve}\approx 3$–$4$. In contrast, the deepest stage 4 keeps the strongest scaling: $\kappa_{ho}$ grows from roughly 1 to beyond 4, and $\kappa_{ve}$ stays in a high but stable band around 6–7. The variances exhibit a similar pattern, with larger dispersion in deeper stages early on that later converges.
>
> Overall, the current results indicate a clear depth-dependent gating pattern: mid–shallow stages converge to mild scaling, while the last stage maintains the most aggressive horizontal and vertical scaling, which aligns with our NLT perspective that deeper blocks bear more non-local evidence aggregation. We will complete the full 120-epoch run ASAP and include the complete per-stage curves in the revised manuscript.
>
> ---
>
> **Q3.1 (& W3): Applicability to irregular/small-batch regimes**
> > **"Is FleS stable in small-batch regimes?"**
>
> **Setup.**
> We examine FleS's adaptability to non-standard small batch constructions by applying a special two-stage setting:
> - First 60 epochs: batch size 132, learning rate 3e-4
> - Next 60 epochs: batch size 92, learning rate 2e-4
>
> | Activation | Backbone | #Batch Setting | Top-1(%) ↑ |
> |---|---|:---:|:---:|
> | GELU | Swin-Min | 1024 | 68.7 |
> | FleS | Swin-Min | 1024 | **71.4** |
> | GELU | Swin-Min | 132 + 92 | 68.2 |
> | FleS | Swin-Min | 132 + 92 | **70.3** |
>
> **Observations.**
> Both the GELU baseline and FleS show a small drop in acc. under non-standard batch settings, with a similar degree of degradation. However, FleS still deliver clear improvements over GELU, indicating comparable tolerance but superior overall performance.
>
> **Q3.2: Interaction with BN**
> > **"How does FleS interact with batch normalization"**
>
> We are running a controlled comparison between the Swin-Min and Swin-Min-FleS with BN (by replacing layer normalization in each attention–FFN block) on ImageNet using the default training recipe. The experiments are in progress; we will report the results in a separate response as soon as they are complete (together with our response to Q2).
>
> **Q4.2: Empirical positive–negative gradient contributions**
> > **Have the authors visualized the gradient contributions from negative vs positive responses?**
>
> **Setup.**
> We analyze how the relative gradient contributions of positive vs. negative responses evolve during training on ImageNet with Swin-Min + FleS. For each epoch and each hierarchical stage, we compute the average gradient magnitude on positions where the FleS output is positive and where it is negative, and then take the ratio “positive/negative” as a measure of their relative influence on optimization.
>
> **Observations.**
> We observe two consistent trends:
>
> - Across depth, (averaged) gradients on positive responses are substantially larger than on negative ones. The mean positive-to-negative gradient ratio is about $5.3\times$ in stage 1, $7.9\times$ in stage 2, $12.7\times$ in stage 3, and $13.8\times$ in stage 4.
> - Across training, this asymmetry strengthens from early to mid training and then remains high:
>   - Early phase (epochs 0–9): the ratios are around $3.1\times$, $4.1\times$, $5.5\times$, and $8.5\times$ for stages 1–4.
>   - Mid. phase (epochs 40–79): they grow to roughly $5.6\times$, $8.5\times$, $13.3\times$, and $14.5\times$.
>   - Late phase (epochs 80–119): they stay high at about $4.9\times$, $6.7\times$, $10.3\times$, and $14.4\times$, all clearly above the early-phase values.
>
> Overall, as training proceeds, positive responses increasingly dominate the effective gradient budget, especially in deeper stages, while negative responses carry much weaker gradients. This empirical pattern helps clarify our intuition behind FleS's design for indicators: the main optimization signal is concentrated on the positive side, and emphasizing it in the indicator avoids neutralization effects while still allowing negative features to be exploited by other filters in the current layer **(further clarified in Q4.1)**. We will include the curves in the manuscript for visualization.

---

> ### Author Response · Authors · 2025-11-21
> **Response to Reviewer rpDy (3)**
>
> **Q4.1: On the interpretation and utility of negative-valued pre-activation responses**
> > **"What happens in architectures where negative values carry semantic meaning?"**
>
> **Response**: We appreciate Reviewer rpDy's constructive comments on the applicability of the positive-only indicator. We acknowledge that these aspects were not sufficiently emphasized in the original submission and thank the reviewer for bringing them to our attention.
>
> It is worth noting that our key modeling principle to constructing channel-wise indicators in FleS is to apply **sign-aware recalibration that monotonically emphasizes positive features and suppresses negative ones** prior to summarizing pre-activation responses. The **positive-only indicator** is the **simplest** instantiation of this idea for distinguishing positive and negative responses. More specifically, **we suggest choosing between the positive-only indicator and its softer alternatives based on the application scenario**: for ImageNet-style vision benchmarks, a positive-only design suffices, as these benchmarks typically exhibit only mild semantic variation in appearance, so we adopt the positive-only indicator as the default, very low-cost configuration. In contrast, for NLP tasks with more abrupt token-level semantic changes, we adopt a Softplus-based indicator (Appendix B.2), which offers a more stable behavior. **Therefore, in applications that demand more numerically safe behavior—for example, *when negative values carry informative semantic meaning*—we recommend using the two additional instantiations described in Appendix C.4, with the Softplus-based variant being generally preferred**. In the revised manuscript, *we will make these modeling principles and the replaceable nature of the indicator more explicit.*
>
> As a further discussion of how our interpretation views positive and negative pre-activation responses, we provide a clearer clarification below.
>
> First, we discuss based on a self-gated activation: $\phi(\tilde{x})=\rho(\tilde{x})\cdot\tilde{x}$, where $\rho(\tilde{x}) \ge 0$. We interpret $\rho(\tilde{x})$ as a soft importance score indicating how strongly a response $\tilde{x}$ should be retained. In this perspective, *positive* and *negative* responses are not globally absolute, but relative to a given filter. For example, suppose the current layer includes two filters $ \boldsymbol{w}_1, \boldsymbol{w}_2 \in \mathbb{R}^C $ with $ C \ge 2 $. For an input feature vector $ \boldsymbol{x} \in \mathbb{R}^C $, the corresponding output $ \tilde{\boldsymbol{x}} \in \mathbb{R}^2 $ has two channels. The same $ \boldsymbol{x} $ can be strongly suppressed by $ \boldsymbol{w}_1 $ (i.e., $ \tilde{x}_1 = \langle \boldsymbol{x}, \boldsymbol{w}_1 \rangle < 0 $, omitting the bias term for simplicity) while still being informative for $ \boldsymbol{w}_2 $ (i.e., $ \tilde{x}_2 = \langle \boldsymbol{x}, \boldsymbol{w}_2 \rangle > 0 $). Being down-weighted in one channel therefore does not mean that the feature is discarded by the layer; it may be emphasized in other channels whose filters align better with it.
>
> Based on this understanding, **we propose that a key principle for constructing channel-wise indicators in FleS is to introduce differential feature rectification that mitigates confusion caused by positive–negative cancellation, by applying a monotonic (sign-aware) recalibration that emphasizes positive features and attenuates negative ones before summarizing pre-activation responses.** This helps the statistics reflect genuine per-channel intensity, rather than confusing truly weak channels with channels that have strong but mixed-sign responses. In contrast, aggregating both positive and negative responses in a symmetric way may result in “$-1+1$”-style cancellation effects, which in turn increases the burden on the MLP to predict scaling factors from the indicator, because the input no longer faithfully reflects filter-level importance statistics (Table 4 (right) empirically validates this). *Notably*, even if a response is negative for the dominant filter of one channel, the same feature can still be positively emphasized in other channels; FleS only rectifies the statistics used for scaling, not the global availability of features.
>
> More interestingly, motivated by the reviewer’s question, **we concretize (under the decision-making lens) the case where negative values carry informative semantic meaning as follows: in sufficiently complex applications, before filters have been updated enough to reliably indicate feature importance, the sign of a pre-activation response (positive vs. negative) no longer reliably reflects utility**. Under this mechanistic view, we therefore **recommend using a numerically safer choice rather than aggressively maximizing efficiency.**
>
> We would like to thank Reviewer rpDy again for helping us improve the clarity of our work by raising this insightful question.

---

> ### Author Response · Authors · 2025-11-21
> **Response to Reviewer rpDy (4)**
>
> **W4: Potential limitations**
> > **"Discussion of potential failures/limitations or stress-tests: In addition to gains in performance, it's worth discussion more about limitations and scenarios that the proposed activation may fail to be useful"**
>
> **Response**: We sincerely thank Reviewer rpDy for this thoughful suggestion.
>
> **Setup.**
> We note one practical consideration for the current vision instantiation of FleS in dense recognition tasks: a spatial window size needs to be chosen for computing the indicator statistics.
> In practice, we select this window empirically based on the image resolution and the typical pixel extent of objects.
> For example, on COCO we adopt a default $9\times15$ window. This additional design choice slightly increases deployment burden.
> To assess the sensitivity to this setting, we vary the window size on COCO detection using PoolFormer-S12 with RetinaNet.
>
> **Observations.**
> We observe that FleS is reasonably robust when reducing the window from $9\times15$ to $9\times9$ (mAP 36.2→36.1), but a smaller $5\times5$ window leads to a noticeable drop, while still outperforming the GELU baseline.
>
> As a possible approach to mitigate this limitation, we consider infusing further adaptability by introducing learnable patch statistics in the indicator module. Currently, we evaluate a variant that replaces the hard “positive-only + rectified mean” aggregation with a Softplus-based recalibration followed by a depthwise convolution with a fixed kernel size (optionally using dilation to enlarge the effective receptive field). This continues the idea behind moving from FleS-Proto to FleS: previously we used token-wise MLPs to mitigate noisy batch statistics, whereas here we use adaptive local depthwise convolutions to better capture scale-related per-channel cues. On ImageNet classification, this modified indicator improves PoolFormer-S12-FleS from 79.4% to 79.9% Top-1 (denoted as FleS-Mod below), compared to 77.2% Top-1 for the original PoolFormer-S12, and it also shows slight improvements on COCO. We plan to further investigate to what extent this design direction can alleviate the window-size limitation on downstream dense visual recognition tasks.
>
> *Comparative evaluation on MS COCO object detection (PoolFormer-S12 encoder with RetinaNet).*
>
> | Activation | Window-size | mAP (%)↑ | AP50 (%)↑ | AP75 (%)↑ | APS (%)↑ | APM (%)↑ | APL (%)↑ |
> |---|---|---|---|---|---|---|---|
> | GELU | — | 35.5 | 55.5 | 37.5 | 19.5 | 38.7 | 46.3 |
> | FleS | $9\times 15$ | 36.2 | 57.0 | 38.1 | 20.7 | 40.1 | 46.8 |
> | FleS | $9\times 9$ | 36.1 | 56.6 | 37.9 | 20.5 | 40.0 | 46.8 |
> | FleS | $5\times 5$ | 35.8 | 57.0 | 37.5 | 20.0 | 39.6 | 46.7 |
> | FleS-Mod | $5\times 5$ | 36.3 | 56.9 | 38.3 | 20.7 | 40.3 | 47.0 |
>
> ---
>
> **Q5: Costs other than FLOPs**
> > **"Is there any other costs other than Flops to be discussed and compared?"**
>
> **Response**: We appreciate Reviewer rpDy's constructive comment.
>
> In response, and going beyond theoretical FLOPs, we profile the practical runtime overhead introduced by FleS. We measure inference throughput on a single RTX 3090 GPU under a common `torch.compile` environment for Swin-Min, Swin-Micro, and Swin-Tiny on ImageNet:
>
> | Activation | Backbone | FLOPs | Throughput (img/s) | Top-1 (%) ↑ |
> |---|---|:---:|:---:|:---:|
> | GELU | Swin-Min | 1.6G | 4207.2 | 68.7 |
> | FleS | Swin-Min | 1.6G | 4011.3 | **71.4** |
> | GELU | Swin-Micro | 2.6G | 2775.6 | 78.7 |
> | FleS | Swin-Micro | 2.6G | 2616.8 | **80.3** |
> | GELU | Swin-Tiny | 4.4G | 1622.5 | 81.3 |
> | FleS | Swin-Tiny | 4.4G | 1545.2 | **82.3** |
>
> FleS incurs an acceptable, modest overhead of about 4–6% in practical efficiency across these backbones, while consistently improving Top-1 accuracy by a substantial margin.
> ___
>
> ***Remarks***
> We once again thank Reviewer rpDy for the careful review and insightful comments, and would be glad to discuss any further comments or perspectives Reviewer rpDy may wish to share.

---

> ### Author Response · Authors · 2025-11-25
> **Response to Reviewer rpDy (5)**
>
> We thank Reviewer rpDy for your patience.
> We have now completed the experiments for **Q2: On the evolution of $\kappa_{ho}$ and $\kappa_{ve}$** and **Q3.2: Interaction with batch normalization**, and we report the corresponding observations below.
>
> **Q2: On the evolution of $\kappa_{ho}$ & $\kappa_{ve}$**
> > **"How do $\kappa_{h}, \kappa_{v}$ evolve during training? & Include training curves"**
>
> **Response**: We sincerely thank Reviewer rpDy for the constructive comments. In accordance with Reviewer rpDy's suggestions, we address this question below.
>
> **Setup.**
> We conduct a targeted 120-epoch ImageNet experiment with Swin-Min-FleS and, for each epoch, record stage-wise (4 stages) mean $\kappa_{ho}$ and the mean over the top 10% $\kappa_{ve}$values.
>
> **Observations.**
> At epoch 0, both $\kappa_{ho}$ and $\kappa_{ve}$ are close to 1 across all four stages, i.e., FleS starts from an approximately identity-like scaling. Within the first few epochs, they quickly become depth-dependent. In stages 2–3, $\kappa_{ve}$ (vertical scaling factor) rises from $\approx 1$ to about $2.4$–$3.1$, and $\kappa_{ho}$ (horizontal scaling factor) rises to around $5$–$7$. Stage 1 also strengthens but more moderately, with $\kappa_{ve}$ peaking at $\approx 3.1$ and $\kappa_{ho}$ at $\approx 2.6$. Stage 4 shows the most aggressive horizontal scaling early on: $\kappa_{ho}$ quickly grows from $\approx 1.1$ to about $7.4$, while $\kappa_{ve}$ increases more gradually over a longer period, eventually peaking around $5.0$.
>
> After this early phase, stages 1–3 gradually relax to stable, mid-range values. By the later part of training (e.g., epoch 119), stage-1 $\kappa_{ve}$ has decreased from its peak $\approx 3.1$ to $\approx 1.23$, and $\kappa_{ho}$ stabilizes around $\approx 2.47$. Stages 2–3 similarly converge to milder scaling, with $\kappa_{ve}$ around $\approx 1.06$–$1.31$ and $\kappa_{ho}$ around $\approx 2.85$–$3.78$. In contrast, the deepest stage 4 consistently maintains the strongest scaling throughout training: even at epoch 119, it still has the largest values, with $\kappa_{ve} \approx 2.14$ and $\kappa_{ho} \approx 4.68$. The variances exhibit a similar depth-dependent pattern: deeper stages show larger dispersion in the early epochs (especially stage 4), which gradually contracts as training converges.
>
> Overall, the full 120-epoch run confirms a clear **depth-dependent gating** pattern: mid–shallow stages converge to mild scaling, while the last stage maintains the most aggressive horizontal and vertical scaling. This aligns with our NLT perspective that *deeper blocks bear more of the non-local evidence aggregation.* The full per-stage $\kappa_{ho}$ and $\kappa_{ve}$ curves will be demonstrated in the revised manuscript.
>
> ---
>
> **Q3.2: Interaction with batch normalization**
> > **"Also, how does FleS interact with batch normalization"**
>
> **Response**: We sincerely thank Reviewer rpDy for raising this insightful question. In response, we address this question below.
>
> **Setup.**
> We conduct a controlled comparison on ImageNet with Swin-Min, where we replace layer normalization (LN) by batch normalization (BN) in each Transformer block, using the default training recipe for all variants.
>
> **Observations.**
> BN improves the original Swin-Min by a modest but noticeable margin (Top-1: 68.7 $ \rightarrow $ 69.3).
> When combined with FleS, however, BN yields a markedly larger gain (71.4 $ \rightarrow $ 73.8), which is notable given that FleS already operates in a higher-accuracy regime with stronger diminishing returns.
> We are interested in interpreting this unexpected phenomenon, which may indicate an interesting interaction between FleS's activation scaling and batch-normalized feature statistics. A deeper analysis is beyond the scope of the present discussion, but it may provide a promising direction for our further investigation.
>
> | Activation | Backbone | Normalization | #Params | FLOPs | Top-1 (%) $\uparrow$ |
> |---|---|---|:---:|:---:|:---:|
> | GELU | Swin-Min | LN | 11.8M | 1.6G | 68.7 |
> | FleS | Swin-Min | LN | 13.0M | 1.6G | **71.4** |
> | GELU | Swin-Min | BN | 11.8M | 1.6G | *69.3* |
> | FleS | Swin-Min | BN | 13.0M | 1.6G | ***73.8*** |

---

> ### Author Response · Authors · 2025-11-28
> **Summary of changes for Reviewer rpDy**
>
> We once again thank Reviewer rpDy for your careful review and insightful comments.
> We have addressed all of your comments and incorporated the corresponding changes into the revised manuscript.
>
> For ease of reference, the changes made in accordance with your constructive suggestions are summarized below, together with where they appear in the revised manuscript **(with all changes highlighted in blue)**:
>
> - **Section E.1 (pages 23-24) – for “W1 \& Q1: On the saturation regime”)**;
> - **Section E.2 (pages 24-25) – for “W2: On the initialization of $\gamma_{ve}$ and $\gamma_{ho}$”)**;
> - **Section E.3 (pages 25-26) – for“Q2: On the evolution of $\kappa_{ho}$ and $\kappa_{ve}$”)**;
> - **Section E.4 (page 26) – for “Q3.1 (\& W3): Applicability to irregular/small-batch regimes”)**;
> - **Section E.5 (pages 26-27) – for “Q3.2: Interaction with batch normalization”)**;
> - **Section L (page 38) – for “W4: Potential limitations”)**;
> - **Section D (pages 22-23) – for “Q4.1: On the interpretation and utility of negative-valued feature pre-activation responses”)**;
> - **Section 5.2, part (3) (page 10) – for “Q4.2: Empirical positive–negative gradient contributions”)**;
> - **Section E.6 (page 27) – for “Q5: Costs other than FLOPs”)**.
>
> We are deeply grateful for the time and care Reviewer rpDy invested in reading our manuscript and engaging with its details.
> Your careful review and thoughtful suggestions have substantially contributed to both the clarity and the scope of this work.

---

### Author Response · Authors · 2025-12-02
**Rebuttal and Revision Roadmap**

We conducted a comprehensive revision of the entire manuscript to fully incorporate the constructive feedback from all reviewers, with all changes highlighted in blue. Below, we provide a roadmap that links each reviewer’s concerns to the corresponding rebuttal and revised sections in the manuscript.

---

**W** = **Weakness**; **Q** = **Question**; **A** = **Answer**.

**Reviewer 1 (rpDy)**

- *W1 & Q1.* (On saturation regime): [A1] Added stage-wise saturation curves for Swin-Micro and PoolFormer-S12 to quantify how often activations saturate (Sec. E.1).

- *W2.* (On the initialization of $\gamma_{ve}$ and $\gamma_{ho}$): [A2] Added an ablation over multiple initializations on Swin-Min, showing stable gains and supporting $\gamma_{ve}=\gamma_{ho}=0.6$ as default (Sec. E.2).

- *Q2.* (On the evolution of $\kappa_{ho}$ and $\kappa_{ve}$): [A3] Logged per-stage $\kappa_{ho}$ and $\kappa_{ve}$ over 120 epochs on Swin-Min-FleS, revealing depth-dependent gating (Sec. E.3).

- *Q3.1 & W3.* (On applicability to small-batch regimes): [A4] Tested a two-stage small-batch schedule where GELU and FleS both degrade, yet FleS retains clear gains (Sec. E.4).

- *Q3.2.* (On interaction with BN): [A5] Compared LN and BN in Swin-Min with/without FleS, showing BN gives a modest baseline gain and a larger gain with FleS (Sec. E.5).

- *W4.* (On potential limitations): [A6] Discussed dependence on window size for dense recognition, added a COCO window-size ablation, and introduced an adaptive FleS-Mod variant (Sec. L, Sec. E.8).

- *Q4.1.* (On negative-valued pre-activations): [A7] Clarified the sign-aware indicator principle, when to use positive-only vs Softplus-based options, and that FleS rectifies statistics rather than discarding features (Sec. D).

- *Q4.2.* (On gradient contributions): [A8] Analyzed gradient magnitudes of positive vs negative outputs across depth and training, showing positives increasingly dominate the gradient budget (Sec. 5.2(3)).

- *Q5.* (On costs beyond FLOPs): [A9] Reported throughput for Swin families with FleS vs GELU, showing modest runtime overhead (Sec. E.6).

---

**Reviewer 2 (Ewa3)**

- *Q1.* (On linear layer vs MLP in FleS): [A1] Compared the FleS MLP with a low-rank linear variant (FleS-LRL) on Swin-Min/Micro, finding the MLP consistently stronger at similar cost (Sec. E.7).

- *W2.* (On further empirical validation): [A2] Added (i) a gradient-contribution probe for positive vs negative outputs, and (ii) an LN→BN comparison with/without FleS, showing BN helps FleS more than baseline (Sec. 5.2(3), Sec. E.5).

- *W3.* (On exposition and terminology): [A3] Improved exposition by removing non-standard abbreviations, standardizing key terms (e.g., “non-local tension”, “convergence limitation”), and rewriting the Fig. 1 caption and text (Sec. 1, Fig. 1).

- *W1.* (On the positive-only indicator): [A4] Stated sign-aware recalibration as the core principle and made explicit softer, safer alternatives (e.g., Softplus-based indicators) when negative values are informative (Sec. D).

- *W3.3.* (On the decision-making connection): [A5] Clarified the multi-criteria decision-making / grey-relational view of the affine–activation pipeline and how this lens motivates FleS’s flexible scaling (Sec. 1).

- *W2.3.* (On the use of MLPs for indicators): [A6] Used FleS-Proto to show the need to refine noisy descriptors and justified a small translation-equivariant MLP as a lightweight map from descriptors to scales (Sec. E.7).

---

**Reviewer 3 (HdQ3)**

- *W4.* (On elaboration of the decision-theoretic perspective): [A1] Please refer to Reviewer 2 (Ewa3), W3.3.

- *W2.* (On practical efficiency): [A2] Please refer to Reviewer 1 (rpDy), Q5.

- *W3 & Q3.1.* (On hyperparameter sensitivity): [A3] Studied window-size sensitivity on COCO with PoolFormer-S12 + RetinaNet and introduced FleS-Mod as an adaptive option (Sec. E.8).

- *Q1.* (On interaction with normalization): [A4] Please refer to Reviewer 1 (rpDy), Q3.2; additionally tested BN/LN on the indicator input and kept the vanilla design (Sec. E.9).

- *Q2.* (On the scope of non-local tension beyond Transformers): [A5] Argued that gains on classical CNNs mainly arise from adaptive scaling, while ablations show the sign-aware indicator remains crucial over other dynamic ones (Sec. E.10).

- *Q3.2.* (On token-level indicators and 1D depthwise conv in NLP): [A6] Motivated token-level indicators and 1D depthwise conv to capture local context under abrupt token semantics while preserving sign-aware recalibration (Sec. K).

- *Q4.* (On FleS with 100M+ models): [A7] Applied FleS to a 100M+ Hiera-Large-Slim, obtaining a meaningful gain (Sec. E.11).

- *Q5.* (On version consistency and a unified module): [A8] Pointed to FleS-Mod as a unified, stable template when a single practical version is preferred (Sec. M).

- *W1.* (On a unified analysis framework for activations): [A9] Summarized activations and FleS within the decision-making view (Sec. 5).

---

### Author Response · Authors · 2025-12-02
**Author Remarks to the Area Chair**

We sincerely thank the Area Chair for the time and effort devoted.

To facilitate your assessment, we provide a brief core storyline, a summary of reviews and responses, and a roadmap of revisions.

---

### **Core Storyline with *Contributions***

- ***Problem identification* – A new puzzling observation.** We empirically identify and characterize a critical, previously unstudied issue in self-gated activations, which we term **non-local tension**: when non-local cues are modeled both *inside* the activation and by surrounding non-local operators, their benefits **hardly accumulate**, so the extra non-local gain of self-gated activations on Transformers is largely *neutralized*.

- ***Theoretical analysis tool* – Making the puzzle transparent.** We develop a theoretical analysis tool that reinterprets the affine–activation pipeline through a **grey decision-making lens**: filters act as updatable ideal alternatives, feature vectors as realistic alternatives, channels as criteria, and pre-activations as importance scores. Under this view, once non-local operators have amplified important features, standard self-gated activations whose weighting functions saturate push these responses into a high-activation, low-slope regime, leaving the extra boost from non-local cues effectively *neutralized*—this dynamic underpins non-local tension.

- ***Practical method* – The first principled remedy.** Guided by this interpretation, we derive **FleS** as a tailored *adaptive scaling remedy*: a self-gated activation that performs dual adaptive scaling (horizontal recalibration of gate steepness and vertical redistribution across channels), so that non-local cues can still modulate activations instead of being choked by saturation.

- ***Empirical validation.*** Across diverse benchmarks and architectures, FleS delivers *consistent and remarkable improvements* over SOTA activations. On ImageNet, FleS yields much larger gains on Transformers / MetaFormers (e.g., +1.0% Top-1 on Swin-T, while other SOTA activations stay < +0.3% at similar cost); on GLUE, FleS(-SeqGate) lifts BERT-T to markedly surpass a much larger (≈2.5×) same-family model, **validating both the effectiveness and transferability of our principled insights**.

---

### **Summary of Reviews and Responses**

**All three reviewers evaluated the paper positively, with scores of 8, 6, and 6**, each meeting or exceeding the acceptance threshold.

Across the three reviews, they recognized and highlighted:

- The **conceptual clarity, originality, and significance** in identifying non-local tension and convergence limitation as key bottlenecks of self-gated activations.
- The **theoretical quality** of the decision-making perspective as a principled lens for analyzing and designing activation mechanisms.
- The **practicality and flexibility** of FleS as a drop-in activation module that improves diverse backbones and tasks.
- The **strong and comprehensive empirical results**, and the effectiveness, generalization, and robustness of FleS across vision and NLP settings and model families.

In the rebuttal, we resolved common concerns by:

- **Accomplishing further relevant scientific investigations.** We deepened the conceptual and empirical analysis of non-local tension and saturation, clarified when and where it appears in modern architectures, traced the evolution of the vertical and horizontal scaling factors ($\kappa_{ve}$ and $\kappa_{ho}$) across depth and training, and quantified positive vs negative gradient contributions, thereby tightening the link between the observed phenomena and the decision-making view behind FleS.

- **Improving the presentation and exposition.** We further clarified our interpretation and methodological insights, standardized terminology and removed non-standard abbreviations, and rewrote the caption and surrounding text of the key illustrative figure so that the core intuition can be understood directly from the abstract and the figure.

- **Completing all requested empirical investigations with clear analysis:** Accordingly, we added the **full set of additional experiments** suggested by the reviewers—including the three scientific investigations above, ablations on the MLP vs. low-rank linear mapping in FleS, sensitivity to the initialization of $\gamma_{ve}$ / $\gamma_{ho}$ and to neighborhood size in dense recognition, interaction with LN/BN and small-batch regimes, evaluations on larger-scale models, and practical cost profiling beyond FLOPs—and provided corresponding analyses, further supporting the robustness, generality, and practical relevance of FleS.

While we had not received any further reviewer responses by Nov. 27, we have **carefully addressed all comments** and **incorporated all corresponding revisions** into the manuscript. We found their questions and suggestions highly constructive, and we believe the revisions significantly improve the manuscript’s clarity, scientific contribution, and experimental breadth.

---

### Meta-Review · Area_Chair_qp92 · 2026-01-05

**Summary:**

The submission identifies a shortcoming present in existing self-gated activation functions and demonstrates how it can be overcome with a novel gating strategy. The reviewers all had a positive assessment of the paper, with a particular emphasis on the originality and significance of the work. The work is considered to be largely sound, but the connection between the theoretical analysis and practical deployment of the approach (and related approaches) was raised.

**Reviewer Concerns:**

The reviewers did not raise any major concerns. Discussion mainly focused on improving clarity of some specific points of the paper and running additional experiments that, while informative, are not crucial to the overall message of the paper.

**Reviewer Scores:**

All reviewers gave a positive score. It seems unlikely they would raise their score given the initial reviews and the responses received.

---

### Decision · Program_Chairs · 2026-01-26

Accept (Poster)